# An extracellular receptor tyrosine kinase motif orchestrating intracellular STAT activation

Katri Vaparanta [1,2], Anne Jokilammi [1,2], Mahlet Tamirat[3],
Johannes A. M. Merilahti [1,2], Kari Salokas [4], Markku Varjosalo [4],
Johanna Ivaska [2,5,6,7,8], Mark S. Johnson[3] & Klaus Elenius [1,2,7,9] ✉

The ErbB4 receptor isoforms JM-a and JM-b differ within their extracellular juxtamembrane (eJM) domains. Here, ErbB4 isoforms are used as a model to address the effect of structural variation in the eJM domain of receptor tyrosine kinases (RTK) on downstream signaling. A specific JM-a-like sequence motif is discovered, and its presence or absence (in JM-b-like RTKs) in the eJM domains of several RTKs is demonstrated to dictate selective STAT activation. STAT5a activation by RTKs including the JM-a like motif is shown to involve interaction with oligosaccharides of N-glycosylated cell surface proteins such as β1 integrin, whereas STAT5b activation by JM-b is dependent on TYK2. ErbB4 JM-a- and JM-b-like RTKs are shown to associate with specific signaling complexes at different cell surface compartments using analyses of RTK interactomes and super-resolution imaging. These findings provide evidence for a conserved mechanism linking a ubiquitous extracellular motif in RTKs with selective intracellular STAT signaling.

Receptor tyrosine kinases (RTK) are a gene family of growth factor receptors that mediate a number of cellular responses, such as proliferation, survival, differentiation, and migration[1]. While the function of most structural domains of RTKs are well-characterized[2], research on the extracellular juxtamembrane (eJM) domain has mainly focused on its role in i) ligand binding specificity[3], ii) RTK cleavage by disintegrin and metalloproteinases (ADAM)[4,5], and iii) glycosphingolipid interactions[6].

ErbB4 is a member of the epidermal growth factor (EGFR)/ErbB subfamily of RTKs, together with EGFR, ErbB2 and ErbB3[7–10]. The natural JM splice variants of ErbB4 provide a unique model for studying the functional relevance of the RTK eJM domain since they stimulate different and sometimes even opposing cellular activities[11–14] but differ only by short amino acid stretches within their membrane-proximal

eJM domains[15]. ErbB4 JM-a promotes survival and JM-b induces apoptosis, when overexpressed in serum-starved fibroblasts[12]. Moreover, ErbB4 JM-a, unlike ErbB4 JM-b, stimulates growth and reduces differentiation in 3D cultures of mammary epithelial cells[13,14]. The JM-a-specific sequence includes a cleavage site for ADAM-17[4], making it susceptible to ectodomain shedding and subsequent release of a soluble intracellular domain (ICD) fragment by gamma-secretase activity[11,16,17]. However, the molecular mechanisms underpinning the isoform-specific and often opposing cellular functions of ErbB4 isoforms remain to be elucidated. Furthermore, it remains a mystery how RTKs, differing only in their eJM sequences, orchestrate distinct intracellular signaling pathways.

Signal transducers and activators of transcription (STAT) are well-characterized intracellular substrates downstream of RTKs[18–25]. The

[1]Medicity Research Laboratories, and Institute of Biomedicine, Faculty of Medicine, University of Turku, 20520 Turku, Finland. [2]Turku Bioscience Centre, University of Turku and Åbo Akademi University, 20520 Turku, Finland. [3]Structural Bioinformatics Laboratory, Biochemistry, Faculty of Science and Engineering, Åbo Akademi University, 20520 Turku, Finland. [4]Helsinki Insititute of Life Science, Institute of Biotechnology, University of Helsinki, 00014 Helsinki, Finland. [5]InFLAMES Research Flagship Center, University of Turku, 20520 Turku, Finland. [6]Department of Life Technologies, University of Turku, 20520 Turku, Finland. [7]Western Finnish Cancer Center (FICAN West), University of Turku, 20520 Turku, Finland. [8]Foundation for the Finnish Cancer Institute, Tukholmankatu 8, 00014 Helsinki, Finland. [9]Department of Oncology, Turku University Hospital, 20521 Turku, Finland. ✉e-mail: klaus.elenius@utu.fi

STATs are the key effectors of the Janus kinase (JAK)/STAT signaling pathway, where the STATs are canonically activated by phosphorylation by the JAKs, JAK1, JAK2, JAK3, or TYK2[21] and subsequent dimerization. The *STAT* gene family consists of 7 genes, *STAT1*, *STAT2*, *STAT3*, *STAT4*, *STAT5a*, *STAT5b*, and *STAT6*[26]. In spite of their highly conserved (96%) primary protein sequences, STAT5a, and STAT5b can carry out both redundant and specific functions[26–28]. RTKs and other kinases have a tendency to selectively activate some, but not all STAT subtypes[21–23]. However, the mechanism underlying the selective STAT subtype activation downstream of RTKs has not been addressed.

Here, we set out to uncover how RTK eJM domains regulate intracellular signaling by utilizing the natural JM isoforms of ErbB4 as a model. We find that RTK eJM motifs dictate receptor localization to distinct plasma membrane microdomains to facilitate isoform-specific STAT activation. JM-b signaling to STAT5b is TYK2-dependent, whereas the JM-a-like eJM sequence activates STAT5a by engaging complex N-glycan decorated cell surface glycoproteins, such as β1 integrins.

## Results

### ErbB4 JM isoforms localize to different cell surface domains
In a quest for mechanisms underlying the functional differences arising from the eJM region, stimulated emission depletion (STED) super-resolution fluorescence microscopy was utilized to explore the subcellular localization of the ErbB4 JM isoforms. To this end, HA- and MYC-tagged ErbB4 JM isoforms as well as the cleavage-resistant ErbB4 JM-a V675A[13,20] mutant were expressed in pairs in MCF-7 cells. ErbB4 JM-b-MYC and ErbB4 JM-a-HA localized to predominantly different cell surface domains and were significantly less colocalized compared to the expression of only the JM-b isoform using the two different tags (JM-b-MYC + JM-b-HA) (Fig. 1a). The cleavage-resistant JM-a V675A-HA localized similarly to wild-type ErbB4 JM-a-HA with limited co-localization with ErbB4 JM-b-MYC at the cell surface (Fig. 1a), indicating that the difference between the JM isoforms was not secondary to their differential susceptibility to proteolytic cleavage at the gamma-secretase-sensitive cleavage site[13,20]. The findings were corroborated with 3D structured illumination microscopy (SIM) of COS-7 cells expressing HA-tagged ErbB4 JM-a or JM-b together with EGFP-tagged ErbB4 JM-a. At the cell periphery, ErbB4 JM-a appeared to localize to small domains higher in the z-plane than JM-b and underneath the cell the two receptor isoforms segregated to distinct punctae (Supplementary Fig. 1). This correlated with significantly lower colocalization of JM-b-HA with JM-a-GFP as compared to JM-a-HA with JM-a-GFP (Supplementary Fig. 1).

### ErbB4 JM-b but not JM-a associates with TYK2
To address whether distinct intracellular signaling results from differential cell surface location of the ErbB4 JM isoforms, proteins selectively associating with the two isoforms were analyzed with mass spectrometry (Supplementary Fig. 2; Supplementary Data 1–4). The analysis was carried out from ErbB4 co-immunoprecipitates of MDA-MB-468 cells expressing ErbB4 JM-a or ErbB4 JM-b. Several differentially regulated signaling pathways induced by ErbB4 JM-a or ErbB4 JM-b were discovered (Supplementary Fig. 3). Since JAK/STAT signaling is one of the well-documented signaling pathways downstream of RTKs[21], the differential interaction of the JAK kinase TYK2 with ErbB4 JM-a and JM-b (Supplementary Fig. 2) was selected for further analysis. Co-immunoprecipitation experiments with MCF-7 cells validated the selective association of TYK2 with ErbB4 JM-b (Fig. 1c) and demonstrated that TYK2 was unique among the JAK kinases in exhibiting this selectivity (Fig. 1d).

### TYK2 is necessary for selective STAT5b activation by ErbB4 JM-b
Since STAT5a is a well-established downstream substrate of ErbB4 JM-a[18–20], the effect of TYK2 knock-down on the activation of STAT5a and the close homolog STAT5b was investigated. RNA interference-

mediated knock-down of TYK2 or JAK2 in MDA-MB-468 cells indicated that ErbB4 JM-a-mediated STAT5a activation (phosphorylation at Y694) was independent of either JAK kinase (Fig. 1e). In contrast, TYK2, but not JAK2, was necessary for ErbB4 JM-b-mediated STAT5b activation (phosphorylation at Y699) ($P < 0.05$, $n = 3$) (Fig. 1f).

To address the contribution of the kinase domains of both ErbB4 as well as TYK2 to STAT5b activation, kinase-dead mutants of ErbB4 JM-b and TYK2 were generated. Substitution of lysine to arginine in the kinase domain of ErbB4 (K741R) and lysine to isoleucine in the kinase domain of TYK2 (K930I) have previously been shown to render the domains inactive[29–32]. Comparison of these kinase-dead mutants and wild-type kinases in MCF-7 cells revealed that only the kinase activity of ErbB4 JM-b, but not TYK2, was necessary for STAT5b activation ($P < 0.05$, $n = 3$) (Fig. 1g). Furthermore, overexpression of the ErbB4 isoforms did not promote TYK2 phosphorylation (Supplementary Fig. 4a–d), suggesting that TYK2 operates as a scaffold in the ErbB4 JM-b/STAT5b activation complex, in a manner independent of TYK2 kinase function.

### Different STAT subtypes are activated by different ErbB4 JM isoforms
Interestingly, ErbB4 JM-a seemed to be more efficient in promoting the phosphorylation of STAT5a, and ErbB4 JM-b in promoting the phosphorylation of STAT5b (Fig. 1d-e). As the natural ErbB4 JM isoforms are not endogenously expressed in the same cell types[15], the observation was validated by overexpressing ErbB4 JM-a or JM-b in four different cell lines – MDA-MB-468, HC11, MCF-7, or COS-7 cells. To minimize any underlying effect from endogenous ErbB receptors, ligand stimulation with neuregulin-1 (NRG-1) was only used in experiments with MCF-7 or COS-7 cells which lack basal ErbB4-mediated STAT5 phosphorylation in the absence of ligand stimulation. Expression of the ErbB4 JM-a isoform promoted phosphorylation of STAT5a at residue Y694, indicating greater STAT5a activation[33] when compared to expression of ErbB4 JM-b in all the four cell lines (Fig. 2a, c). ErbB4 JM-a also demonstrated more efficient association with STAT5a in a proximity ligation assay (PLA) (Fig. 2e, f), and induced more nuclear accumulation of STAT5a (Fig. 2i, j), as compared to ErbB4 JM-b. Expression of ErbB4 JM-b, in contrast, promoted the activation (phosphorylation at Y699) and nuclear accumulation of STAT5b to a greater extent than ErbB4 JM-a (Fig. 2b, d, k–l). A greater physical association was similarly observed between STAT5b and ErbB4 JM-b than ErbB4 JM-a (Fig. 2g, h).

The distinct ability of the ErbB4 JM isoforms to activate STAT5a and STAT5b was independent of the presence or absence of an alternatively spliced exon (CYT-1 *versus* CYT-2 variant[34]) in the cytoplasmic domains of the receptors (Supplementary Fig. 5a, b). Moreover, the selective STAT activation was not restricted to STAT5 subtypes, as ErbB4 JM-b, but not JM-a, activated STAT3 (phosphorylation at Y705) (Supplementary Fig. 5c, d).

In addition to promoting STAT5a signaling, activation of ErbB4 JM-a suppressed STAT5b activity by inducing phosphorylation at the residue Y739 (Supplementary Fig. 5e), which has been suggested to negatively regulate STAT5b activation[35]. In contrast, overexpression of ErbB4 JM-b induced STAT5b phosphorylation only on the activating Y699 residue (Supplementary Fig. 5f). Taken together, these results indicate that ErbB4 JM-a and JM-b trigger distinct intracellular signaling via STATs.

### STAT5a activation by ErbB4 JM-a is independent of cleavage or glycosylation
Next putative mechanisms by which the extracellular eJM domain could account for distinct STAT activation by the isoforms were explored. Since the susceptibility of the ErbB4 JM-a isoform to proteolysis remains the best-characterized biochemical difference between the ErbB4 JM variants[11,15], and the cleavage-

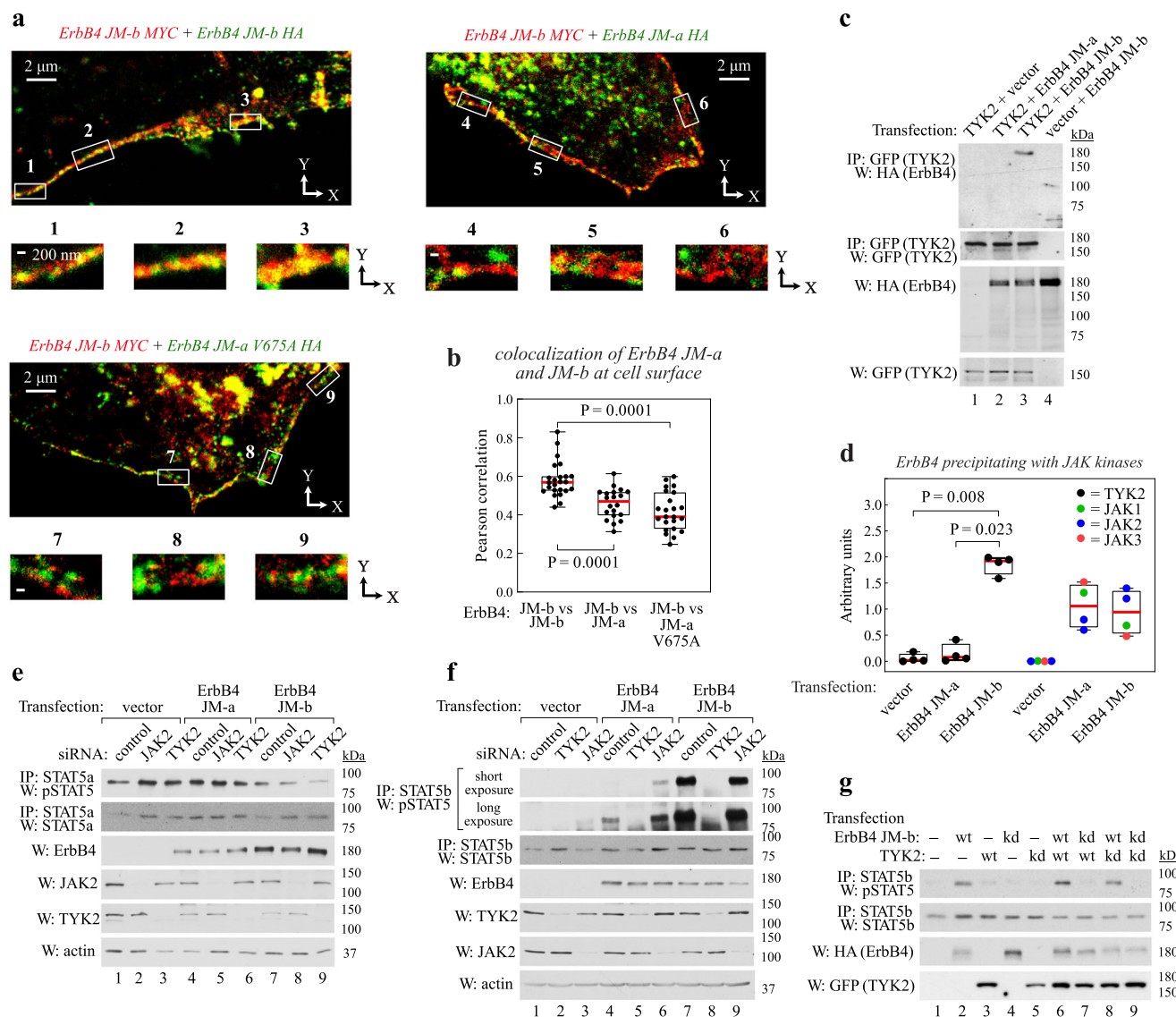

**Fig. 1 | ErbB4 JM isoforms localize to different compartments at the cell surface and associate with different signaling molecules such as TYK2. a, b** STED super-resolution immunofluorescence analysis of MCF-7 cells expressing the indicated MYC-tagged (red) and HA-tagged (green) ErbB4 constructs. The white boxes highlight regions of interest in the x-y plane that are magnified below. Panel B depicts quantification of co-localization of MYC- and HA-specific signals at the cell surface in the x-y plane where each dot represents the correlation of the signals in one cell (n = 20-24) pooled from three independent experiments. One way ANOVA. Benjamini, Krieger and Yekutieli adjusted *P*-values. **c, d** Co-immunoprecipitation analyses of JAK kinase family members and ErbB4 JM isoforms in MCF-7 cells expressing GFP-tagged TYK2 (c), JAK1, JAK2, or JAK3 and HA-tagged ErbB4 JM-a or JM-b after NRG-1 stimulation. Panel **d** depicts densitometric quantification of independent experiments (n = 4). Kruskal Wallis ANOVA. Benjamini, Krieger, and Yekutieli adjusted *P*-values. **e, f** Phosphorylation of STAT5a (**e**) and STAT5b (**f**) in siRNA-treated MDA-MB-468 cells expressing ErbB4 JM-a or JM-b. Cells were treated with siRNAs targeting TYK2 or JAK2 where indicated. Activating STAT5 phosphorylation on Y694/699 was detected. Representative blots of n = 2 (**e**) and n = 3 (**f**) independent experiments. **g** Phosphorylation of STAT5b (on Y699) in MCF-7 cells expressing kinase-dead (kd) or wild-type (wt) TYK2 or ErbB4 JM-b. Cells were stimulated with NRG-1. Representative blots of n = 3 independent experiments. In the boxplots the line represents the median, the box the interquartile range and whiskers the whole range of values. Source data are provided as a Source Data file.

resistant V675A ErbB4 JM-a localized similarly to the wild-type receptor on the cell membrane, the role of ErbB4 JM-a cleavage in STAT5 isoform activation preference was also addressed. Expression of V675A ErbB4 JM-a mutant in mammary epithelial cells suppressed generation of the 80 kD ICD fragment (Supplementary Fig. 6a, b) and enhanced ErbB4 JM-a-mediated STAT5a activation (Supplementary Fig. 6c, d), in accordance with accumulation of the cleavage-resistant full-length receptor (Supplementary Fig. 6a, c). The moderate basal level STAT5b activation observed in cells overexpressing JM-a was, however, reduced in response to expression and accumulation of the non-cleavable JM-a mutant (Supplementary Fig. 6c, d), possibly due to the negative regulation of STAT5b activity by ErbB4 JM-a (Supplementary Fig. 5e). Observations consistent with the analyses of cleavage-

resistant ErbB4 were made when ErbB4 JM-a cleavage was inhibited using chemical small molecule inhibitors of gamma-secretase (GSI IX) or ADAMs (TAPI-0) (Supplementary Fig. 6e–l).

The eJM domain of an RTK like ErbB4 could in theory also modulate signaling by serving as a site for differential glycosylation. To examine the role of potential N- or O-linked glycosylation sites of ErbB4 in STAT5 activation, all asparagine, serine and threonine residues within the JM-a- or JM-b-specific sequences were mutated. Expression of JM-a ΔNST (a JM-a construct with mutations at asparagine, two serine and two threonine residues) (Supplementary Fig. 7a, b) and JM-b ΔS (a JM-b construct with mutations at two serine residues) (Supplementary Fig. 7c, d) in COS-7 cells led to similar levels of STAT5a and

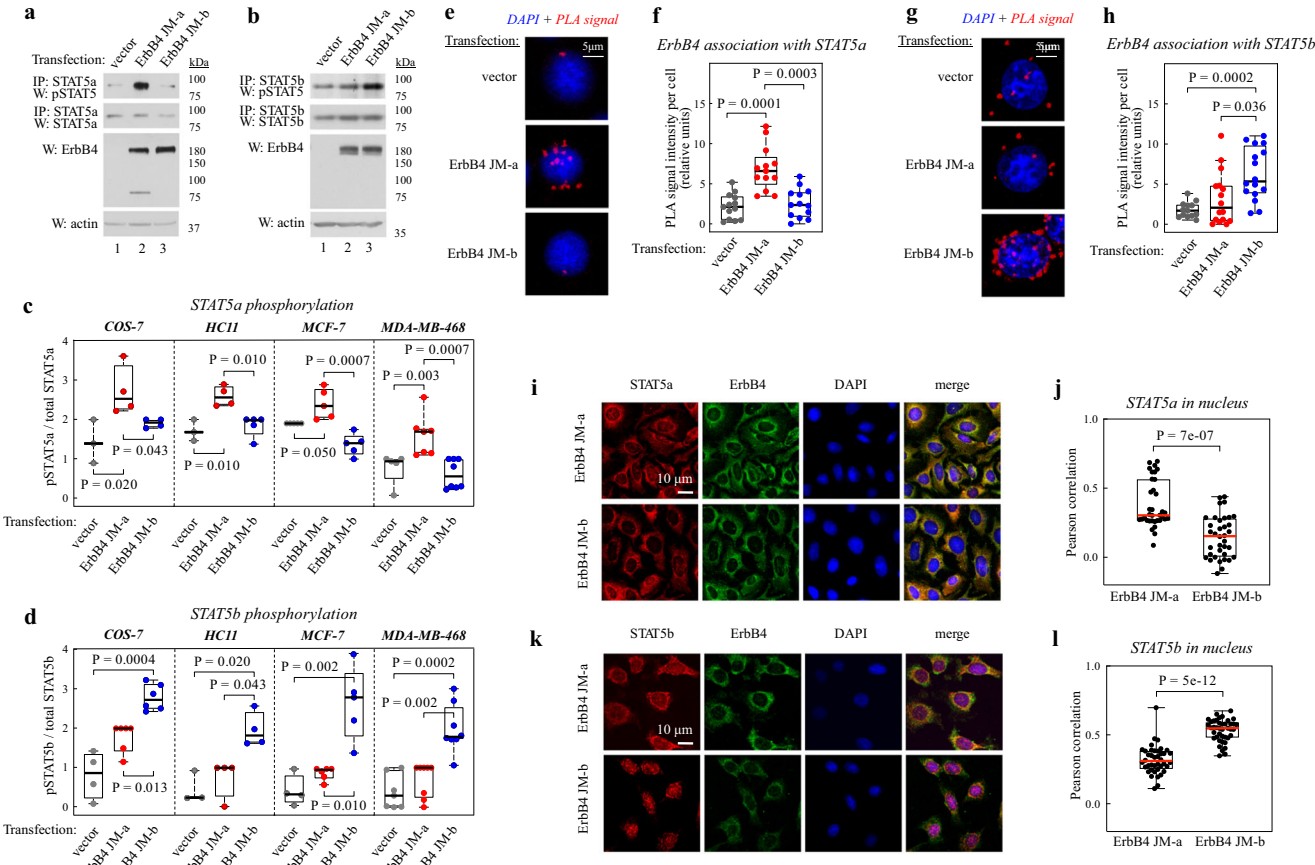

**Fig. 2 | ErbB4 JM-a and JM-b selectively activate different STATs.**
**a**, **b** Phosphorylation status of STAT5a and STAT5b (activating phosphorylation on Y694/699) in MDA-MB-468 cells expressing ErbB4 JM-a or JM-b. Representative blots of n = 7 (STAT5a) and n = 8 (STAT5b) independent experiments. **c**, **d** Densitometric quantification of STAT5 Western analyses, such as shown in panels **a** and **b**, from experiments with the indicated cell lines. COS7: n = 4 (STAT5a) and n = 6 (STAT5b) independent experiments. HC11: n = 4 (STAT5a and STAT5b) independent experiments. MCF-7: n = 5 (STAT5a and STAT5b) independent experiments. MDA-MB-468: n = 7 (STAT5a) and n = 8 (STAT5b) independent experiments. Kruskal Wallis ANOVA. Benjamini, Krieger and Yekutieli adjusted P-values. **e–h** Proximity ligation assays (PLA) of association of ErbB4 JM-a or JM-b with STAT5a or STAT5b in MDA-MB-468 cells. Quantification of the data is shown in panels **f** and **h**. n = 13–16 cells examined over two independent experiments. Brown Forsythe and Welch ANOVA and Dunnet's multicomparison test.
**i–l** Immunofluorescence analysis of nuclear localization of the STAT5 subtypes in HC11 cells expressing ErbB4 JM-a or JM-b after NRG-1 stimulation. Confocal microscopy images (**i**, **k**) and quantification of the co-localization of STAT5 signal and the chromatin stain DAPI (**j**, **l**) are shown. n = 36 (STAT5a) and n = 41 (STAT5b) examined cells over two independent experiments. Two-tailed Mann–Whitney U test. In the boxplots the line represents the median, the box the interquartile range, and the whiskers the whole range of values. Source data are provided as a Source Data file.

STAT5b activation as the expression of their wild-type counterparts. These data demonstrate that the selective STAT5 activation pattern is independent on both the differential sensitivity of the ErbB4 JM isoforms on proteolysis as well as of potential N-linked or O-linked glycosylation within the eJM domain of the isoforms.

### Differences in the structural models of ErbB4 JM isoforms
As differential cleavage or glycosylation did not provide mechanisms for the observed differences in the signaling of the receptor variants, chimeric ErbB4 constructs were designed to map critical sequences. Analyses of constructs in which nine N-terminal isoform-specific residues were swapped into the other isoform (Supplementary Fig. 8a–d), suggested that five residues between positions 626 and 631 that differ between JM-a and JM-b (Fig. 3a) may be critical for the observed functional differences. Visual inspection of the ErbB4 JM-a isoform structure shows that the two disulfide bonds – C617 paired with C625 and C612 with C633, located within a region where five amino acid differences occur between the isoforms – would likely prevent significant alterations in the main-chain conformation over residues 617–633 in both isoforms (Fig. 3a). Some residue differences, e.g. P628

in JM-a versus S628 in JM-b, would in other, less-constrained circumstances likely lead to main-chain conformational changes. In the ErbB4 ectodomain structure, residues 634–650 of JM-a have an extended structure, where aromatic stacking with adjacent side chains and hydrophobic interactions take place between the two monomers. In the JM-b isoform structural model, the sequence $^{634}$IYYPWT$^{639}$ of the JM-a isoform is replaced by $^{634}$IGLMDR$^{639}$; whereas aromatic stacking interactions are not present, the hydrophobic side chains would still be capable of interacting with each other, including interactions between I634 and I626.

Molecular dynamics (MD) simulations suggested that it is unlikely that substantially differing conformations between the two isoforms are involved in selective STAT5 activation. The JM-a and JM-b isoforms exhibit similar conformational stability profiles during the first and last 40 ns of the simulation time (Fig. 3b). Both isoforms also maintain very similar structures and flexibility patterns along the ectodomain and transmembrane sequence (Fig. 3c): residues C617-C633, having five amino acids differences, superposed with near identical average RMSFs for the two isoforms, 0.39 ± 0.06 Å for JM-a and 0.4 ± 0.1 Å for JM-b, and JM-a superposed on JM-b with an RMSD of 0.43 Å, 0.65 Å and 0.68 Å, respectively at 0 ns, 50 ns, and 100 ns, showing the disulfide

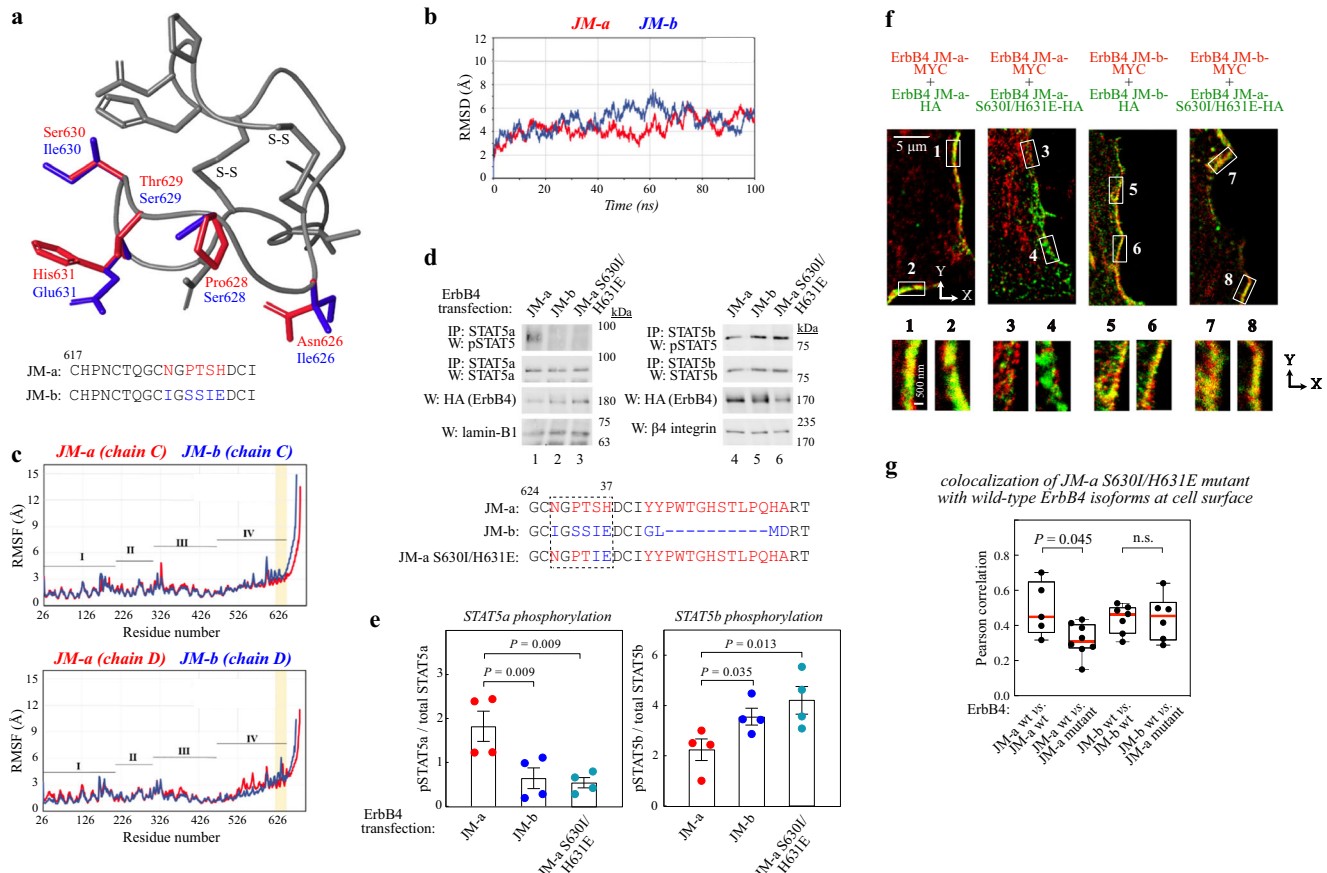

**Fig. 3 | JM-a-specific amino acid residues necessary for selective STAT5 activation. a** Overlaid structures of the JM-a and JM-b isoforms within the eJM residues 617–634. Residues differing between the isoforms are shown in red for JM-a and blue for JM-b. The two disulfide bridges (S-S) are indicated. **b**, **c** Conformational stability, and flexibility of ectodomain-transmembrane domain dimers of ErbB4 JM-a (red) or JM-b (blue). RMSD over backbone atoms (**b**) and RMSF over C atoms during the 100 ns simulations (**c**). In panel **c** the ectodomain sequence location of domains I-IV are indicated and the location of the sequence differences between isoforms JM-a and JM-b are highlighted in yellow. The two chains of the dimer are based on chains C and D of PDB structure 3U7U. **d**, **e** Phosphorylation of STAT5a and STAT5b (on Y694/699) in COS-7 cells expressing ErbB4 JM-a, ErbB4 JM-b, or ErbB4 JM-a S630I/H631E mutant. Cells were stimulated with NRG-1. The sequence alignment of the constructs is shown at the bottom of panel **d**. The dashed box

indicates the isoform-specific sequence shown in panel **a**. Densitometric quantification of *n* = 4 independent experiments (mean ± SEM) is shown in panel **e**. One way ANOVA. Benjamini, Krieger, and Yekutieli adjusted *P*-values. **f**, **g** STED super-resolution immunofluorescence analysis of MCF-7 cells expressing the indicated MYC-tagged (red) and HA-tagged (green) ErbB4 constructs. The white boxes highlight regions of interest in the x-y plane that are magnified below. Panel **g** depicts quantification of co-localization of MYC- and HA-specific signals at the cell surface in x-y plane where each dot represents the correlation of the signals in one cell. *n* = 5–7 cells examined over two independent experiments. Two-tailed Mann–Whitney U test. In the boxplots the line represents the median, the box the interquartile range and whiskers the whole range of values. Source data are provided as a Source Data file.

constrained region to be well-aligned between the isoforms (Supplementary Fig. 8e). The average RMSF for the entire region spanning the amino acid differences was larger for JM-a with a longer, flexible C-terminal loop (0.9 ± 0.2; residues C617-A648) compared to JM-b (0.6 ± 0.1; residues C617-D638).

Given the results from the MD simulations, it seemed much more likely that differences in the amino acid properties and their interactions were involved in the regulation of selective STAT5 activation by the JM-a and JM-b isoforms. The simulations focused our attention on position 631. In JM-a H631 interacts with T629, S630, and D632 of the same monomer, the imidazole ring δ-nitrogen of H631 was also observed to interact with the phosphate oxygen atoms of the POPC head groups in the upper leaflet of the membrane, albeit for only 12% of the time in both chains C and D. In JM-b, H631 is replaced by glutamic acid maintaining an interaction with positon 629 (serine), but negatively charged E631 was observed to form an ionic interaction with the cationic choline moiety of POPC (average electrostatic interaction energy of −14 ± 7.0 kcal/mol). Based on these results, we suspected that the chemical nature of the residue at position 631 might

differentially affect the type of interactions mediated by the JM-a and JM-b isoforms with other molecules such as membrane-associated proteins and lipids.

### Residues S630 and H631 of ErbB4 JM-a are necessary for both selective STAT5a activation and specific membrane localization

The structural models and MD simulations of ErbB4 eJM regions suggested that the properties of the amino acid in position 630 and 631 may account for the isoform-selective ability to activate STAT5a or STAT5b. To experimentally address this, a ErbB4 JM-a S630I/H631E mutant was generated. Residues S630 and H631 in the JM-a sequence *versus* the corresponding residues I630 and E631 in the JM-b sequence are markedly different in terms of hydrophilicity, aromacity, polarity, and charge (Fig. 3a). The ErbB4 JM-a S630I/H631E variant, while functionally competent in promoting ErbB4 autophosphorylation (Supplementary Fig. 8b), was not capable of inducing STAT5a phosphorylation similar to the wild-type JM-a receptor (Fig. 3d, e). However, STAT5b was equally phosphorylated by ErbB4 JM-b and the ErbB4 JM-a S630I/H631E variant (Fig. 3d, e). Accordingly, ErbB4 JM-

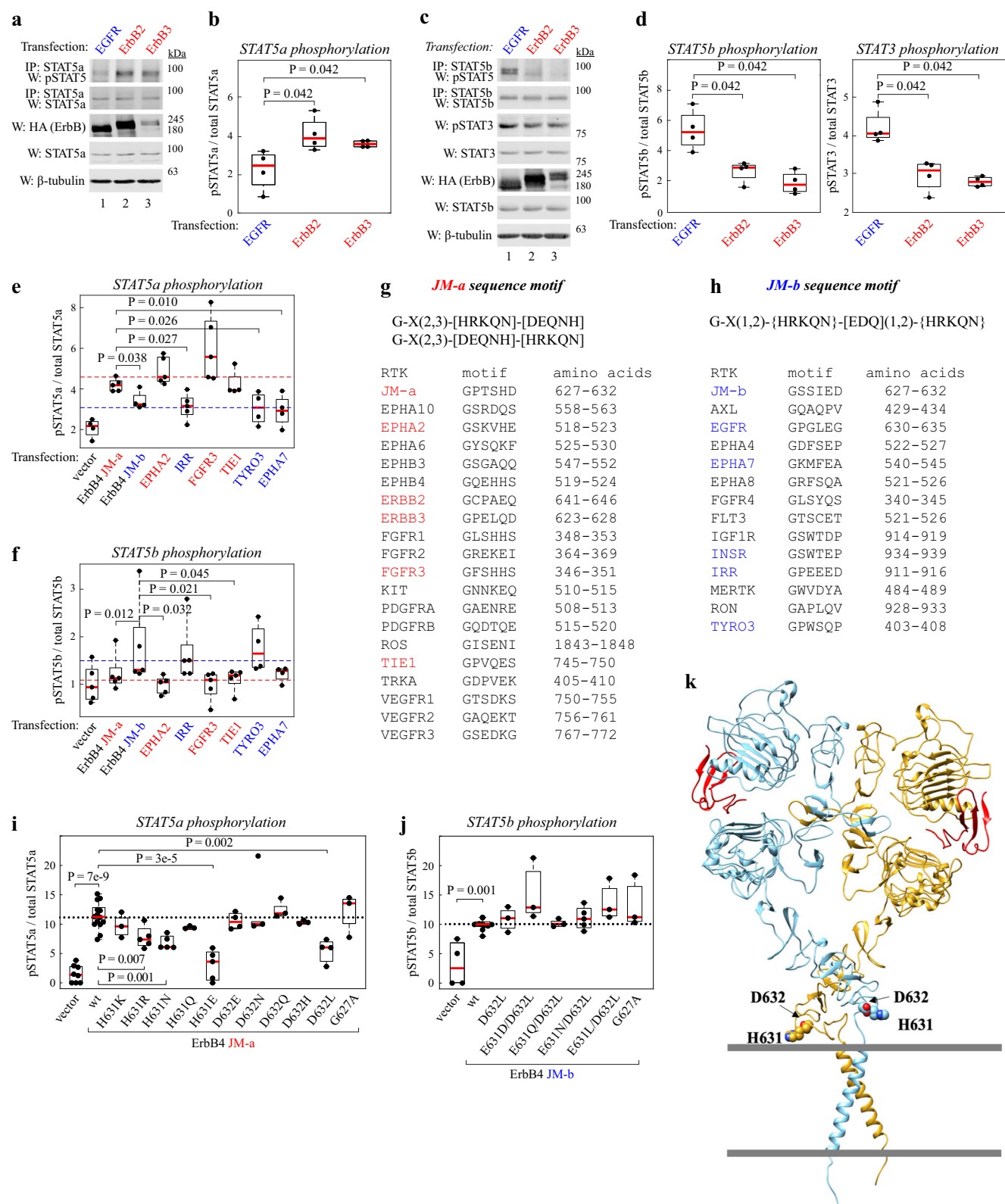

a S630I/H631E demonstrated reduced co-localization with the wild-type ErbB4 JM-a (Fig. 3f, g) and colocalized with wild-type ErbB4 JM-b to the same extent as two differently tagged ErbB4 JM-b wild-type receptors (Fig. 3f, g). These data suggest that the same eJM structures regulating the differential STAT5 activation downstream of ErbB4 isoforms, are also critical for the isoform-specific cell surface localization.

## STAT activation preference is a shared feature of RTKs

To address the STAT activation patterns in other RTKs, other members of the ErbB family were first analyzed. The overexpression of ErbB2 and ErbB3 selectively activated STAT5a (Fig. 4a, b), similar to ErbB4 JM-a, while EGFR activated STAT5b and STAT3 (Fig. 4c, d), similar to ErbB4 JM-b. Consistent with differential subcellular localization associating with the differential STAT signaling, STED microscopy demonstrated

**Fig. 4 | An extracellular RTK sequence motif is associated with selective STAT signaling. a–d** Phosphorylation of STAT5a, STAT5b and STAT3 in COS-7 cells expressing EGFR, ErbB2, or ErbB3. Activating STAT5 and STAT3 phosphorylation was detected with antibodies specific for pY694/699 and pY705, respectively. Cells were treated with NRG-1 or EGF. Panels **b** and **d** depict densitometric quantification of $n = 4$ independent experiments. Mack-Skillings two-way ANOVA. **e, f** Densitometric quantification of data from Western analyses of STAT5a and STAT5b phosphorylation (on Y694/699) in COS-7 cells expressing the indicated RTK constructs. Cells were treated with 1% FCS. Representative Western blots are shown in Supplementary Fig. 10a, b. $n = 5$ independent experiments. Mack-Skillings two-way ANOVA. **g, h** JM-a- and JM-b-like sequence motifs in the eJM domain of RTKs. **i, j** Densitometric quantification of data from Western analyses of STAT5a and STAT5b phosphorylation (on Y694/699) in COS-7 cells expressing the indicated wild-type (wt) or mutant ErbB4 constructs. Cells were treated with NRG-1. Representative Western blots are shown in Supplementary Fig. 10c, d. $n = 3–5$ independent experiments. One way ANOVA. Benjamini, Krieger and Yekutieli adjusted P-values. **k** Position of residues H631 and D632 in the ErbB4 model structure. A compilation model of the ErbB4 ectodomain homodimer X-ray structure with bound NRG-1 (PDB ID: 3U7U) and NMR structure of ErbB4 transmembrane dimer (PDB ID: 2LCX) is shown. The location of the key residues H631 and D632 are indicated in atom fill. The borders of plasma membrane are marked with grey horizontal lines. In the boxplots the line represents the median, the box the interquartile range and whiskers the whole range of values. Source data are provided as a Source Data file.

that ErbB4 JM-a preferentially colocalized with ErbB2 and ErbB3, while ErbB4 JM-b preferentially colocalized with EGFR (Supplementary Fig. 9).

Further analyses of RTKs outside of the ErbB gene family demonstrated that expression of EPHA2, FGFR3 or TIE1 promoted activation of STAT5a similar to ErbB4 JM-a (Fig. 4e; Supplementary Fig. 10a), and expression of IRR and TYRO3 activation of STAT5b similar to ErbB4 JM-b (Fig. 4f; Supplementary Fig. 10b). The overexpression of EPHA7-seemed unable to induce the activation of either STAT5. However, the expression of none of the RTKs was sufficient to activate both STAT5s. These data suggest that selectivity in STAT5 subtype activation pattern is common among RTKs and follows the colocalization of the RTKs with the two ErbB4 JM isoforms at the cell surface.

## Uncovering a conserved sequence motif in the eJM of RTKs

Different sequence motif models were fitted into the eJM sequences (30 residues upstream of the transmembrane domain) of the experimentally validated STAT5a- and STAT5b-activating RTKs (Fig. 4a–f) to find a shared sequence motif around the experimentally discovered essential residues 627-632 (Fig. 3; Supplementary Fig. 8) in the ErbB4 JM sequences. The best fitting model (Fig. 4g, h; Supplementary Data 5) correctly categorized all experimentally validated RTKs ($P < 0.0001$) into JM-a-like STAT5a- or JM-b-like STAT5b-activating RTKs. The model suggested the presence of two sequence motifs marked by the presence (JM-a motif) or absence (JM-b motif) of a residue with a hydrogen bond donor group beside a residue with a hydrogen bond acceptor group three to four residues apart from a glycine residue (Fig. 4g, h). Further statistical analysis discovered that the JM-a motif was significantly enriched in the eJM regions among 53 human RTKs ($P = 0.017$). The JM-b motif, in turn, was not enriched in the eJM domains of human RTKs ($P = 0.97$), indicating that the JM-b motif might in fact correspond to the absence of a JM-a motif.

The interspecies conservation of the JM-a and JM-b motif was assessed in 16 taxonomic categories. Both the JM-a and the JM-b motif displayed statistically significant conservation across species ($84.4 \pm 13\%$ similarity at amino acid level, $P < 0.0001$), indicating the importance of maintaining the motif region to conserve receptor function. Together the statistical analyses suggest a conserved sequence motif, both responsible for selective STAT5 activation, and explicitly present in the eJM region of RTKs.

## Mutational analysis of the RTK eJM motif

The eJM sequence motifs were validated by a point mutation panel to assess which substitutions – present in the motif in a similar positon in other RTKs – in ErbB4 JM-a and ErbB4 JM-b disrupt the downstream activation of STAT5s. Constructs harboring G627A, H631R, H631K, H631Q, H631N, H631E, D632E, D632Q, D632N, D632H, or D632L mutations in JM-a, or G627A, D632L, E631D/D632L, E631Q/D632L, E631N/D632L, or E631L/D632L mutations in JM-b, were expressed in COS-7 cells. The activating phosphorylation of STAT5a (Fig. 4i; Supplementary Fig. 10c) and STAT5b (Fig. 4j; Supplementary Fig. 10d) was measured.

Substituting H631 in JM-a with an arginine, lysine, glutamine, or asparagine retained the activity of the receptor to activate STAT5a to some extent (Fig. 4i). The H631 substitution to glutamic acid in turn resulted in a loss of the ability of ErbB4 to activate STAT5a. However, substituting D632 with a glutamic acid, glutamine, asparagine and histidine did not impair the ability of ErbB4 JM-a to activate STAT5a. ErbB4 JM-a-mediated STAT5a activation was reduced only when the aspartic acid residue was replaced by leucine residue at position 632 (D632L) (Fig. 4i).

Interestingly, none of the tested substitutions in JM-b had an impact on the ability of ErbB4 JM-b to activate STAT5b (Fig. 4j). These data, along with the lack of conservation in the eJM region, support the hypothesis that the JM-b motif is functionally passive and represents the absence of an active JM-a motif. The G627A substitution was ineffective in both ErbB4 JM-a and JM-b contexts, which may be due to the insufficiency of the glycine to alanine substitution to change the amino acid properties sufficiently to alter receptor function. Alternatively, the glycine in the first position of the motif might not be required for a functional sequence motif (Fig. 4i, j). Taken together, these findings indicate that the coupling of a hydrogen bond donor residue (H631 in Fig. 4k) and a hydrogen bond acceptor residue (D632 in Fig. 4k) in the JM-a sequence motif is crucial for the function of the motif.

## STAT5 selectivity of endogenously expressed RTKs

To assess the STAT5 subtype activation pattern by endogenously expressed RTKs, HC11 cells were simulated with NRG-1, FGF-3, or PDGF-AA to activate their cognate receptors harboring a JM-a-like eJM motif (ErbB4 JM-a, both FGFR1 and FGFR2, or PDGFRA, respectively), or with insulin to activate the JM-b-like INSR. Furthermore, EGF was tested as a ligand capable of activating both a JM-a-like (ErbB2) and a JM-b-like (EGFR) RTK in HC11 cells (Fig. 5a), as a result of receptor heterodimerization. As a mammary epithelial cell line, the HC11 cells exclusively express the JM-a isoform of ErbB4[36]. As expected, stimulation with either NRG-1, FGF-3, or PDGFR-AA primarily promoted STAT5a activation, stimulation with insulin primarily STAT5b, and stimulation with EGF both STAT5a and STAT5b (Fig. 5a, b).

To address the functional significance of specific STAT5 subtypes in signaling downstream of the endogenous receptors, STAT5 expression was down-regulated by subtype-specific RNA interference. As a functional read-out, the activity of NRG-1 or insulin in rescuing HC11 cells from starvation-induced death was measured with live cell imaging using IncuCyte. Down-regulation of STAT5a by two independent siRNAs significantly ($P = 0.04$ for siRNA #1; $P = 0.01$ for siRNA #2) reduced the capacity of NRG-1 to promote survival, while it had no significant effect on insulin-promoted survival ($P = 0.91$ for siRNA #1; $P = 0.21$ for siRNA #2) (Fig. 5c). In contrast, downregulation of STAT5b suppressed the survival induced by insulin ($P = 0.03$ for siRNA #1; $P = 0.02$ for STAT5b siRNA #2) with a relatively minor effect for survival induced by NRG-1 ($P = 0.06$ for siRNA #1; $P = 0.20$ for siRNA #2) (Fig. 5c). Taken together these observations indicate that the different

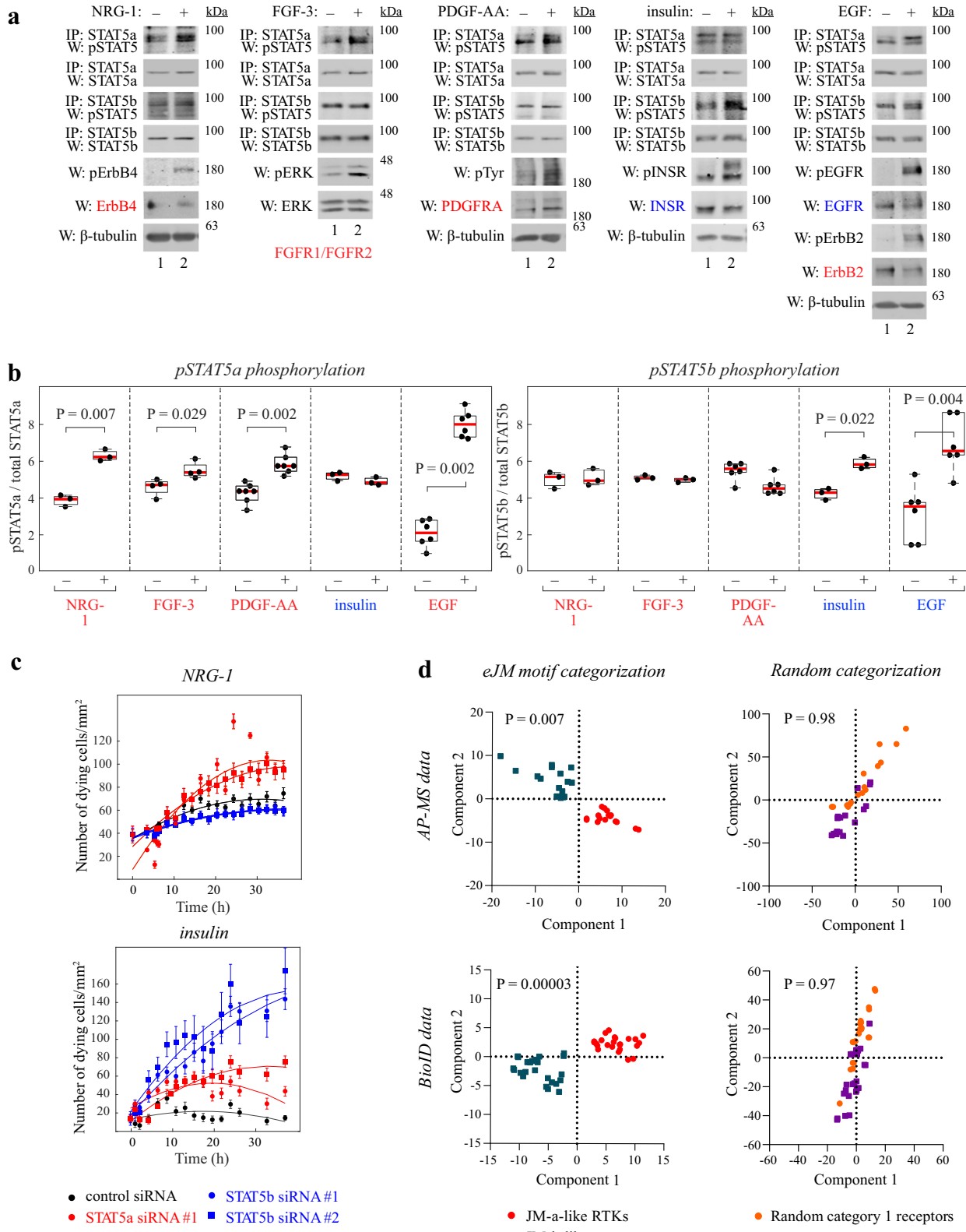

eJM motifs are functionally associated with selective STAT5 signaling in a cellular context with endogenous RTK expression.

**Unique interactomes of the JM-a-like and JM-b-like RTKs**

To examine the clusterization of JM-a-like RTKs (RTKs with a JM-a sequence motif) and JM-b-like RTKs (RTKs with a JM-b sequence motif) on the basis of RTK protein-protein associations, affinity purification

coupled to mass spectrometry (AP-MS) and proximity-dependent biotin identification (BioID) data on RTK interactomes[37] were analyzed. The interactome data were subjected to neighborhood component analysis, where the RTKs where either categorized based on their eJM sequence motifs or by 1000 randomized classes to estimate the statistical significance of the clusterization. As expected, the JM-a-like and JM-b-like RTKs – unlike the random categorizations – significantly

**Fig. 5 | Differential signaling by JM-a- and JM-b-like RTKs. a, b** Phosphorylation of STAT5a and STAT5b (on Y694/699) in HC11 cells after stimulating endogenous ErbB4 JM-a, FGFR1/FGFR2, PDGFRA, INSR or EGFR/ErbB2 with their respective ligands NRG-1, FGF-3, PDGF-AA, insulin or EGF. Panel b depicts densitometric quantification of independent experiments (*n* = 3-6). Two-tailed Mann–Whiney U (EGF, STAT5b) or T-test (rest). **c** Cell death of serum-starved HC11 cells cultured in the presence of the indicated siRNAs and either NRG-1 or insulin. Cell death was measured by analyzing cellular morphology with IncuCyte live cell imaging. Pooled

results from *n* = 17 wells (mean ± SEM) from 3 independent experiments. Two-tailed Mann–Whitney U-test (from the 24 h time point). **d** Clusterization of JM-a-like and JM-b-like RTKs based on their mass spectrometry-derived interactomes. AP-MS, affinity purification coupled to mass spectrometry; BioID, proximity-dependent biotin identification. The *P*-values were estimated from the simulated cumulative distribution function of random categorization. In the boxplots the line represents the median, the box the interquartile range and whiskers the whole range of values. Source data are provided as a Source Data file.

clustered into separate groups based on the proteins they associate with (Fig. 5d).

Next, subsets of the the AP-MS- and BioID-derived RTK interactomes that were specific for the JM-a-like versus JM-b-like RTKs were identified (Supplementary Data 6). Consistently with the STAT3 phosphorylation analyses (Supplementary Fig. 5c, d; Fig. 4c, d), STAT3 was found to associate only with the JM-b-like RTKs in the AP-MS-derived RTK interactomes.

The subcellular localization of the interacting proteins of JM-a-like and JM-b-like RTKs was examined with localization enrichment analysis. Consistent with the super-resolution imaging data (Fig. 1a; Supplementary Fig. 1; Supplementary Fig. 9), the enrichment analysis (Supplementary Data 7) demonstrated a significant lack of colocalization between the JM-a-like versus the JM-b-like RTKs (*P* = 0.002). However, when the subcellular localization of the proteins specifically associating with ErbB4 JM-a or ErbB4 JM-b (Supplementary Fig. 2; Supplementary Data 7) was analyzed, a significant overlap was discovered between the interactome of JM-a-like RTKs versus the interactome of ErbB4 JM-a, as well as between the interactome of JM-b-like RTKs versus the interactome of ErbB4 JM-b (*P* < 0.0001) (Supplementary Fig. 11). These findings support a concept of a unique pattern of localization and molecular interactions associating with the presence of the JM-a versus JM-b sequence motif in eJM domains of RTKs.

### Interaction of the JM-a motif with a complex N-glycan

To further investigate molecular interactions guiding the localization and complex formation of JM-a and JM-b-like RTKs, cyclized peptides corresponding to residues 621–633 of both ErbB4 JM-a and JM-b with a C625S substitution (to avoid artefactual disulfide bridging) were synthesized (Supplementary Fig. 12a). To assess the affinity of the two peptides for different lipids, a membrane lipid overlay assay was performed (Supplementary Fig. 12b). Both the JM-a and the JM-b peptide showed affinity only for phosphatidylinositol at low lipid concentrations, while no binding to diacylglycerol, phosphatidic acid, phosphatidylserine, phosphatidylethanolamine, phosphatidylcholine, phosphatidylglycerol, or sphingomyelin was detected (Supplementary Fig. 12b, c). These data suggest that the eJM motifs in ErbB4 JM-a and JM-b do not differentially interact with cell surface lipids.

To assess the interaction of the eJM motifs with glycans, a mammalian glycan microarray from the Consortium for Functional Glycomics was incubated with two different amounts (5 μg and 50 μg) of the synthetic JM-a and JM-b peptides. The JM-a peptide bound several glycans at a significantly greater affinity than the JM-b peptide, which only seemed to favor a few glycans over the JM-a peptide (Fig. 6a; Supplementary Fig. 13). The glycan with the greatest affinity for the JM-a peptide [GalNAcb1-4GlcNAcb1-2Mana1-6(GalNAcb1-4GlcNAcb1-2Mana1-3)Manb1-4GlcNAcb1-4GlcNAc-Asn] bound the peptide with approximately five-fold greater affinity as compared to the JM-b peptide (with normalized relative fluorescence unit (RFU) levels of 677 and 131, respectively). Structurally, this glycan resembles a complex bi-antennary N-glycan. Of the top 18 glycans that demonstrated the strongest interaction with the JM-a peptide compared to the JM-b peptide, 11 were complex N-glycans (Fig. 6a, b; Supplementary Fig. 13). These findings suggest that the JM-a-specific

residues 626 to 631 interact with a complex N-glycan at the cell surface.

### Regulation of RTK-induced STAT5 activation by lectins

To address the functional role of glycans in selective STAT5 subtype activation by RTKs, COS-7 cells expressing ErbB4 JM-a or JM-b were incubated for 18 h in the presence of lectins with different glycan binding specificities, followed by Western analysis of STAT5 phosphorylation (Fig. 6c, d). ErbB4 JM-a-mediated STAT5a activation was disrupted with the lectins DSL, STL, RCA I and PHA-L, while ErbB4 JM-b-mediated STAT5b activation was unaltered by all the lectins except RCA I (Fig. 6c, d). RCA I also led to significant amount of cell death (~41% reduction of viable cells based on cell morphology) and ErbB4 downregulation (85.8 – 99.6% less signal in Western analyses), indicative of a non-specific toxic effect. DSL, STL, and PHA-L lectins are known to bind complex N-glycans at the cell surface[38]. Previous research suggests that PHA-L specifically binds the 1,6 beta branch in complex N-glycans, while DSL and STL prefer the 1,4 beta branch (Fig. 6b)[38]. N-glycans with a beta 1,6 branch were also significantly enriched in the glycan structures with high affinity for JM-a in the glycan microarray (*P* = 0.023).

In addition to their effects on ErbB4 JM-a, DSL and STL lectins efficiently reduced ErbB2- and ErbB3-mediated STAT5a activation, but not EGFR-mediated STAT5b activation (Supplementary Fig. 14a–c). Together with the observations of differential activation of STAT5 phosphorylation (Fig. 4c, d), and localization at the cell surface (Supplementary Fig. 9), these findings imply that the JM motifs in different RTKs predict downstream STAT5 activation preference by the same glycan-dependent mechanism that controls it downstream of the ErbB4 JM isoforms.

To investigate the role of glycan interactions in the context of endogenously expressed RTKs, the effect of DSL, STL, and PHA-L lectins on cellular survival promoted by endogenous ErbB4 JM-a, PDGFRA, FGFR1/2 (JM-a-like RTKs) or EGFR and INSR (JM-b-like RTKs) was tested in HC11 cells. Treatment with any of the three lectins suppressed NRG-1, PDGFAA and FGF3-promoted survival (Fig. 6e; Supplementary Fig. 14d, e), while none had a significant effect on survival induced by EGF or insulin (Fig. 6f; Supplementary Fig. 14f). These findings indicate that the interaction with N-linked glycans is necessary for STAT5a activation by JM-a-like RTKs, but not for STAT5b activation by receptors with a JM-b-like eJM domain.

### Lectin-sensitive association of ErbB4 JM-a with β1 integrin

To identify cell surface glycoproteins that could potentially selectively interact with the JM-a-type RTKs as they carry N-linked glycans, proteins associating with the PHA-L lectin were analyzed by lectin pull-down followed by mass spectrometry. The analysis was carried out in three different cell backgrounds shown to recapitulate the STAT5 activation selectivity and differential cell surface localization by the two ErbB4 JM isoforms: COS-7, HC11, and MCF-7 cells. Altogether 12 proteins were identified that were pulled down with PHA-L in all the three cell lines (Fig. 6g). Interestingly, β1 integrin was discovered as a glycoprotein that was recognized by the PHA-L lectin in all tested cell lines, consistent with the finding that two β1 integrin pathways were identified to differentially associate with ErbB4 JM-a versus JM-b

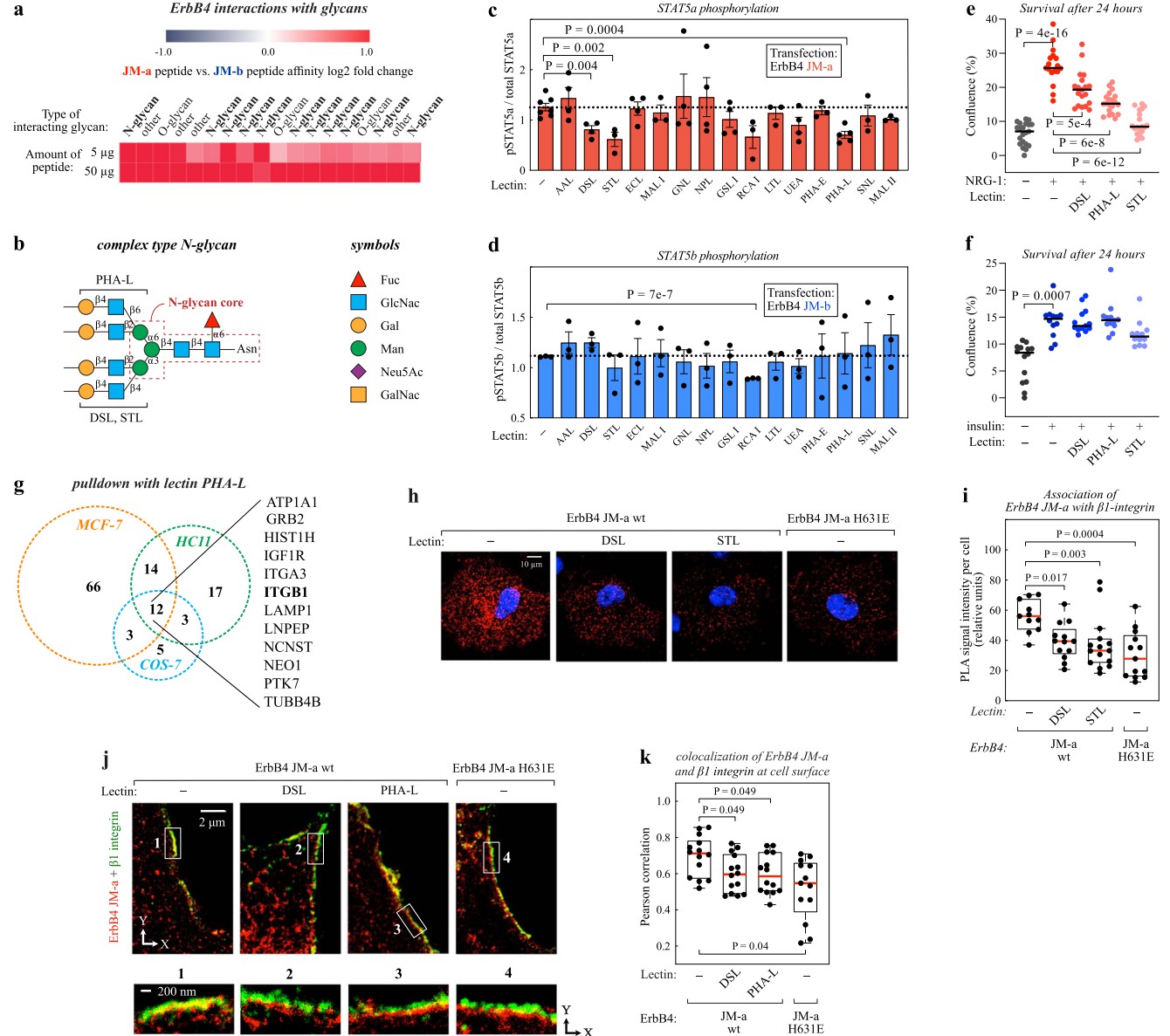

**Fig. 6 | Glycan interaction with ErbB4 JM-a. a** Interactions of peptides derived from JM-a- or JM-b-specific sequences with glycans in a mammalian glycan array from CFG. For details, see Supplementary Fig. 13. **b** An example of a structure of a complex type N-glycan. Core structure and binding sites for indicated lectins are shown. **c**, **d** Densitometric quantifications (mean ± SEM) of data from Western analyses of STAT5a (**c**) and STAT5b (**d**) phosphorylation (on Y694/699) in COS-7 cells expressing ErbB4 JM-a (**c**) or JM-b (**d**). Cells were treated or not with the indicated lectins and stimulated with NRG1. $n = 3–5$ independent experiments. One way ANOVA. **e**, **f** Growth of serum-starved HC11 cells cultured in the presence of the indicated lectin and either NRG-1 (**e**) or insulin (**f**). Cell confluence was measured using live cell imaging with IncuCyte. ***$P < 0.001$ as compared to the treatment with ligand in the absence of lectin. $n = 18–27$ wells examined over three independent experiments. Brown–Forsyte and Welch ANOVA. **g** Mass-spectrometry analysis of proteins interacting with biotinylated PHA-L lectin in the indicated cell lines. **h**, **i** Proximity ligation assay (PLA) of association of wild-type (wt) or H631E mutant

ErbB4 JM-a with endogenous β1 integrin in COS-7 cells treated or not with DSL or STL. Quantification of the data is shown in panel i. $n = 12–15$ cells examined over two independent experiments. Kruskal-Wallis ANOVA. Benjamini, Krieger, and Yekutieli adjusted $P$-values. **j**, **k** STED super-resolution immunofluorescence analysis of colocalization of endogeously expressed β1 integrin (green) and ectopically expressed wild-type or H631E mutant ErbB4 JM-a (red) in COS-7 cells treated or not with the indicated lectins. The boxes highlight regions of interest that are magnified below. Panel **k** depicts the quantification of co-localization of ErbB4- and β1 integrin-specific signals at the cell surface where each dot represents the correlation of the signals in one cell. $n = 14$ cells examined over two independent experiments. One-way ANOVA. Benjamini, Krieger, and Yekutieli adjusted $P$-values. In the boxplots the line represents the median, the box the interquartile range and whiskers the whole range of values. $P$-values were Benjamini, Krieger, and Yekutieli adjusted. Source data are provided as a Source Data file.

(marked in blue in Supplementary Fig. 3). To confirm that the association between ErbB4 and β1 integrin was selective for the JM-a isoform, anti-ErbB4 immunoprecipitates from MDA-MB-468 cells expressing either ErbB4 JM-a or JM-b were analyzed by a β1 integrin-targeted mass spectrometry. Parallel reaction monitoring (PRM) of the fragment ion peaks of a unique β1 integrin peptide indicated that significantly more β1 integrin was precipitated with ErbB4 JM-a as

compared to ErbB4 JM-b in three independent experiments (Supplementary Fig. 15a–c).

SIM imaging of COS-7 cells expressing ErbB4 JM-a or JM-b further demonstrated differential colocalization of β1 integrin with the two JM isoforms at the cell surface (Supplementary Fig. 15d–g). There was significantly more colocalization of endogenously expressed β1 integrin with ErbB4 JM-a as compared to JM-b (Supplementary

Fig. 15d–h). These findings suggest that the preferred precipitation of β1 integrin with the JM-a isoform of ErbB4 is associated with preferred cell surface colocalization with the same isoform.

The coprecipitation of β1 integrin with the PHA-L lectin (Fig. 6g), as well as previous reports[39,40], indicated that the mature β1 integrin protein harbors side chains of complex N-glycans that could potently mediate interactions with the JM-a motif of ErbB4. To experimentally investigate the role of N-glycans in the connection between ErbB4 JM-a and β1 integrin, both physical association and colocalization of the two proteins were studied in the presence and absence of the lectins DSL and PHA-L. Both the association (as measured by PLA) (Fig. 6h, i), as well as colocalization at the cell surface (Fig. 6j, k), were indeed disrupted by the presence of the lectins. Moreover, the H631E variant of ErbB4 JM-a, which lacks both the characteristic STAT5 activation (Fig. 3d, e; Fig. 4i) and localization pattern of the JM-a isoform (Fig. 3f, g), associated with β1 integrin to a significantly lesser extent as compared to wild-type ErbB4 JM-a (Fig. 6h–k). These findings indicate that complex N-linked glycans are involved in mediating the association between the JM-a motif of ErbB4 and the β1 integrin. Taken together, these data suggest that an N-glycan-mediated interaction with proteins, such as β1 integrin, may target ErbB4 JM-a-like RTKs to specific cell surface compartments for selective STAT activation.

## Discussion

The eJM region of RTKs has been shown to include proteolytic cleavage sites necessary for ectodomain shedding[4,5]. In addition, structural differences in the eJM region may affect the selectivity by which different ligands recognize the RTK ectodomain, as for example, in the case of alternatively spliced isoforms of FGFR2[3]. Moreover, interaction with the glycosphingolipid GM3 in the junction of the transmembrane helix and the eJM domain of EGFR has been reported to inhibit EGFR autophosphorylation by retaining the receptor in a monomeric form[6]. Here we propose an additional, previously uncharacterized function for the eJM region of RTKs. We demonstrate that a glycan binding sequence motif within the eJM domain of RTKs dictates cell surface localization and selective intracellular STAT signaling.

The naturally occurring eJM splice variants of ErbB4 were used as a model to address signaling regulated by the RTK eJM domain. The JM-a isoform was found to selectively activate STAT5a, and the JM-b isoform STAT5b and STAT3. While the soluble ICD of the cleavable JM-a has previously been reported to associate with STAT5a in the nucleus[41], our findings indicate that JM-a-mediated STAT5a activation was resistant to inhibition of ErbB4 cleavage, consistent with other reports[17,20]. Moreover, the non-cleavable ErbB4 JM-b also readily activated STAT5b, introducing a previously unpublished downstream effector for ErbB4. Together, these observations suggest that a mechanism independent of differential susceptibility to eJM cleavage determines the selectivity in ErbB4 isoform-promoted STAT activation. This conclusion is also consistent with data from our recent screen of gamma-secretase-sensitive human RTKs[42] that, together with data presented here, indicate no clear association between the sensitivity to cleavage and the STAT activation preference by the RTK.

The mechanism of STAT5 subtype-specific activation by ErbB4 isoforms was shown to involve different intracellular molecular associations. ErbB4 JM-b-mediated STAT5b activation was dependent on the expression of TYK2, whereas JM-a-mediated STAT5a activation was not. Analyses with kinase-dead ErbB4 JM-b and TYK2 constructs further implied that the ErbB4 JM-b/TYK2-mediated STAT5b activation was independent of TYK2 kinase activity. While JAK kinase-mediated STAT activation has been studied downstream of RTKs[43,44], this is, to our knowledge, the first observation of JAK-dependent RTK-mediated STAT activation that is independent of the kinase function of the JAK kinase.

While the activation of STAT5b by ErbB4 JM-b involved TYK2, the activation of STAT5a by JM-a-like RTKs was shown to be regulated by structural characteristics inherent to a JM-a-like motif. Point mutations in the eJM region of ErbB4 JM-b did not alter ErbB4 JM-b-mediated STAT5b activation, suggesting that the JM-b eJM region does not similarly function as an interaction interface. The structural analyses, together with the mutation data, are consistent with a model in which a hydrogen bond donor and a hydrogen bond acceptor group in the side chains of key residues in the eJM region (such as residues H631 and D632 in ErbB4 JM-a) enable selective coupling with extracellular molecules, such as those that ultimately lead to STAT activation. In accordance, STED and SIM super-resolution microscopy as well as mass spectrometry analyses of interaction complexes implied a role for JM-a sequence-directed localization of ErbB4 JM-a to cell surface compartment as a factor contributing to differential STAT5 activation. The cell surface microdomain localization of RTKs has previously been reported to be affected by receptor glycosylation[45], and by ligand binding[46]. Here we add association with N-glycosylated cell-surface proteins as another mechanism by which the cell surface microdomain localization of RTKs can be controlled.

β1 integrin was identified as an N-glycosylated cell surface glycoprotein that preferably associated with ErbB4 JM-a. This observation is consistent with previous reports indicating that both β1 integrin[47,48] as well as ErbB4[18,19] are necessary for STAT5a-dependent mammary gland differentiation in vivo. Whether β1 integrin is the only N-glycosylated cell surface protein that can confer receptor tyrosine kinase-mediated selective STAT5 activation, remains to be elucidated.

Here we describe a RTK eJM motif that controls RTK-mediated selective STAT activation at the cell surface. RTKs, however, are not the only plasma membrane-spanning receptors that can activate STATs[21]. Interestingly, several cytokine receptor subunits and receptors (such as a number of receptors belonging to the interleukin and tumor necrosis factor receptor families) also harbor sequences in their eJM regions that are homologous to the JM-a sequence motif. Thus, it is plausible that the selective STAT subtype activation mediated by other receptors, in addition to RTKs, could also be regulated by an eJM motif. Similarly, homologous eJM motifs may control the plasma membrane microdomain localization of cell surface proteins that are not known to activate STATs.

## Methods

### Cell lines

COS-7 and Phoenix Ampho HEK293 cells were cultured in DMEM (Lonza) supplemented with 10% FCS. HC11 cells retrovirally transduced with pBABE-Puro ErbB4 JM-a CYT-1, pBABE-Puro ErbB4 JM-a CYT-2, pBABE-Puro JM-b CYT-1, pBABE-Puro JM-b CYT-2, or pBABE-Puro empty vector were cultured in RPMI 1640 supplemented with 10% FCS, 1:1000 insulin, 10 ng/ml EGF and 3 μg/ml puromycin. MCF-7 cells were cultured in RPMI 1640 (Lonza) supplemented with 10 nM estrogen and 1:1000 insulin. MDA-MB-468 cells retrovirally transduced with pBABE-Puro ErbB4 JM-a CYT-1, pBABE-Puro ErbB4 JM-a CYT-2, pBABE-Puro ErbB4 JM-a CYT-2-HA, pBABE-Puro ErbB4 JM-a CYT-2-V675A-HA, pBABE-Puro JM-b CYT-1, pBABE-Puro JM-b CYT-2, or pBABE-Puro empty vector were cultured in DMEM supplemented with 10% FCS and 3 μg/ml puromycin. MCF-7 cells tranfected with px330 neo, px330 neo TYK2 KO2, or px330 neo TYK2 KO4 transfectants were cultured in RPMI 1640 (Lonza) supplemented with 10 nM estrogen, 1:1000 insulin, and 200 μg/ml G418. All cell lines were maintained at 37 °C in 5% $CO_2$ in media supplemented with 2 mM glutamine (Lonza) and 1 U penicillin-streptomycin (Lonza). The description of the cell lines and reagents used for cell culture can be found in the Resources Table in Supplementary Data 10.

### Plasmids

The constructs pBABE-puro, pBABE-puro ErbB4 JM-a CYT-1, pBABE-puro ErbB4 JM-a CYT-2, pBABE-puro ErbB4 JM-b CYT-1, pBABE-puro ErbB4 JM-b CYT-2, pBABE-puro ErbB4 JM-b CYT-2-HA, pcDNA 3.1

Hyg(+), pcDNA 3.1 ErbB4 JM-a CYT-2 −eGFP Hyg(+), pcDNA 3.1 ErbB4 JM-a CYT-2 −HA Hyg(+), pcDNA 3.1 ErbB4 JM-a CYT-2 −MYC Hyg(+), pcDNA 3.1 ErbB4 JM-b CYT-2 −HA Hyg(+), pcDNA 3.1 ErbB4 JM-b CYT-2 −MYC Hyg(+), pLX302-EphA2-V5, pLX302-EphA7-V5, pLX302-FGFR3-V5, pLX302-INSRR-V5, pLX302-Tie1-V5, pLX302-TYRO3-V5, pME18S-STAT5a, pME18S-STAT5b, pRc/CMV STAT5b Y699F, pRc/CMV STAT5b Y699F/Y724F and pRc/CMV STAT5b Y699F/Y739F have been described elsewhere[11,42,49–51]. Constructs pDest-eGFP-N1, pDONR223-JAK1, pDONR223-JAK2, pDONR223-JAK3, pDONR223-TYK2, and px330 were purchased from Addgene. Constructs pDest-JAK1-eGFP-N1, pDest-JAK2-eGFP-N1, pDest-JAK3-eGFP-N1, and pDest-TYK2-eGFP-N1 were generated with gateway cloning using Gateway LR Clonase II Enzyme mix (Thermo Fisher Scientific) from the corresponding pDONR223 vectors and the pDest-eGFP-N1 backbone. Constructs pcDNA3.1 ErbB4 JM-a ΔNST CYT-2 −HA Hyg(+), pcDNA3.1 ErbB4 JM-a G627A CYT-2 −HA, pcDNA3.1 ErbB4 JM-a H631K CYT-2 −HA, pcDNA3.1 ErbB4 JM-a H631R CYT-2 −HA, pcDNA3.1 ErbB4 JM-a H631Q CYT-2 −HA, pcDNA3.1 ErbB4 JM-a D632H CYT-2 −HA, pcDNA3.1 ErbB4 JM-a D632L CYT-2 −HA, pcDNA3.1 ErbB4 JM-a D632N CYT-2 −HA, pcDNA 3.1 ErbB4 JM-a D632Q CYT-2 −HA, pcDNA3.1 ErbB4 JM a/b chimeric CYT-2 −HA Hyg(+), pcDNA 3.1 ErbB4 JM b/a chimeric CYT-2 −HA Hyg(+),pcDNA3.1 ErbB4 JM-b G627A CYT-2 −HA, pcDNA 3.1 ErbB4 JM-b D632L CYT-2 −HA, pcDNA3.1 ErbB4 JM-b E631D/D632L CYT-2 −HA, pcDNA3.1 ErbB4 JM-b E631L/D632L CYT-2 −HA and pcDNA3.1 ErbB4 JM-b E631Q/D632L CYT-2 −HA were created with Gibson assembly using synthetic gBlocks (IDT) or GENEFragments (Gene Universal). pcDNA 3.1 ErbB4 JM-a CYT-2 −HA Hyg(+) or pcDNA3.1 ErbB4 JM-b CYT-2 −HA Hyg(+) was digested with XcmI and FastAP or with KpnI and Bsp1407I and FastAP, and the fragments were run on an agarose gel. The backbone was extracted from the gel and assembled with the corresponding synthetic gBlocks or GENEFragments with NEBuilder HiFi DNA Assembly Master Mix according to manufacturer's protocol.

Constructs pDest-TYK2 K930I-eGFP-N1, pcDNA3.1 ErbB4 JM-b ΔS CYT-2 −HA Hyg(+), pcDNA3.1 ErbB4 JM-b K741R CYT-2 −HA Hyg(+), pcDNA3.1 ErbB4 JM-b Y974F CYT-2 −HA Hyg(+), and pcDNA3.1 ErbB4 JM-b Y1012F CYT-2 −HA Hyg(+) were constructed using PCR with corresponding site-specific primers and backbone pDest-TYK2-eGFP-N1 or pcDNA3.1 ErbB4 JM-b CYT-2 −HA Hyg(+) with KAPA HiFi PCR kit (Kapa Biosystems). Constructs pcDNA3.1 ErbB4 JM-a Y984F CYT-2 −HA Hyg(+), pcDNA3.1 ErbB4 JM-a Y1022F CYT-2 −HA Hyg(+), pcDNA3.1 ErbB4 JM-a H631N CYT-2 −HA, pcDNA3.1 ErbB4 JM-a H631E CYT-2 −HA and pcDNA3.1 ErbB4 JM-b E631N/D632L CYT-2 −HA were constructed using PCR with corresponding site-specific primers and pcDNA3.1 ErbB4 JM-a CYT-2 −HA Hyg(+) or pcDNA3.1 ErbB4 JM-b CYT-2 −HA Hyg(+) backbone with Phusion High-Fidelity DNA Polymerase. The px330 neo backbone was constructed by extracting the neomycin resistance cassette from pDEST-eGFP-N1 plasmid with PCR reaction using KAPA HiFi PCR kit according to manufacturer's protocol with site-specific primers and ligating it with T4 DNA ligase into EcoRV digested pX330-U6-Chimeric_BB-CBh-hSpCas9 backbone according to manufacturer's protocol. Constructs px330 Neo TYK2 KO2 CRISPR and px330 Neo TYK2 KO4 CRISPR plasmids were constructed by digesting the px330 neo backbone with BbsI and the fragments were run on an agarose gel. The bands with correct size were extracted from the gel. The extracted digested backbone was ligated with corresponding synthetic oligonucleotides with T4 DNA ligase. All inserts were sequenced after cloning. The details of the plasmids and oligos and reagents used can be found in the resources table at Supplementary Data 10.

### Transfection and viral transduction
Different transfection strategies were opted to control expression levels in different cell types. Fugene 6 (Promega) or Hilymax (Dojindo) was used for DNA plasmid transfection in MCF-7 and COS-7 cells, siLentFect (Bio-Rad) for siRNA transfection in MCF-7 cells, and Lipofectamine 2000 (Thermo Fisher Scientific) for siRNA transfection in MDA-MB-468 and HC11 cells, as recommended by the manufacturers. Retroviral constructs pBABE-Puro ErbB4 JM-a CYT-1, pBABE-Puro ErbB4 JM-a CYT-2, pBABE-Puro ErbB4 JM-a CYT-2-HA, pBABE-Puro ErbB4 JM-a CYT-2-V675A-HA, pBABE-Puro JM-b CYT-1, pBABE-Puro JM-b CYT-2, or pBABE-Puro (empty vector) were generated and transfected to Phoenix Ampho HEK293 cells. The supernatant of the transfected packaging Phoenix Ampho HEK293 cells was collected 24 and 48 h after transfection, filtered and applied to the target cell line for transduction with 3 μg/ml polybrene. Puromycin selection was used at 3 μg/ml concentration. The details of the transfection and transduction reagents used can be found in the Resources Table in Supplementary Data 10.

### Immunoprecipitation and western analysis
Cells were starved overnight in serum-free medium for all Western and immunoprecipitation analyses. Cells were lysed with a buffer containing 0.1% Triton X-100, 1 mM EDTA, 5 mM NaF, 10 mM Tris-HCl, pH 7.4, and a protease inhibitor cocktail (Thermo Fisher). For anti-STAT5a, anti-STAT5b, anti-GFP, and anti-ErbB4 immunoprecipitation experiments, 0.5-2 mg of cell lysate was precleared with G-sepharose for 1 h at 4 °C, and then incubated with 1–3 μg of the antibody overnight at 4 °C. The lysate, including the antibodies was incubated with G-sepharose for 1 h at 4 °C and then washed 3–6 times with 0.2% Tween 20 in PBS, before elution with denaturing SDS sample buffer for 5 min at 100 °C and separation of the sample on SDS-PAGE gel. Signals on Western blots were detected with HRP-conjugated secondary antibodies and x-ray film or CCD camera, or with IR-conjugated secondary antibodies and Odyssey CL-x Imaging system (LI-COR). The details of the antibodies and reagents used for immunoprecipitation and western blotting can be found in the Resources Table in Supplementary Data 10. In immunoprecipitation experiments, 1–3 μg of primary antibody per 1 mg of lysed protein was used. In western analyses primary antibodies were used at 1:500 to 1:2000 dilutions and the secondary antibodies at 1:10000 dilutions.

To quantify Western data, Image Studio Lite software with median background correction setting was used. The densitometry data were either normalized to median density within the experiment or to the density sum of common samples in the experiment in the case of only two conditions. Different normalization procedures were selected for visualization purposes and did not alter the quantification results. Experiments in which a background signal was measured, were additionally background-corrected by subtracting the lowest value of all experiments from all values.

### Chemical cross-linking
For mass-spectrometry studies and anti-GFP coimmunoprecipitation analyses (Fig. 1c, d, Supplementary Figs. 2 and 15a–c), the cells were cross-linked with 1–3 mM DSP or DTPB for 2 min and quenched with 50 mM Tris-HCl pH 7.4 for 15 min at room temperature before cell lysis. For mass-spectrometry studies, 160 μl of magnetic G-protein beads were blocked with 5% BSA, incubated with 8 μg anti-ErbB4 (HFR-1) and 8 μg anti-ErbB4 (E200) for 3 h at 4 °C and cross-linked with 3 mM BS$_3$ crosslinker for 30 min at room temperature. The mixture was quenched with 50 mM Tris pH 7.5 for 15 min and washed 3 times with 0.1% Tween 20 in PBS. The cross-link reagents used can be found in the Resources Table in Supplementary Data 10.

### Proximity ligation assay (PLA)
Samples were fixed and permeabilized with methanol for 10–15 min at −20 °C. Proximity ligation assays were conducted according to the manufacturer's (Navinci) protocol with 1:100 primary antibody dilutions. The primary antibodies were incubated 18 h at +4 °C. The coverslips were mounted with mowiol. The antibodies and reagents used for PLA can be found in the Resources Table in Supplementary Data 10.

## Immunofluorescence and microscopy

For immunofluorescence analysis, samples were fixed and permeabilized with methanol for 10-15 min at −20 °C and stained with the indicated primary antibodies (1:50 to 1:200 dilutions). To remove nonspecific binding, 3% BSA in PBS was used as a blocking agent. Primary antibodies were detected with Alexa-conjugated secondary antibodies (1:500 dilutions) and nuclei were visualized with DAPI. Mowiol was used for mounting.

Mounted samples were imaged with Zeiss LSM 510, Zeiss LSM 780 or Zeiss LSM 880 confocal microscopes in Turku Bioscience Centre Imaging Core. For super-resolution 2D STED (stimulated emission depletion) microscopy, Alexa 488-, Abberior STAR 580-, and Abberior STAR 635-conjugated secondary antibodies at 1:100 dilutions were used for detection and Abberior STED microscope for imaging. For SIM (structured illumination microscopy) super-resolution imaging, Alexa 488 and Alexa 555 secondary antibodies were used for detection and Deltavision OMX microscope for imaging. For nuclear localization and 2D STED plasma membrane colocalization analyses, 0.5 μm sections were acquired from the middle of the nuclei (40x Zeiss C-Apochromat objective, Numerical Aperture: 1.2), or from the edges of the cells (100x Olympus UPLSAPO objective, Numerical Aperture: 1.42). The STED experiments, were carried out using 580 nm pulsed and 775 nm continuous depletion lasers with Alexa-488 and Abberior-STAR-635 probes, respectively, or only the 775 nm depletion laser with the combination of Abberior-STAR-595/Alexa-555 and Abberior-STAR-635 probes. For PLA analyses 1 μm (ErbB4 and STAT5 association) or 0.5 μm (ErbB4 and β1 integrin association) sections were acquired from the plane with most PLA signal (40x Zeiss LD LCI Plan-Apochromat, Numerical Aperture: 1.2). For SIM cell surface colocalization analyses, a stack of images were acquired (60x SIM Olympus Plan Apo N objective, Numerical Aperture: 1.42) of which the plane closes to the coverslip were used to analyze the bottom and peripheral cell surface location. The details of the antibodies and reagents used for immunofluorescence analysis can be found in the Resources Table in Supplementary Data 10.

## Sample preparation for mass spectrometry

**Affinity enrichment samples.** Cells were starved overnight in serum-free medium, cross-linked as described above, and lysed with buffer containing 0.1% Triton X-100, 1 mM EDTA, 5 mM NaF, 10 mM Tris-HCl, pH 7.4, and a protease inhibitor cocktail (ThermoFisher Scientific). Thirty μl of protein G magnetic beads was incubated with 32 μg of normal mouse IgG and 32 μg of normal rabbit IgG for 1 hour at 4 °C, and washed 3 times with 0.1% Tween 20 in PBS. Five mg of protein lysate was precleared with 10 μl of G-protein magnetic bead and normal IgG mixture for 1 h 4 °C. Fifty μl of a mixture of G-protein magnetic beads and crosslinked anti-ErbB4 antibody was added to the precleared protein lysate and incubated overnight at 4 °C. The beads were washed 7 times with 0.1% Tween 20 in PBS and eluted by heating in 2x Laemmli sample buffer for 5 min at 100 °C. The samples eluted in Laemmli buffer were analyzed with SDS-PAGE and Coomassie or PageBlue staining. The samples analyzed with PageBlue were fixed overnight in 25% isopropanol and 10% acetic acid mixture prior to staining. Protein bands were excised from the stained SDS-PAGE gel, destained, dried, alkylated, and trypsin-digested for 18 h at 37 °C. The peptides were extracted and dried using speed vac. The details of the reagents used for sample preparation can be found in the Resources Table in Supplementary Data 10.

**PRM samples.** The samples used for targeted PRM mass spectrometry were similarly processed as above until eluted with 6 M guanidine hydrochloride, 5 mM tris(2-carboxyethyl)phosphine, 10 mM chloroacetamide and 100 mM Tris, pH 8.5 in 95 °C for 10 min. Guanidine hydrochloride was diluted to 2 M and eluted samples were digested with 1:50 (w/v) trypsin (Thermo Scientific) overnight at 37 °C.

Following digestion, the reactions were quenched with 10% tri-fluoroacetic acid at a final concentration of approximately 0.5% (pH -2), desalted on tC18 SepPak 96-well plate (Waters), and dried by vacuum centrifugation (Heto Lab). The details of the reagents used for sample preparation can be found in the Resources Table in Supplementary Data 10.

**PHA-L lectin pull-down samples.** For PHA-L lectin pulldown, HC11, MCF-7, and COS-7 cells were lysed and precleared with 50 μl of streptavidin-sepharose. Streptavidin-sepharose (150 μl) was incubated with 300 μg of biotinylated PHA-L lectin for 2 h at room temperature. The mixture was washed 3 times with 0.2% Tween 20 in PBS. The precleared lysates were incubated with the mixture of PHA-L and streptavidin-sepharose or streptavidin-sepharose alone for 24 h in 4 °C. The beads were washed 7 times with 0.2% Tween 20 in PBS, 2 times with PBS, and 1 time with Milli-Q water. Bound proteins were eluted with 6 M guanidine hydrochloride, 5 mM tris(2-carboxyethyl) phosphine, 10 mM chloroacetamide and 100 mM Tris, pH 8.5 in 95 °C for 10 min. Guanidine hydrochloride was diluted to 2 M and eluted samples were digested with 1:50 (w/v) trypsin (Thermo Scientific) overnight at 37 °C. Following digestion, the reactions were quenched with 10% trifluoroacetic acid at a final concentration of approximately 0.5% (pH -2), desalted on tC18 SepPak 96-well plate (Waters), and dried by vacuum centrifugation (Heto Lab). The details of the reagents used for sample preparation can be found in the Resources Table in Supplementary Data 10.

## Mass spectrometry

The mass spectrometry analyses were performed in the Turku Proteomics Facility at Turku Bioscience Centre (University of Turku and Åbo Akademi University). The Facility is supported by Biocenter Finland. All of the dried peptides samples were resuspended in 0.1% formic acid before analysis with mass spectrometers.

For affinity enrichment samples, the peptides of the first experiment were analyzed with a nanoflow HPLC system (Easy-nLCII; Thermo Fisher Scientific) coupled to the LTQ Orbitrap Velos Pro mass spectrometer (Thermo Fisher Scientific) equipped with a nano-electrospray ionization source. Mass spectrometry data were acquired automatically using Thermo Xcalibur software (Thermo Fisher Scientific).

The peptides of the second, third and fourth experiment were analyzed on an Easy-nLC 1200 liquid chromatography system coupled to an Orbitrap Q-Exactive HF instrument (Thermo Fisher Scientific) equipped with a nanoelectrospray source. Peptides were loaded on an in-house packed 100 μm × 2 cm precolumn packed with ReproSil-Pur 5 μm 200 Å C18-AQ beads (Dr. Maisch) using 0.1% formic acid in water (buffer A) and separated by reverse phase chromatography on a 75 μm × 15 cm analytical column packed with ReproSil-Pur 5 μm 200 Å C18-AQ beads. All separations were performed using a 40 minute gradient at a flow rate of 300 nl/minute. The gradient ranged from 6% buffer B (80% acetonitrile in 0.1% formic acid) to 36% buffer B in 30 min. Buffer B concentration was ramped up to 100% in 5 min, and 100% buffer B was run 5 min for washout. The DDA method consisted of a full MS scan (120,000 resolution, 3E6 automatic gain control (AGC) target, 100 ms maximum injection time, 300 to 1750 m/z, profile mode), followed by data-dependent MS/MS acquisitions on the top 20 most intense precursor ions (15,000 resolution, 5E4 AGC target, 100 ms maximum injection time, 2.0 m/z isolation window, normalized collision energy = 27, centroid mode).

The targeted PRM samples were analyzed on an Easy-nLC 1200 liquid chromatography system coupled to an Orbitrap Lumos Fusion instrument (Thermo Fisher Scientific) equipped with a nanoelectrospray source. Peptides were loaded on an in-house packed 100 μm × 2 cm precolumn packed with ReproSil-Pur 5 μm 200 Å C18-

AQ beads using 0.1% formic acid in water (buffer A) and separated by reverse phase chromatography on a 75 μm × 15 cm analytical column packed with ReproSil-Pur 5 μm 200 Å C18-AQ beads. All separations were performed using a 40 minute gradient at a flow rate of 300 nl/minute. The gradient ranged from 6% buffer B (80% acetonitrile in 0.1% formic acid) to 36% buffer B in 30 min. Buffer B concentration was ramped up to 100% in 5 min, and 100% buffer B was run 5 min for washout. A scheduled PRM method was used to simultaneously target all selected β1 integrin peptides. The β1 integrin peptides were selected using an online PRM method designer Picky[52] with default parameters, except miscleaved peptides were allowed, and the maximal number of features monitored in parallel was set to 31. The PRM method consisted of a targeted MS/MS spectra acquisition with a resolution of 30,000, automatic gain control target of 5e[4], isolation window of $m/z$ 1.6, maximum injection time set at 54 ms, and normalized collision energy = 27.

The PHA-lectin pull-downs samples were analyzed on an Easy-nLC 1000 coupled to a Q-Exactive HF instrument (Thermo Fisher Scientific) equipped with a nanoelectrospray source. Peptides were loaded on in-house packed 100 μm × 2 cm precolumn packed with ReproSil-Pur 5 μm 200 Å C18-AQ beads using 0.1% formic acid in water (buffer A) and separated by reverse phase chromatography on a 75 μm × 15 cm analytical column packed with ReproSil-Pur 5 μm 200 Å C18-AQ beads. All separations were performed using a 30 minute gradient at a flow rate of 300 nl/min. The gradient ranged from 12% buffer B (80% acetonitrile in 0.1% formic acid) to 45% buffer B in 25 min. Buffer B concentration was ramped up to 100% in 2 min and 100% buffer B was run 3 min for washout. The DDA method consisted of a full MS scan (120,000 resolution, 3E6 automatic gain control (AGC) target, 100 ms maximum injection time, 300 to 1750 m/z, profile mode), followed by data-dependent MS/MS acquisitions on the top 20 most intense precursor ions (15,000 resolution, 5E4 AGC target, 200 ms maximum injection time, 2.0 m/z isolation window, normalized collision energy = 27, centroid mode).

## Mass spectrometry data analysis

Mass spectrometry raw data files were searched with Metamorpheus[53] (version 0.0.303 for lectin pull-down samples or version 0.0.312 for affinity enrichment samples) against human, murine or green monkey proteome according to the source organism of the analyzed sample. Known post-translational modifications were included in the proteomes downloaded from Uniprot[54] (accessed January 10, 2021). Built-in calibration, post-translational discovery and search tasks of Metamorpheus were utilized for calibration, post-translational modification search, and peptide and protein identification. Peptides and proteins were quantified with FlashLFQ[55] (version 1.1.2). Trypsin was set as the protease and two missed cleavages were allowed. Minimum peptide length was set to seven amino acids. Precursor mass tolerance was set to 5 ppm and fragment ion tolerance was set to 20 ppm. Cysteine carbamidomethylation (57.021463) was set as a constant modification and methionine oxidation was set as a variable modification. All other possible modifications were set using the Post-translational discovery search in Metamorpheus. Search results were filtered to a 1% FDR at PSM, peptide and protein levels. Peptides were accepted with search engine score above 5 and at least one unique peptide was required for protein identification.

Acquired targeted PRM proteomics raw data were analyzed with Skyline[56], version 20.1.1.83, to identify peptides and quantify peak intensities. The ratio of peak areas was used to compare the relative abundance of peptides. The human protein database (Uniprot, accessed September 9, 2019) was used as the background proteome. Spectra for selected peptides were predicted using Prosit[57]. The details of the software used can be found in the Resources Table in Supplementary Data 10.

## Ligand stimulation, chemical inhibition, lectin incubation and siRNA treatment

For Western analyses, cells were stimulated with 50 ng/ml NRG-1, FGF-3, PDGF-AA, EGF or 100 ng/ml insulin for 1-3 min where indicated. For Western analyses in Fig. 4a–d, cells were stimulated for 2 min with 25 ng/ml NRG-1 and 25 ng/ml EGF. For Western analyses in Fig. 4e, f and Supplementary Fig. 10a, b, cells were stimulated with 1% FCS for 2 min. For immunofluorescence analyzes, cells were stimulated with 50 ng/ml NRG-1 for 15 min where indicated. For live cell imaging with IncuCyte, cells were stimulated or not with 100 ng/ml NRG-1, FGF-3, PDGF-AA, EGF or insulin. For proteinase inhibition, cells were treated with 20 μM TAPI-0 or 10 μM GSI IX for 4-5 h. TYK2, JAK2, and control siRNAs were used at 25 nM concentration and incubated with the cells for 48 h. STAT5a, STAT5b, and control siRNAs were used at 50 nM concentration and incubated with the cells for 24 h before the cells were plated for live cell imaging. To block glycan binding for Western, proximity ligation and immunofluorescence analyses, cells were incubated in the presence of 20 μg/ml of lectins for 18 h. For live cell imaging with IncuCyte, cells were pretreated or not with 20 μg/ml of lectins for 18 h, prior to imaging in the presence or absence of freshly added lectins (20 μg/ml). The details of the chemicals and siRNAs used can be found in the Resources Table in Supplementary Data 10.

## Lipid overlay assay and mammalian glycan array

Cyclic synthetic peptides corresponding to JM-a and JM-b eJM region were purchased from JPT. The peptides were desalted with tips with Empore Extraction disc C18 membrane (3 M) according to manufacturer's protocol and eluted into 0.1% formic acid. A dilution of 5 μg/ml of peptide in 3% BSA in PBST was incubated Membrane lipid assay membranes (Echelon Bioscience) for 2 h following manufacturer's protocol. The peptides were detected with 1:2500 dilution of IR-800 streptavidin in 3% BSA in PBST.

The mammalian glycan array analysis was ordered from Consortium for Functional Glycomics' (CFG) Protein-Glycan Interaction Core (Boston, USA) and conducted with 5 μg and 50 μg of the synthetic peptides reconstituted in TSM [150 mM NaCl and 1 mM CaCl₂ in 20 mM Tris-HCl (pH 7.4)] buffer.

## Live cell imaging with IncuCyte

For RNA interference experiments, HC11 cells were plated at a concentration of 32,000 cells/ml into 96-well plates. After 24 h, the cells were starved in 0% FCS for 18 h. RTK ligands were added to the media and the cells tranferred to IncuCyte ZOOM incubator (Essen BioScience). For experiments testing the effects of lectins, HC11 cells were plated at 10% confluency into 96-well plates. The cells were pre-incubated in the presence or absence of lectins for 24 h in medium with 1% FCS. Before transferring the cells into IncuCyte ZOOM, fresh serum-free medium supplemented with RTK ligands and lectins or not was exchanged to the cells. IncuCyte ZOOM Software was used to determine the number of dying cells and confluency from phase contrast images. The acquired time series were normalized to have the same median value at initial time point and fitted with quadratic regression curves. The presented curves indicate the median values and the whiskers the standard errors of mean.

## Structural and model data of ErbB4 isoforms

The consequences of amino-acid differences between the JM-a and JM-b isoforms of human ErbB4 at the C-terminal region of the extracellular domain were examined. The 3.03 Å resolution X-ray structure of the ErbB4 ectodomain homodimer with bound NRG-1β growth factor (PDB ID: 3U7U[58] chains C and D) was obtained from the Protein Data Bank (PDB)[59]. Missing loops in the ectodomain structure were built with the Modeler program[60]. Model 1 of the NMR structure of the human ErbB4 transmembrane domain (PDB ID: 2LCX[61]) was then connected to the

ectodomain structure, resulting in the transmembrane domain-bound ErbB4 JM-a isoform. The JM-a composite structure was subsequently energy minimized with Chimera[62]. A conservative model of the JM-b isoform was built by side-chain replacement of the JM-a structure using Chimera[62]. The protein preparation wizard in Maestro[63] was used to add hydrogen atoms, determine protonation states of ionizable groups at pH 7.0, and energy minimize the two isoform structures. Preliminary visualization and analysis of the isoforms was made with Chimera[62] and Bodil[64]. The details of the software used for the structural modeling can be found in the Resources Table in Supplementary Data 10.

### Molecular dynamics simulation

The ErbB4 JM-a/JM-b−membrane systems were built using the CHARMM-GUI web server[65]. The membrane bilayer was composed of 100% POPC (1-palmitoyl-2-oleoyl-sn-glycero-3-phosphocholine) on the upper leaflet and 70% POPC with 30% POPS (1-palmitoyl-2-oleoyl-sn-glycero-3-phospho-l-serine) on the lower leaflet. The systems were solvated in a rectangular box (150 Å × 150 Å × 235 Å) with TIP3P water molecules[66] and neutralized by adding Na+ counterions. To mimic physiological conditions additional Na + /Cl- ions were added bringing the salt concentration to 150 mM. The simulations were performed with the Amber program[67] using the ff14SB[68] (protein) and LIPID17 (lipid) force fields.

The generated systems were initially energy minimized for 5000 cycles with the steepest descent and conjugate gradient methods setting a 10 and 2 kcal mol-1Å−2 restraint on protein and lipid atoms, respectively. The systems were then heated to 303.15 K during 125 ps. Next, a five-step equilibration was carried out for 1.75 ns where the restraint force on the protein and lipid atoms were gradually decreased respectively to (1) 5 and 2.5 kcal mol-1Å−2, (2) 2.5 and 1 kcal mol-1Å−2, (3) 1.0 and 0.5 kcal mol-1Å−2, (4) 0.5 and 0.5 kcal mol-1Å−2, and (5) 0.1 and 0 kcal mol-1Å−2. The equilibration was concluded with an unrestrained 10 ns simulation. Finally, a 100 ns production simulation was performed with a 2 fs time step as an isothermal-isobaric ensemble (303.15 K, 1.0 bar) using the Langevin thermostat and the Berendsen barostat[69] to maintain the temperature and pressure, respectively. The Particle Mesh Ewald algorithm[70] was employed to calculate long-range electrostatic interactions and a 9 Å distance cut-off was set for non-bonded interactions. Coordinates were saved every 20 ps and analysis was carried out using the programs CPPTRAJ[71] and VMD[72]. Root-mean-square deviation (RMSD) and root-mean-square fluctuation (RMSF) were respectively computed based on backbone atoms and Cα atoms relative to the starting structure. Hydrogen bonds were monitored by specifying a bond distance ≤ 3.5 Å and a bond angle ≥ to 135°. The details of the software used for the molecular dynamics simulations can be found in the Resources Table in Supplementary Data 10.

### Statistical analyses

For statistical analyses, the corresponding functions in Matlab R2016a (MathWorks) or GraphPad Prism (GraphPad Software) were used. The normality and homoscedasticity of the experimental data were estimated with the Kolmogorov-Smirnov, Shapiro-Wilkis, Brown-Forsythe and Bartlett's tests and nonparametric or parametric testing and Brown−Forsythe correction was utilized accordingly. The statistical significance of comparisons between multiple groups was calculated with either the Mackskill two-way non-parametric ANOVA (analysis of variance), Kruskal-Wallis ANOVA, parametric one-way ANOVA or Brown-Forsythe adjusted one-way ANOVA. Experiments conducted at different times were defined as a covariate for the two-way ANOVA. The visualized posthoc pairwise P values for experiments were calculated with either the two-tailed Mann−Whitney U test, two-tailed T-test or Dunnet's multicomparison test. The post-hoc P-values were Benjamini-Krieger and Yekutieli -corrected for multiple testing

error. Pairwise P values were calculated with either the two-tailed Student's T-test or two-tailed Mann−Whitney U test.

### Data handling and analysis

All boxplots were drawn using a standard function in Matlab R2016a (MathWorks) or with GraphPad Prism (GraphPad Software), where the box represents the interquartile range, the red line the median, whiskers minimum and maximum values, and red crosses outliers as determined by the default settings in the Matlab function. The dots in the boxplots represent replicate experiments in Western quantification data and number of cells in immunofluorescence colocalization data.

The colocalization in immunofluorescence samples was calculated with the algorithm of Villalta et al. 2011[73], that produces 4 different measures of colocalization (Pearson correlation, Manders overlap, m1/m2, and k1/k2), of which only Pearson correlation is shown, although rest of the measures demonstrated a similar trend. The PLA signal intensity in confocal images was measured with FiJi[74] and normalized to the number of nuclei.

The mass-spectrometry interactome data was normalized to the sum of LFQ intensities in every sample fraction and the normalized intensity of each sample fraction were summed to derive the total protein intensity in each sample for each protein. The total protein intensity of each protein in each experiment was additionally median normalized to correct for total abundance differences across experiments. An empirical probability density function was fitted to the total sum of intensities in each experiment of each protein in the vector control sample with an epanechnikov Kernel and the corresponding P-value for JM-a and JM-b condition were estimated from the resulting cumulative density function. The P-values were FDR-corrected for multiple testing error. Proteins with an average pseudolog2 fold change above 0.5 against the vector control and adjusted P ≤ 0.05 were considered significant interactors for JM-a and JM-b conditions. The complexes from the mass-spectrometry-derived interactome data were finally modeled based on experimentally determined interactions found in PSIQUIC[75] and STRING[76] database and the subcellular localization data found in Compartments[77] database. The interactions on PSIQUIC database were accessed through a R vignette[78]. For some interaction partners, no internal interactions among the interactome were found and consequently were solely defined into clusters based on subcellular localization. For the lectin-pulldown masspectrometry data, the proteins identified in the PHAL lectin condition and not in the control condition (only beads) were considered significant interactors.

The sequence motif analysis was performed with a sliding window analysis similarity score function with site specific scoring based on the residues 627-632 in ErbB4 JM-a and in ErbB4 JM-b using PAM250 matrix as a basis for amino acid substitution score. The final sequence model was acquired by only weighing residues in positions 1,4 and 5 or 1,5 and 6 as the only model able to correctly categorize the experimental data. The tested sequence models are listed as a Supplementary Data 5. The P-value for categorization accuracy was estimated from the cumulative probability density function of a binomial distribution where an equal probability for correct and incorrect categorization was assumed (parameter p = 0.5).

The glycan array data were normalized according to total relative fluorescence unit (RFU) values in the sample. Glycan structures with percent coefficient of variation (CV%) values over 50 and RFU values under 50 were categorized as background and filtered out (92.3% of all values). The heatmap values are shown as pseudolog2 fold change of JM-a peptide RFU against JM-b peptide RFU. The enrichment of 1,6 beta branch in the N-glycans that associated with ErbB4 JM-a peptide more than ErbB4 JM-b peptide was calculated by the non-parametric two-tailed Mann-Whitney test with Benjamini-Hochberg correction. In the analysis the normalized RFU values of all filtered N-glycans with certain glycan structure determinants ('Gala1-3','Galb1-3', 'Galb1-4', 'GlcNAcb1-

2', 'GlcNAcb1-3', 'GlcNAcb1-4', 'GlcNAcb1-6', 'GalNAca1-3', 'GalNAcb1-4', 'Neu5Aca2-3', 'Neu5Aca2-6', 'Mana1-3', 'Manb1-4', 'Mana1-6', 'Fuca1-2', 'Fuca1-3', 'Fuca1-6') were compared against the RFU values of N-glycans that associated similarly or less with ErbB4 JM-a peptide than with ErbB4 JM-b peptide (less than 50% more affinity for ErbB4 JM-a).

The cluster analysis of JM-a-like RTKs and JM-b-like RTKs was performed with neighborhood component analysis. To estimate a P-value for the clusterization, the clustering of 1000 randomly assigned categories was simulated, and the relative distance within category and between categories was calculated. A probability distribution function was fitted to the relative distance scores with an epanechnikov kernel and the corresponding P-value for the relative distance score of the JM-motif categorization was drawn from the cumulative probability distribution function.

The eJM sequence motif related interactome of JM-a-like RTKs and JM-b-like RTKs was determined by statistical testing. Due to the zero-inflated nature of the interactome data, the data was transformed to binary data and the P-values were drawn from the binomial cumulative distribution function. Proteins with 5-fold representation in their respective groups compared to the other group and with a P-value less than 0.05 were chosen.

The pathway analysis of the ErbB4 JM-a and JM-b interactome was performed by estimating a value for a pathway annotation for each sample with stochastic neighborhood embedding. The pathway annotations were acquired from the Molecular Signatures Database v7.5.1 (MsigDB)[79–84]. Only pathway annotations where at least 5 genes were present in the analyzed datasets were utilized. The differential pathway expression analysis was conducted with two-tailed Mann-Whitney U-test to find pathways significantly differentially regulated by ErbB4 JM-a and ErbB4 JM-b.

The localization enrichment analysis of the ErbB4 JM-a and JM-b interactome and the JM- sequence motif related JM-a-like and JM-b-like interactome data was performed with PANTHER Overrepresentation Test[85] (Released 20220202). The GO[86,87] cellular component annotation dataset (v 10.5281, Released 2022-03-22) and the Fisher's Exact test was utilized. The statistical significance of the overlap of sub-cellular localization annotations between JM-a-like and JM-b-like RTKs was estimated by randomly drawing 30 sample sets of the same size from the MS-AP and BioID interactome data and estimating the overlap of their enriched subcellular locations. A normal probability distribution function was fitted to the overlap values and the corresponding P-value was drawn from the resulting cumulative probability distribution function The statistical significance of the overlap of subcellular localization annotations between ErbB4 JM-a and JM-a-like RTKs and ErbB4 JM-b and JM-b-like RTKs was performed with the $\chi^2$-test.

### Reporting summary

Further information on research design is available in the Nature Portfolio Reporting Summary linked to this article.

## Data availability

The raw MS proteomics data have been deposited to the ProteomeXchange Consortium via the PRIDE[88] and Panorama Public[89] partner repositories with the data set identifiers PXD017783, PXD026546, and PXD026617. The protein abundances derived from the analyzed MS proteomics data are additionally provided as Supplementary Data 1-4 and 9. The raw data from the glycan screen are provided as a Supplementary Data 8. The raw data on the molecular dynamics simulations have been supplied to Mendely data: https://doi.org/10.17632/y7b9mgdhrb.1. The ErbB4 structures 3U7U and 2LCX, can be accessed through PDB. The raw data on RTK interactomes can be accessed through MassIVE with dataset ID MSV000087816 [https://doi.org/10.25345/C5KJ9Q]. The subcellular locations on COMPARTMENTS (https://compartments.jensenlab.org/), protein-protein interactions on STRING (https://string-db.org/) and PSIQUIC (http://www.ebi.ac.uk/

Tools/webservices/psicquic/view/main.xhtml) and sequence information in Uniprot (https://www.uniprot.org/) can be accessed through the respective website of the database. Source data for Figs. 1–6 and Supplementary Figures 1-15 are provided with this paper. Additional source data that support the findings of this study are available from the corresponding author upon request. Source data are provided with this paper.

## Code availability

The R code used to discover JM sequence motifs in the eJM sequence of receptor tyrosine kinases is available through github: https://github.com/kvaparanta/JM_motif_algorithm.

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

## Acknowledgements

We thank Minna Santanen, Mika Savisalo, and Maria Tuominen for excellent technical assistance, Jukka Lehtonen (Biocenter Finland bioinformatics network) for scientific IT support and CSC IT Center for Science for supercomputing. STAT5b tyrosine to phenylalanine point mutant plasmid constructs were a kind gift from the laboratory of Jeffrey M. Rosen. We thank Sauli Haataja for loaning lectin reagents. Turku Doctoral Program of Molecular Medicine, Åbo Akademi Graduate School and Doctoral Network of Informational and Structural Biology are acknowledged for doctoral training. This work was financially supported by the Academy of Finland, the Cancer Foundation of Finland, the Sigrid Juselius Foundation, the Turku University Central Hospital, the Cancer Society of Southwestern Finland, Turku University Foundation, the Finnish Cultural Foundation Varsinais-Suomi Regional Fund, the Finnish Foundation for Cardiovascular Research, The Emil Aaltonen Foundation, The Maud Kuistila Memorial Foundation, Orion Research Foundation, K. Albin Johanssons stiftelse, the Foundation of Åbo Akademi and the Joe, Pentti and Tor Borg Memorial Fund.

## Author contributions

K.V. and K.E. designed the experiments and interpreted the main results. K.V. conducted the biochemical, cell biological, and bioinformatics analyses. A.J. participated in biochemical and cell biological experimentation. J.A.M.M. participated in cloning and mass spectrometry analyses. M.T. and M.S.J. performed the structural analyzes. The contribution of J.I. was essential with the visualization of the super-resolution microscopy images and formulation of the rebuttal letter. K.S. and M.V. provided us with the RTK interactome data. K.V. and K.E. wrote the manuscript with the help of A.J., J.A.M.M., M.T..M.S.J., and J.I.

## Competing interests

The authors declare no competing interests.
