## [Peer Review File · Nature Communications]

Reviewers' comments:

Reviewer #1 (Remarks to the Author):

Vaparanta et al investigate differences in Stat activation for ErbB4 isoforms, seeking molecular explanation. The paper leads with the assertion that overexpression of the JM-a isoform leads to STAT5a phosphorylation whereas overexpression of the JM-b isoform causes STAT5b phosphorylation. The data to support this conclusion are weak. The authors then generate chimeric receptors (a/b and b/a) where nine amino acids are swapped between the two isoforms. It is claimed that the STAT5 activation tracks with the nine amino acids so that the a/b variant behaves like JM-a and the b/a variant like JM-b. Again, the data for this are not convincing. The authors then attempt to find a molecular explanation for the differences. The only structural information available is for the JM-a isoform. The authors model the JM-b isoform by simple side chain replacement and argue that there will be no mainchain conformation change the loops between C617 and C633 due to the disulfide bonds. I disagree. The loops are long enough that they could adopt different conformations – especially the loop that differs between the two isoforms, which contains a proline (P628) in JM-a that is a serine in JM-b. Beyond the last disulfide (C633), no statement can be made about the structure, which will not be the same in the context of isolated fragments or ErbB4 and the intact membrane protein. The section “ErbB4 eJM sequence differences in the ErbB4 crystal structure” should be removed. The MD simulations add nothing to the analysis and should be removed. The primary sequence comparison is all that is needed to identify the His to Glu change at position 631 as of potential significance in the 626-634 region. The data in Figure G, H are inadequate to claim that H631 is required the STAT5a selectivity of ErbB4 JM-a. The very minor differences in phosphorylation, for STAT5b in particular, may be explained by differences in expression levels of the mutated ErbB4 JM-a and/or differences in the amount of receptor at the cell surface, which should have been assessed in live cells with FACS analysis.

Using sequence analysis, the authors identify other RTKs with similar “motifs” to this region of ErbB4 JM-a and JM-b, and they analyze STAT5 phosphorylation following overexpression of a subset of these RTKs. The analysis suffers the same problems as for the JM-a/JM-b comparison – differences are very small and statistical significance is not demonstrated.

Mass spectrometry was then used to assess differences in proteins chemical cross-linked to ErbB4 in co-immunoprecipitates from in MDA-MB-468 cells overexpressing ErbB4 JM-a or JM-b. I feel that insufficient information is given about the mass spectrometry to assess the quality of these data. Based in part on differences in the interactome of the JM-a and JM-b isoforms, the authors next investigate differences in co-localization of a range of RTKs and variants in fixed cells using STED microscopy.

To identify a function for the eJM region, the authors turn to interaction analysis with a cyclized synthetic peptide. The cyclized peptide choice is odd. The lack of the C617-C625 disulfide will completely change the structure of the 621-633 segment, so this is of no relevance to the eJM structure. Nothing can be concluded from the analysis of this peptide with lipids or glycans. The

subsequent experiment with lectins show limited effects and these are not demonstrated to be linked to ErbB4. These data do not support the conclusion that the eJM contains a glycan binding site.

Overall, I felt that the data did not strongly support the authors conclusions, especially for the leading Western blot experiments. It is possible that this would seem less problematic if the paper was organized in a different way. The difference in the ErbB4 isoforms is well established in the literature, so, perhaps leading with an expansion of the mass spectrometry analysis and co-localization experiments might allow a difference emphasis that would rely less heavily on the somewhat marginal difference between the Stat5a and Stat5b phosphorylation.

Additional major points:

1. Much of the data in this manuscript relies on Western blot observations – all using overexpressed receptors. The differences observed are not always clear, and at times there is inconsistency between the shown blot and the quantitation (see below). Combining results from multiple cell types – some with added NRG and some without – dilutes rather than enhances confidence. It is, for example, unclear whether there is any difference in levels of pSTAT5a with the JM-a or JM-b isoform using HC11 cells since there is only one repeat with this cell shown.
2. In Figure 2B there is clearly a much higher level of phosphorylation of STAT5b for the JM-b/a chimera than for the JM-b isoform. This is stated in the text. Yet the quantitation in panel C shows no difference for these two proteins. Since the variation between the two repeats (based on number of displayed points) is extremely small, it is not clear how the data in Figure 2B could lead to the data in Figure 2C (lower panel). The number of biological repeats need to be stated for every experiment in the manuscript, and statistical significance in the differences from these Western blot data reported.
3. The structural modeling adds little to the paper and is misleading. The concatenation of coordinates from 2LCX onto 3U7U is arbitrary. There is no overlap between the two structures – indeed there are two missing amino acids. Only one of the molecules in 3U7U is modeled to amino acid 639, with significantly less of the eJM region modeled for the other molecules. 2LCX starts at 642 but there are very limited restraints on mainchain conformation until the start of the TM helix. The structural model in Figure 3K should be removed – it is clear that 631 and 632 must be close to the membrane. Additional details implied by this figure are not useful and potentially highly misleading.

4. How was the analysis to obtain the sequence motifs conducted? The motif is very general and should have some statistical significance associated with it – does this occur more frequently in the eJM regions of RTKs compared to similar regions of other single TM domain proteins?

5. In Figure 3, receptor expression levels vary between different RTKs. It is unclear whether this is taken into account in comparing STAT5 phosphorylation levels. The quantitation from the Western blots is poorly described. Normalization procedures are not consistent across all experiments, and some data are background corrected.

6. The number of cells analyzed in the STED microscopy experiments should be noted.

Minor points

1. Numbering for the JM-b consensus motif is wrong in text and figure – it starts at 627 (as for JM-a). Sequence numbers should all be verified, and a note added to related the sequence numbering here to those used in the pdb files.

2. There is inconsistency in whether or not NRG stimulation used – for example text states that NRG not used for MDA-MB-468 cells due to high basal ErbB4 activation, but Figure 1K shows these cells stimulated with NRG.

Reviewer #2 (Remarks to the Author):

Vaparanta et al have investigated the underlying mechanism for selective signaling of two naturally occurring splice variants of ErbB4 RTK that differ in their membrane proximal extracellular juxtamembrane (eJM) domain. The JM-a isoform activated STAT5a and JM-b STAT5b and STAT3 and eJM cleavage was not determining selectivity of signaling. Surprisingly, expression of TYK2 but not other JAKs was involved in ErbB4 JM-b mediated STAT5b activation while the JM-a mediated STAT5a activation is not dependent on any JAK kinase. Kinase activity of TYK2 was not required for STAT5b activation suggestion a scaffolding role for the protein.

STAT5a activation by JM-a was found to rely on structural characteristics of eJM and particularly H631 and D632 residues were found to be critical possibly via hydrogen bond formation that would

enable selective coupling with extracellular molecules. In JM-b the 631-632 residues did not affect signaling. Glycosylation of ErbB4 was not affecting the signalling characteristics and both variants bound equally to lipids but JM-a was found to bind selectively complex N-glycans. B1 integrin was identified as a selective ErbB4 eJM domain binder suggesting that N-glycans mediate interaction between ErbB4 eJM and B1 integrin that target ErbB4 to specific cell surface compartment for STAT5a selective activation.

This is a well-written profound characterization of two ErbB4 variants that identified novel mechanisms for selective signalling and STAT5 activation. Extensive methodology has been exploited and the results seem reliable. Some Westerns are a bit hazy but the quantifications seems right. Molecular dynamic simulations and structural analysis seem reliable.

Specific comments

Characterizing the eJM sequence motif as “consensus” in Fig.3A may not be exactly right term as quite many RTKs do not seem to follow that. More extensive analysis of glycan-lectin sensitivity of RTKs could strengthen the concept.

Comparison of phosphoproteomes in endogenously ErbB4 JM-a and JM-b expressing cells after ligand stimulation would be interesting to obtain a more objective insight of signalling landscape, obviously the cell type differences will cause some variation.

Reviewer #3 (Remarks to the Author):

ErbB4 is the ErbB4 family member whose signaling activities/functions are perhaps least well characterized. The authors here characterize the differential signaling by two ErbB4 splice variants designated JM-a and JM-b, whose sequences differ only in the extracellular juxtamembrane domain. The primary observation is that the JM-a and JM-b isoforms preferentially activate the downstream signal transducers Stat5a and Stat5b, respectively. The authors rule out various mechanisms that might explain the differential signaling of JM-a and JM-b (such as differing susceptibility to extracellular proteases), and ultimately make conclusions regarding their differential signaling, including that the isoforms might traffic differentially to membrane subdomains (caveolae), that only JM-b appears dependent upon the Tyk2 kinase, that the JM-a and JM-b sequences show differential interactions with glycan among a commercial array of glycan substrates, and that JM-b appears to interact with a specific N-glycan-linked molecule present in the cell lines examined (specifically integrin-beta-1). It is suggested that a JM-b/integrin-beta-1 interaction is somehow involved in JM-b-specific activation of Stat5b.

The authors present a large body of data, generated with a wide-ranging combination of methods, in their attempt to elucidate the mechanism(s) of ErbB4 signaling to Stat5a and Stat5b. However, while the essential conclusions of the manuscript might be valid, the manuscript stops short of elucidating the full mechanism by which Stat5b is activated by JM-b, and in many cases presents ancillary observations of unclear significance to the activation mechanism. It appears yet more uncertain how Stat5a is activated by JM-a, although the mechanism is apparently different in ways from that of JM-b/Stat5b. It seems then that the manuscript is somewhat unfocused and attempts to present too many observations, versus solidly nailing down a specific mechanism. Although the proposed mechanism is an interesting one and apparently might extend to a number of different cell surface receptors, it might be premature to publish it in this form.

There are also concerns with the quality of some of the experimental data. Various Western blots do not seem to show very definitive bands, although quantification of replicates blots was apparently done to demonstrate significance. For example, the blots of Sup. Fig. 1A-D, which aims to demonstrate a differential sensitivity to ErbB4 phosphorylation site mutation, do not seem to show a convincing effect. Also, the conclusions about Stat5 translocation to caveolin-rich domains (Sup. Figure 1E-G) seem questionable in that "translocation" (movement) is not really analyzed and the entire gradient was not analyzed to see exactly how Stat5b and ErbB4 were distributed among the various fractions. Another example is the reciprocal immunoprecipitation used to show ErbB4 integrin-beta-1 association (Fig. 7I), in which at least one of the two IPs does not show a conclusive interaction.

A minor point is the ErbB4 variants CYT-1 and CYT-2 are presented, but the reader will likely not know the meaning of these terms and the purpose for which they are used is unclear.

Reviewer #4 (Remarks to the Author):

The manuscript by Varapanta et al explores the significance of differences in the extracellular juxtamembrane (eJM) domain of the natural juxtamembrane isoforms of ErbB4. This is an interesting and worthwhile subject of research, not just in the ErbB4 field, but also as part of the community endeavours to understand structure-function relationships in receptor tyrosine kinases (RTKs).

ErbB4 is an ideal target for these studies because it is the only member of the EGFR family that has naturally evolved into two eJM isoforms. It is also known that there is differential activation of STAT proteins not just by the two ErbB4 isoforms, but also in several other RTKs.

Compelling results are included in this manuscript. For example, the link between the JM-a isoform and the recruitment and activation of STATa and JM-b and STATb and STAT3, the importance of residue H631 for selective STATa activation by JM-a, and the consensus motif in the eJM.

The molecular dynamics simulations don't appear to show a big difference between the two isoforms. It would have been reasonable to expect (and one imagine the authors may have thought the same) that a change of JM structure might explain at least some of the differences in isoform function. Indeed, that change would have been revealing in explaining how extracellular interactions could initiate and propagate diverging signalling events across the membrane. However, as recorded in the literature, and also in my own experience, the absence of differences in MD simulation results in solvated protein fragments often don't correlate with the findings if the membrane bilayer is present (this would be potentially more acute for the JM segment). Restraints from other protein domains may also matter. Without both of this, it is hard, in my opinion, to draw many conclusions from the MD results.

My main concern is with the STED experiments and the conclusions drawn regarding the localisation of RTKs into different plasma membrane microdomains as the means to controls a subtype specific STAT activation mechanism.

The STED experiments are not at all well described.

I could not find a description of the type of STED deployed here: Is it 2D or 3D? What is the resolution and/or the NA of the objective? There is no scale bar in the figures.

The STED data show what it seems to be to be zoom in images from what it appears to be membrane cross-sections. To have a zoom out image from where these regions have been selected would be quite helpful.

If the orientation of the plasma membrane in the images is as it seems, I am unsure of how these experiments can be robustly interpret.

If one is looking for domains on the plasma membrane, I would assume one needs to look at the plasma membrane head on, i.e. to image the x,y in-plane of the membrane. If you look at the plasma membrane sideways (x,z or y,z direction), i.e. like in an equatorial section of a confocal microscope, which is what it looks like to me in the images provided, then if STED is performed in 2D one would see a >500 nm projection (i.e. the membrane in cross-section). In 3D one would still be projecting across >80 nm depth. I fail to see how one would draw robust conclusions about protein colocalisation on the x,y plasma membrane plane if one is looking at x,z or y,z projections.

If, as it looks to me from the images, this is the case, why has such an orientation been chosen? The deleterious risks of doing so would need to be explained or rectified by recording x,y areas. It is difficult to ascertain what has been done. All this needs to be corrected, explained, or both.

There also appears to be chromatic aberration in some STED images. If so has this been considered? This will change the results.

Other points.

The means of analysis could also be considered, for example, whether the presence of clusters may affect the means to analyse colocalisation using Pearson?

The rationale of the choice of the four cell lines could be better explained; part of it becomes clear throughout the manuscript but it is on occasions confusing.

In Fig. 1B, the two spliced isoforms of ErbB4 are tested. This could be mentioned in the introduction to help readers outside the EGFR field.

End of second paragraph, page 5, I would replace 'structure' with 'primary structure'

When overexpression is mentioned, it would be helpful to get an idea of what this is meant to be and whether it is reproducible.

Typos: In Methods substitute Abbrerrior for Abberior (twice).

Reviewer #5 (Remarks to the Author):

The manuscript by Vaparanta et al. describes the discovery of a sequence motif in the extracellular juxtamembrane regions of ErbB4 receptor isoforms JM-a and JM-b, which is responsible for selective activation of STAT5a or STAT5b and subsequent differential cellular effects. The authors show that this feature is not restricted to ErbB4 isoforms but its present in many RTKs. The study suggests that the JM-a and JM-b receptors associate with different signaling complexes at distinct compartments at the cellular surface. JM-a preferentially interacts with beta1-integrin via N-glycans, while Tyk2 association with JM-b is necessary for the activation of STAT5b. It is potentially very interesting and relevant discovery. The authors have done massive amount of work involving numerous experiments and advanced technologies. However, the technical quality of many of the experiments is unfortunately quite poor, questioning the validity of findings and claims.

Specific points:

1 > Virtually all claims and conclusions in the manuscript are based, and rely entirely, on accurate quantitation of small differences among various experimental conditions. And in many cases, the differences do not appear dramatic. Statistical significance tests should be performed and shown for all quantitative results (all boxplot panels in all figures).

2 > The quality of some Western blots (WBs) used for quantitation is very poor (e.g. Fig. 2B, Fig. 3L, Supplementary Fig. 1A and 1C, Supplementary Fig. 4A). And in some cases, it seems to be discrepancies between the WBs and the quantitation derived from it. For example, on the WB in Fig. 3L phosphoSTAT5b band seems ~ 2x stronger in the NRG-1 treatment compared to control, while the quantitation of it on Fig. 3M shows reduction. Similar in Suppl. Fig. 1C and 1D, phosphoSTAT5b band on the WB is the weakest in lane 1, while the quantitation is telling us that it's the strongest among all. Similarly, it is very difficult to imagine how the WBs on Suppl. Fig. 4A have yielded the quantitative data shown in main Fig. 3G. Please explain?

3 > Authors state that ErbB4 JM-b co-immunoprecipitates more STAT5b based on the experiment shown on Fig 1F. However, the figure shows IP of ErbB4 followed by WB with phospho STAT5b antibody. The phosphoSTAT5 antibody recognizes equally well both STAT5a and STAT5b. The stronger band in the JM-b IP could as well be STAT5A. The experiment should be done correctly, similar to the STAT5A experiment (Fig 1E).

4 > Page 5: "...overexpression of ErbB4 JM-a promoted the translocation of STAT5b to caveolin-1-rich membrane microdomains in MDA-MB-468 cells (Supplementary Fig. 1E-G), in which STAT5 is dephosphorylated". Equivalent experiment for STAT5a should be presented for comparison - is overexpression of ErbB4-JMa driving STAT5a to a different location? Or STAT5a is also translocated to the caveolin-1-rich microdomains – but then it should get dephosphorylated like STAT5b?

5 > The mass spectrometric data for the JM-a/b protein complexes is unacceptable in its current form. The variability in label-free quantitation is bigger compared to other mass spectrometric approaches. Therefore, it strictly requires minimum of 3-5 replicates and proper statistical analyses of the data in order to define with higher probability the differences between the samples. Most of the proteins presented in the manuscript as ErbB4 JM-a/JM-b interactors are based on a single experiment. Label-free quantitation based on 1-2 replicates is very dangerous and much of the observed differences can be false. And thereby many of the assumed interactors. Especially when combined with puny criteria for accepted changes. To convince the reader that this is not the case, the authors should present the overlap of the two experiments as well as correlation analysis, e.g. Pearson.

In this respect, 7 out of the 9 complexes shown as unique to JM-a are based on Grb2 and Src, and both proteins are described as exclusively interacting with the JM-a isoform (Fig 5A). Consulting with Suppl. Table, Src is found only in one of the two experiments and identified by merely 2 peptides (and we have no information what criteria were used for accepting peptide identifications, comment below!). The values given for Grb2 in the same experiment for Control:JM-a:JM-b are 15:14:13. How could these proteins be counted as exclusively interacting with the JM-a and dictating most of the conclusions from these experiments?

6 > What is currently done, in terms of mass spectrometric analyses, may be ok for cherry-picking individual candidates for follow-up experiments amongst the proteins with clear-cut differences between conditions. But accepting the current mass spectrometric analyses as bonafide evidence that all these proteins are in complex with ErbB4 will be naïve and wrong. Also, different criteria for “interactors” were used for the 2 datasets? The criteria for processing the data should be identical to avoid biases. In addition, no information is provided regarding peptide identifications: what search parameters were used, the criteria used for accepting peptide identification – FDR, minimum peptide score, etc.? The reader is left with no other choice, but to trust that all is done correct!

The analyses need to be performed in a minimum 3 replicates, data should be properly analyzed, and only reproducible & significant changes must be considered as ErbB4 JM-a/JM-b interactors. Otherwise, these sections should be removed from the manuscript.

7 > Tables with identifications from the PHA-L pull-downs should be included in the supplementary as well.

Response to Reviewers

Comments received from the Reviewers on 16th of April 2020 are included in plain font, followed point-by-point by the authors' comments in bold.

Reviewer #1 (Remarks to the Author):

Vaparanta et al investigate differences in Stat activation for ErbB4 isoforms, seeking molecular explanation. The paper leads with the assertion that overexpression of the JM-a isoform leads to STAT5a phosphorylation whereas overexpression of the JM-b isoform causes STAT5b phosphorylation. The data to support this conclusion are weak.

--- We thank the reviewer for the insightful comments.

In the manuscript we provide experimental evidence supporting the difference in STAT5 subtype activation by ErbB4 JM-a and JM-b by the following methods: i) Western analysis on ErbB4-induced STAT5 activation (original Fig. 1A-D, new Fig. 3A-D and new Suppl. Fig. 4A-B; original Fig. 2B-C, new Suppl. Fig. 7C-D; original Fig. 2G-H, new Fig. 4D-E; original Fig. 3G-H, new Fig. 5E-F; original Fig. 3G-H, new Fig. 5E-F; original Fig. 6A-D, new Fig. 2F-G; original Suppl. Fig. 9F), ii) co-immunoprecipitation of STAT5 with ErbB4 (original Fig. 1E-F, new Fig. 3E-F), and iii) immunofluorescence analysis of ErbB4-induced nuclear localization of STAT5 (original Fig. 1G-J, new Fig. 3G-J; Rebuttal Fig. 1 below).

As recommended by the Reviewers, we have now expanded the number of repetitions for several experiments enabling improved statistical analyses of the Western data on ErbB4 JM isoform-mediated STAT5 activation (new Fig. 3C-D). While carrying out the experimentation with endogenous ErbB4 isoforms in a single cell background was not feasible due to the fact that the expression of ErbB4 JM-a and JM-b is mutually exclusive (Elenius et al., 1997. J. Biol. Chem. 272:26761-8; Veikkolainen et al., 2011. Cell Cycle. 10:2647-57), the basic finding obtained with ectopic ErbB4 expression was obtained in several different cell contexts (COS-7, HC11, MCF-7, MDA-MB-468) (new Fig. 3A-D; new Suppl. Fig. 4A-B; please see also Rebuttal Fig. 1 below). The authors feel that these independent lines of evidence strongly support our conclusion of different ErbB4 isoforms activating the different STAT5 subtypes.

Rebuttal Figure 1. Nuclear localization of STAT5 subtypes in various cell lines expressing ErbB4 JM isoforms. The indicated cell lines overexpressing ErbB4 JM-a or JM-b (A and B) or endogenously expressing ErbB4 JM-a (C and D) were stimulated with NRG-1, fixed with methanol, and subjected to immunofluorescence staining with the indicated antibodies and dyes. Confocal images from the middle of the nuclei were acquired. The relative average staining intensity in the nucleus was measured with ImageJ. Two-tailed Mann-Whitney U test was used for statistical analyses.

The authors then generate chimeric receptors (a/b and b/a) where nine amino acids are swapped between the two isoforms. It is claimed that the STAT5 activation tracks with the nine amino acids so that the a/b variant behaves like JM-a and the b/a variant like JM-b. Again, the data for this are not convincing.

--- As suggested by the Reviewers, we have now repeated the experiment with chimeric receptors and performed statistical analysis on the Western quantification. As other data even more relevant for the structure/function relationship of the eJM sequence are extensively provided in the manuscript, and we aimed at a more concise presentation following recommendations by the Reviewers, these data are now shown in the Supplement (new Suppl. Fig. 7C,D). The results from the new statistical analyses support our original hypothesis that the most N-terminal 9 amino acid residues in the variable JM region include structures central for the unique properties of the JM-a isoform.

The Reviewer may have rightfully paid attention to the observation

that, while the JMb/a chimeric construct nicely recapitulates the activity of JM-b, the JMa/b construct does not seem to fully recapitulate the activity of JM-a. This seems to be a reproducible phenomenon that is consistent with the concept of JM-a sequence being actively functional and JM-b sequence representing a passive lack of the JM-a sequence. In this scenario, one can envision that a change C-terminal to the critical 9 N-terminal JM-a amino acids could still affect the active JM-a motif function, *e.g.* by changing the orientation of the active motif, its distance from the plasma membrane, or its conformation. However, given the inherent difficulties in reliably interpreting data from large mutational analyses, such as those carried out by our chimeras, we quickly moved on in the manuscript to address the same questions with mutagenesis covering only one or two residues (new Fig. 4D-E; new Fig. 5I-J; new Suppl. Fig. 9C). Importantly, these experiments produced data in line with the analysis of the chimeras. ---

The authors then attempt to find a molecular explanation for the differences. The only structural information available is for the JM-a isoform. The authors model the JM-b isoform by simple side chain replacement and argue that there will be no mainchain conformation change the loops between C617 and C633 due to the disulfide bonds. I disagree. The loops are long enough that they could adopt different conformations – especially the loop that differs between the two isoforms, which contains a proline (P628) in JM-a that is a serine in JM-b.

--- We have indeed highlighted in the manuscript that a proline to serine mutation in other circumstances would lead to structural changes. However, given the loop that accommodates the mutations between C617 and C633 is constrained by two disulfide bridges, we suggested that there might not be a significant backbone conformational change. This initial statement made after examining the X-ray structure of ErbB4 is also supported by a 100 ns molecular dynamics simulation, which showed small (average) and almost identical backbone atom deviations for the disulfide constrained regions of JM-a and JM-b (new Fig. 4B,C). Furthermore, we have now modeled JM-b based on the JM-a structure using the Modeller program in contrast to our previous conservative strategy of side-chain replacement. The new model is essentially identical to the previous model (new Fig. 4A) at the disulfide constrained region that contains P628S, again in-line with our statement on the main-chain conformational invariability of this region. ---

Beyond the last disulfide (C633), no statement can be made about the structure, which will not be the same in the context of isolated fragments or ErbB4 and the intact membrane protein.

--- We also now have carried out the MD simulation on the ErbB4 structure with the intact transmembrane domain: Similar structural stability in terms of the root-mean-square fluctuations (new Fig. 3C) was observed for the two isoforms as reported for the previous simulations where the transmembrane domain was not included. The new analysis, however, gives additional insight on the interactions involving the mutated residues, particularly the key amino acids H631/E631 and their interactions within ErbB4 and with the lipid bilayer. ---

The section “ErbB4 eJM sequence differences in the ErbB4 crystal structure” should be removed. The MD simulations add nothing to the analysis and should be removed. The primary sequence comparison is all that is needed to identify the His to Glu change at position 631 as of potential significance in the 626-634 region.

--- We believe the structural analysis does provide insight regarding the conformational states and interactions of JM-a and JM-b. It highlights that the two ErbB4 isoforms exhibit minor conformational differences but a pair of sequence differences in JM-a and JM-b are still capable of forming different interactions both within the protein and the surrounding membrane bilayer. ---

The data in Figure G, H are inadequate to claim that H631 is required the STAT5a selectivity of ErbB4 JM-a.

--- In addition to original Fig. 2G-H (new Fig. 4D-E) we have carried out extensive analyses of STAT5a and STAT5b phosphorylation with individual point mutants (original Fig. 3I-J, new Fig. 5I-J). New statistical analysis of the quantitation of these Western analyses indicated a statistically significant difference (P-value less than 0.05) for the activation of STAT5a between wild-type and H631E or S630I/H631E mutant ErbB4 JM-a (new Fig. 4E; new Fig. 5I).

In addition to STAT5 phosphorylation, we provide data demonstrating that the H631E or S630I/H631E mutants regulate the cell surface localization of ErbB4 JM-a (original Suppl. Fig. 7, new Suppl. Fig. 8), as well as coprecipitation (original Fig. 7I, new Fig. 7H) and colocalization (original Fig. 7J-K, new Fig. 7I-J) of ErbB4 JM-a with β 1 integrin. All these observations are consistent with the H631 residue being important for the STAT5a selectivity of ErbB4 JM-a. ---

The very minor differences in phosphorylation, for STAT5b in particular, may be explained by differences in expression levels of the mutated ErbB4 JM-a and/or differences in the amount of receptor at the cell surface, which should have been assessed in live cells with FACS analysis.

--- As the Reviewer’s comment was possibly raised from interpreting the Western data shown in original Fig. 2G (right panel) which included a blot with uneven loading of ErbB4 JM-a and JM-b, we have now replaced the representative blot shown in the new Fig. 4D (right panel). Please note that the expression of the mutant ErbB4 JM-a in this new blot is even below the level of wild-type JM-a or JM-b expression, while STAT5b is still phosphorylated at a level similar to JM-b, indicating that the modest experimental variation in the expression of the ErbB4 JM-a does not explain the difference in STAT5b activation. The actual experiment has now also been carried out for one more time enhancing also the statistical power of the quantitation shown in the new Fig. 4E.

The authors fully agree with the Reviewer that differential cell surface expression can cause differences in the level of downstream signaling. To experimentally address the issue, we have now quantified the cell surface expression of the ErbB4 JM-a mutant vs wild-type JM-a in the context used for the STAT5 phosphorylation analyses. As FACS analyses are suboptimal for addressing the cell surface expression of the protease-sensitive ErbB4 JM-a in adherent cells, we chose to quantify the cell surface

protein level with immunofluorescence staining of fixed cells in their natural adherent environment. These data demonstrate no significant differences in the cell surface immunostaining between wild-type ErbB4 and the two JM-a mutants S630I/H631E or H631E (Rebuttal Fig. 2).

Rebuttal Figure 2. Quantification of ErbB4 protein at the cell surface. Methanol fixed MCF-7 (A) and COS-7 (B) cells overexpressing the indicated HA-tagged ErbB4 variants were stained with anti-HA and imaged with immunofluorescence microscopy. Immunosignals at the cell surface were quantified with Image J. The statistical significance in A was assessed with one-way ANOVA and Tukey's multicomparison test was used for post-hoc analyses. The statistical significance in B was assessed with two-tailed Mann-Whitney U test. One dot corresponds to one cell. ns = non significant ($P > 0.05$).

Using sequence analysis, the authors identify other RTKs with similar “motifs” to this region of ErbB4 JM-a and JM-b, and they analyze STAT5 phosphorylation following overexpression of a subset of these RTKs. The analysis suffers the same problems as for the JM-a/JM-b comparison – differences are very small and statistical significance is not demonstrated.

--- We have now estimated the statistical significance for all Western analyses of other RTKs (new Fig. 5B,D,E,F) and repeated the experiments presented in the original Fig. 3C-F (new Fig. 5A-D). The new statistical analyses all support our hypotheses. Additionally, we have expanded the amount of other RTKs we have analyzed by researching the STAT5 activation selectivity of endogenous PDGFRA, FGFR1/FGFR2 and EGFR/ErbB2 (new Fig. 6A-B). The results also from these analyses support our hypotheses. ---

Mass spectrometry was then used to assess differences in proteins chemical cross-linked to ErbB4 in co-immunoprecipitates from in MDA-MB-468 cells overexpressing

ErbB4 JM-a or JM-b. I feel that insufficient information is given about the mass spectrometry to assess the quality of these data.

--- We have now repeated the mass spectrometry experiments three additional times and added a more detailed description of the methods in the Materials and Methods section. ---

Based in part on differences in the interactome of the JM-a and JM-b isoforms, the authors next investigate differences in co-localization of a range of RTKs and variants in fixed cells using STED microscopy.

To identify a function for the eJM region, the authors turn to interaction analysis with a cyclized synthetic peptide. The cyclized peptide choice is odd. The lack of the C617-C625 disulfide will completely change the structure of the 621-633 segment, so this is of no relevance to the eJM structure. Nothing can be concluded from the analysis of this peptide with lipids or glycans.

--- We thank the reviewer for the insightful comment. Indeed the lack of a disulphide bridge can potentially disrupt the conformation of the peptide, which is overall less constrained by the lack of the neighboring regions than the corresponding region in the full-length ErbB4 structure. To ensure that the synthesis product will only include peptides with the C621-C633 bridge which is essential to the cyclization of the peptide, we needed to substitute the C625 to S. If we had chosen to either not substitute the C625 or synthesize a longer peptide including the C617, the end product would have contained a mix of peptides with varying disulphide bridges (C617-C625, C617-C621, C617-C633, C621- C625, C621-C633, C625-C633) and varying cyclization. To avoid this eventuality we decided on the substitution.

To further alleviate the reviewers concerns over the cyclized peptides, we have performed a 20 ns molecular dynamics simulation on the two cyclic peptides to assess the structural dynamics of the peptides and examine whether the peptide fold is preserved without the C617-C625 disulfide bridge. The simulations revealed that the peptides largely maintain a similar conformation relative to their initial structures, which is based on the ErbB4 crystal structure (Rebuttal Fig. 3). The average RMSDs for the isoform 1 and isoform 2 peptides were 1.57 Å and 1.59 Å, respectively.

Rebuttal Figure 3. Sampled conformations during the simulation for the ErbB4 JM-a and JM-b cyclic peptides. First and last sampled conformations are colored green and pink, respectively. Other recorded conformations are in faded khaki.

The subsequent experiment with lectins show limited effects and these are not demonstrated to be linked to ErbB4. These data do not support the conclusion that the eJM contains a glycan binding site.

--- In the experiment with lectins, the ErbB4-mediated STAT5 activation was assessed in COS-7 cells overexpressing ErbB4 JM-a or JM-b (original Fig. 7C-D, new Fig. 7C-D). This has now been indicated more clearly in the text of Results on p. 17. We have now also carried out a statistical analysis of independent experiments, and the data demonstrate a significant reduction of ErbB4 JM-a-mediated STAT5a activation with the lectins DSL, STL and PHA-L (new Fig. 7C). As a control of non-specific effects, the same lectins did not affect JM-b-mediated STAT5b activation (new Fig. 7D).

Data consistent with the glycan-binding lectins specifically interfering with the eJM of ErbB4 JM-a is also provided by analyses demonstrating that lectins reduced the colocalization and abolished the interaction of JM-a, but not of the point mutant H631E JM-a, with β 1 integrin (original Fig. 7I-K, new Fig. 7H-J). These data are corroborated by assays demonstrating that lectins specifically blocked ErbB4 ligand-stimulated, but not insulin-stimulated, growth of cells endogenously expressing both ErbB4 JM-a and the JM-b-like insulin receptor (original Fig. 7E-F, new Fig. 7E-F). Similarly, lectins interfered with STAT5a activation promoted by the JM-a-like ErbB2 and ErbB3, but not with STAT5b activation promoted by the JM-b-like EGFR (original and new Suppl. Fig. 12). Lastly, lectins interfered with the survival of HC11 cells stimulated with ligands activating JM-a-like receptors, but not with the survival of cells stimulated with ligands activating JM-b like receptors (Rebuttal Fig. 4).

Rebuttal Figure 4. Sensitivity of survival mediated by JM-a- or JM-b-like receptors to treatment with lectins. HC11 cells endogenously expressing ErbB4 JM-a, PDGFRA, FGFR1/2, INSR and EGFR/Erbb2 were cultured for 24 hours in the presence of the indicated cognate ligands and/or the indicated lectins. Cell confluence was measured by Incucyte Live imaging. Statistical significance was assessed with one-way Brown-Forsythe and Welch ANOVA. Welch corrected two-tailed Student's T-test was used for post-hoc analyses. The resulting P-values were adjusted by the method of Benjamini, Krieger and Yekutieli.

To further experimentally address the Reviewer's comment about the significance of glycans in ErbB4 signaling, the effect of PNGaseF, an enzyme that specifically cleaves N-glycans, on the ErbB4-mediated STAT5 activation was tested (Rebuttal Fig. 5). Together, these data support the role of glycan binding in the eJM domain as a mechanism significantly regulating its unique signaling properties.

Rebuttal Figure 5. Role of N-glycans in ErbB4 isoform-specific STAT5 activation. COS-7 cells expressing ErbB4 JM-a or JM-b were incubated for 18 hours in medium supplemented with 5000 units of PNGaseF (New England's Biolabs, Catalog #P0704), an enzyme that specifically degrades N-linked oligosaccharides. The cells were stimulated with NRG-1 for 2 minutes prior to lysis. Activation of STAT5a or STAT5b was analyzed by western and the densitometry of the resulting bands were quantified with Image Studio Lite. The statistical significance was analyzed with a two-tailed Student's T-test. One dot corresponds to one replicate experiment. ns = non significant ($P > 0.05$).

Overall, I felt that the data did not strongly support the authors conclusions, especially for the leading Western blot experiments. It is possible that this would seem less problematic if the paper was organized in a different way. The difference in the ErbB4 isoforms is well established in the literature, so, perhaps leading with an expansion of the mass spectrometry analysis and co-localization experiments might allow a difference emphasis that would rely less heavily on the somewhat marginal difference between the Stat5a and Stat5b phosphorylation.

--- **The authors wish to thank the Reviewer for the constructive comments. To follow the recommendation, we have now completely reorganized the presentation. As proposed, the presentation is now led with an expansion of the mass spectrometry analysis and co-localization experiments. The mass spectrometry analysis has been carried out for three additional rounds resulting in a totally new dataset (visualized in new Fig. 1A). The co-localization of ErbB4 JM isoforms with each other, as well as of ErbB4 JM-a with $\beta 1$ integrin, have been analyzed with an additional technology, SIM (structured illumination microscopy) super-resolution imaging (new Suppl. Fig. 1 and 13) providing sufficient resolution to image also different cell surfaces in the 3D space.**

To improve statistical power of the analyses of STAT5a and STAT5b phosphorylation several new experiments have been carried out, and statistical analyses are now presented throughout the manuscript (new Fig. 2E,F-H; new Fig. 3C-D; new Fig. 4E; new Fig. 5B,D-F,I-J; new Fig. 6B; new Fig. 7C-D; new Suppl. Fig. 4A-B,D,F,H; new Suppl. Fig. 5C,E,G,I; new Suppl. Fig. 6B,D; new Suppl. Fig. 7D; new Suppl. Fig. 12A-C). Finally, leading Westerns in new Fig. 4D and new Suppl. Fig. 4E,G have been replaced by more illustrative ones. ---

Additional major points:

1. Much of the data in this manuscript relies on Western blot observations – all using overexpressed receptors.

--- As indicated in the authors' response above, the comparative analyses of the ErbB4 JM isoforms is unfortunately restricted to engineered set-ups, as the two isoforms are not naturally co-expressed in the same cell background. As the authors also acknowledge the limitations inherent to overexpression models, the experimentation was carried out in different cell backgrounds with different expression levels (please see *e.g.* new Fig. 3C,D; Rebuttal Fig. 1).

Unlike the two ErbB4 JM isoforms, other JM-a- or JM-b-like RTKs are, however, frequently co-expressed in the same cell. As an approach to further address Reviewer's comment we have now expanded the analysis of STAT5 subtype activation preference to cover more endogenously expressed JM-a- and JM-b-like RTKs (new Fig. 6A-B). In addition, new data is provided to illustrate further differences in how endogenous JM-a- and JM-b-like RTKs i) differ in their sensitivity lectin inhibition of ligand-stimulated growth (new Fig. 7E-F; Rebuttal Fig. 5 above), and ii) in colocalization with endogenous $\beta 1$ integrin (Rebuttal Fig. 6). These new data are consistent with our previous conclusions.

Rebuttal Figure 6. Colocalization of JM-a- and JM-b-like RTKs with $\beta 1$ integrin at the cell surface. COS-7 (A) and MCF-7 (B) cells endogenously expressing ErbB2, EGFR and AXL, or ErbB3, INSR and IGF1R, respectively, were imaged with STED super-resolution fluorescence microscopy. Colocalization of the indicated RTK and $\beta 1$ integrin was measured by the colocalization coefficient k_1 . Statistical significance was assessed with one-way Brown-Forsythe ANOVA and Welch corrected two-tailed Student's T-test was used for post-hoc analyses. The resulting P-values were corrected by the method of Benjamini, Krieger and Yekutieli.

The differences observed are not always clear, and at times there is inconsistency between the shown blot and the quantitation (see below). Combining results from multiple cell types – some with added NRG and some without – dilutes rather than enhances confidence. It is, for example, unclear whether there is any difference in levels

of pSTAT5a with the JM-a or JM-b isoform using HC11 cells since there is only one repeat with this cell shown.

--- To address Reviewer's comment concerning the HC11 data, we have now repeated the experiments with these cells (new Fig. 3C-D). As previously noted, the JM-a isoform consistently activated STAT5a more than the JM-b isoform. The authors would also like to point out that the quantifications as they were originally presented in the original Fig. 1C-D (new Suppl. Fig. 4A-B) only included the experiments that were carried out with the sole purpose of comparing the wild-type ErbB4 isoform-promoted STAT5 subtype activation, in the absence of other reagents or ErbB4 variants. However, the same basic finding has been reproduced as an internal control in a number of other experiments as well. We have now quantified all of these experiments and present them by cell line in the new Fig. 3C-D. In total, the experiment addressing differences in STAT5 activation has been repeated in each cell line at least four times.

Furthermore, we provide evidence in rebuttal Fig. 1 that the difference in the amount of nuclear STAT5 in cells expressing the different ErbB4 JM isoforms is recapitulated in all the four cell backgrounds – MDA-MB-468, COS-7, MCF-7, and HC11 – and not only in HC11 cells, as was presented in the original Fig. 1G-J (new Fig. 3G-J). ---

2. In Figure 2B there is clearly a much higher level of phosphorylation of STAT5b for the JM-b/a chimera than for the JM-b isoform. This is stated in the text. Yet the quantitation in panel C shows no difference for these two proteins. Since the variation between the two repeats (based on number of displayed points) is extremely small, it is not clear how the data in Figure 2B could lead to the data in Figure 2C (lower panel). The number of biological repeats need to be stated for every experiment in the manuscript, and statistical significance in the differences from these Western blot data reported.

--- The authors agree that the quantification shown in the lower panel of original Fig. 2C appears not to match the blot shown in the original Fig. 2B (the panels are now included as new Suppl. Fig. 7C-D). The fact that the lane 4 of the pSTAT5b blot including the JM-b/a sample includes an artefact on the right edge produces a signal that is not derived from the actual pSTAT5b as observed in a short exposure of the same blot (Rebuttal Fig. 7).

Rebuttal Figure 7. Densitometric quantification of the pSTAT5b blot of new Suppl. Fig. 7C. Please note the black artefact on the right of the pSTAT5b band on lane 4.

On p. 5 of the original submission we stated that "... while wild-type JM-b and chimeric JM-b/a induced the activating phosphorylation of STAT5b

to an approximately similar extent ...” which was actually consistent with the original quantification. However, as the data has been replaced into the Supplement, this interpretation has also been omitted from the text.

As the authors agree that the small number of experiments was a limitation for analyzing data shown in the original Fig. 2B-C, the experiment was repeated, and additional datapoints, as well as statistical analysis, added to the new Suppl. Fig. 7D. Each dot in the plots represent independent biological repeats throughout the manuscript, which is now also indicated in the Materials and Methods.

3. The structural modeling adds little to the paper and is misleading. The concatenation of coordinates from 2LCX onto 3U7U is arbitrary. There is no overlap between the two structures – indeed there are two missing amino acids. Only one of the molecules in 3U7U is modeled to amino acid 639, with significantly less of the eJM region modeled for the other molecules. 2LCX starts at 642 but there are very limited restraints on mainchain conformation until the start of the TM helix. The structural model in Figure 3K should be removed – it is clear that 631 and 632 must be close to the membrane. Additional details implied by this figure are not useful and potentially highly misleading.

--- We have used different modeling tools such as Modeller and ITASSER to model the extracellular domain with extending transmembrane domain in its entirety, but in practice there are difficulties in creating a dimer as those constraints on the complexes seen in the X-ray and NMR structure are lost. The result is a set of highly variable monomers for ectodomain-TM regions.

The obvious approach is to build an initial structure from the two dimeric structures. The ectodomain (residues 25-639) was joined to the best representative conformer of the ensemble (Model 1) of the NMR models for the TM domains (structure 2LCX, residues 642-685), and the two missing residues (640-641) were added. The C-terminal ends of the ErbB4 ectodomain structure would be flexible, residues 634-650, but the ectodomain dimer interactions, the disulfide linked arms, and the TM regions will function to limit the flexibility to some degree; and MDS will allow exploration of the range of those movements.

The resulting ensemble differs from the starting structure along this flexible region with a range of expected motions given the restraints placed on them in the TM-tethered dimer model. The models we have made, with and without the TM region as a C-terminal constraint on the ectodomain, have indeed proved useful. They show that for the region bounded by the two disulfide bonds – experimentally shown to be important for selective STAT5 activation – that it is unlikely that conformational changes in this region of structure are involved in the process. ---

4. How was the analysis to obtain the sequence motifs conducted? The motif is very general and should have some statistical significance associated with it – does this occur more frequently in the eJM regions of RTKs compared to similar regions of other single TM domain proteins?

--- Different motif models starting from the JM-a and JM-b sequences of ErbB4 were fitted into the data and the one able to categorize all the experimentally verified receptors was chosen (please see Rebuttal Fig. 8).

The method is now described in the Materials and Methods in more detail.

Model	Number of accurately categorized	% of accurately categorized	BH corrected P-value*
GPTSHDC; GSSIEDC	6	75	0.09
GPTSHD; GSSIED	4	50	0.58
PTSHDC; SSIEDC	5	62.5	0.29
PTSHD; SSIED	6	75	0.09
TSHDC; SIEDC	3	37.5	0.85
TSHD; SIED	1	12.5	0.96
SHDC; IEDC	4	50	0.58
SHD; IED	4	50	0.58
HDC; EDC	5	62.5	0.29
HD; ED	3	37.5	0.85
GXXXHDC; GXXXEDC	6	75	0.09
GXXXHD; GXXXED	5	62.5	0.29
PXXHDC; SXXEDC	5	62.5	0.29
PXXHD; SXXED	5	62.5	0.29
PXXHD/PXXDH; SXXED	5	62.5	0.29
GXXXHD/GXXXDH; GXXXED	3	37.5	0.85
GXXXHDC/GXXXDHC; GXXXEDC	4	50	0.58
G[X] _{1,2,3} HDC; G[X] _{1,2,3} EDC	7	87.5	0.02
G[X] _{1,2,3} HD; G[X] _{1,2,3} ED	6	75	0.09
P[X] _{1,2} HDC; S[X] _{1,2} EDC	5	62.5	0.29
P[X] _{1,2} HD; S[X] _{1,2} ED	5	62.5	0.29
G[X] _{1,2,3} [H,N,R,K,Q,N][D,E,N,Q,H]; G[X] _{1,2,3} [E,D] _{1,2}	6	75	0.09
G[X] _{1,2,3} [H,N,R,K,Q,N][D,E,N,Q,H]/G[X] _{1,2,3} [D,E,N,Q,H][H,N,R,K,Q,N]; G[X] _{1,2,3} [E,D] _{1,2}	7	87.5	0.02
G[X] _{1,2,3} [H,N,R,K,Q,N][D,E,N,Q,H]/G[X] _{1,2,3} [D,E,N,Q,H][H,N,R,K,Q,N]; G[X] _{1,2} (H,N,R,Q,K)[E,D]	7	87.5	0.02
G[X] _{1,2,3} [H,N,R,K,Q,N][D,E,N,Q,H]/G[X] _{1,2,3} [D,E,N,Q,H][H,N,R,K,Q,N]; G[X] _{1,2} (H,N,R,Q,K)[E,D,Q,N,H](H,N,R,Q,K)	8	100	0.00

*Estimated from the binomial cumulative probability density function with probability of 0.5 for being accurately categorized. BH=Benjamini Hochberg

Rebuttal Figure 8. Sequence motif models fitted to experimental data to derive the final sequence motif model.

The probability of finding a JM-a motif in exactly 22 eJM domains (30 amino acids from the transmembrane domain) of all the tested 53 human RTKs was 0.017, when estimated with a binomial cumulative probability distribution. The parameter p of the binomial distribution was estimated from the occurrence of sequence areas fitting the JM-a motif in 30 residue sequence windows in the extracellular and intracellular domains of all the 53 RTKs. The probability of discovering a JM-b motif in turn in 14 RTK eJM domains was 0.97 indicating that the JM-b motif is not enriched in the RTK eJM regions, but a similar sequence area can be found elsewhere in RTK sequences with equal probability. The lack of enrichment is consistent with the experimental findings presented in the original Fig. 3J (new Fig. 5J) that the JM-b motif *per se* is not functional. The interspecies conservation of the JM-a motif was also estimated. The JM-a motif was conserved in an average of $84.4 \pm 13\%$ of the 16 tested taxonomic classes of species ($P = 1.11e-16$).

The occurrence of the JM-a motif in other single pass transmembrane proteins was also explored. The occurrence of the JM-a motif in 80 cytokine receptors of the type I and type II cytokine receptor, immunoglobulin, tumor necrosis factor receptor and TGF-beta receptor families was evaluated. The JM-a motif could be found in a total 20 out of 80 eJM domains, which was not statistically significant ($P = 0.14$). The finding suggests that the JM-a motif is especially enriched in the eJM domains of RTKs. The enrichment of the JM-a motif was additionally assessed by individual cytokine receptor family. Only within the TGF-beta receptor family of cytokine receptors, the JM-a motif was significantly enriched among the eJM domains ($P = 0.025$). ---

5. In Figure 3, receptor expression levels vary between different RTKs. It is unclear

whether this is taken into account in comparing STAT5 phosphorylation levels. The quantitation from the Western blots is poorly described. Normalization procedures are not consistent across all experiments, and some data are background corrected.

--- The authors agree that the expression of the different receptors is not perfectly equal in spite of expressing the receptors in the same cell background, under the same promoter, and preparing the samples simultaneously. The differences may be an outcome of natural differences in the turnover of the proteins. We are however confident that the observed differences in the expression levels do not play a major role since either the phosphorylation of STAT5a or STAT5b did not, as a general observation, reflect the expression levels of the different receptors. The new Fig. 5A and C (original Fig. 3C and E) provide an example: In the Western analyses ErbB3 is clearly expressed at a lower level than EGFR, and ErbB2 is expressed at the highest level. However, both ErbB2 and ErbB3 similarly promote STAT5a phosphorylation (new Fig. 5A) and suppress STAT5b phosphorylation (new Fig. 5C) as compared to EGFR.

The quantitation from the Western blots is now described in more detail in the Materials and Methods. Different normalization procedures were mainly chosen for visualization purposes and do not skew or change the results. All data for which an estimate of the background level was acquired were background-corrected. Experimental data for which an estimate of the background signal was not acquired, were not background-corrected. ---

6. The number of cells analyzed in the STED microscopy experiments should be noted.

--- The number of cells analyzed in the STED microscopy experiments was noted in the original Materials and Methods section of the manuscript on p. 32. To better visualize the number of cells analyzed, dots reflecting individual cells have been included in the box plots shown in new Fig. 1C, new Fig. 3 G-H, new Suppl. Fig. 2B, new Suppl. Fig. 5K,M, and new Suppl. Fig. 8B. ---

Minor points

1. Numbering for the JM-b consensus motif is wrong in text and figure – it starts at 627 (as for JM-a). Sequence numbers should all be verified, and a note added to related the sequence numbering here to those used in the pdb files..

--- The authors wish to thank the Review for pointing the mistake out. This has now been corrected to the new Fig. 5H. All other sequence numbers in the new Fig. 5G and H have also been verified. ---

2. There is inconsistency in whether or not NRG stimulation used – for example text states that NRG not used for MDA-MB-468 cells due to high basal ErbB4 activation, but Figure 1K shows these cells stimulated with NRG.

--- The authors agree that the referred text may not have been unequivocally formulated. The original manuscript indeed included the sentences: “To examine the potential of the ErbB4 JM isoforms in promoting activation of endogenous STAT5a or STAT5b in a shared cellular background, ErbB4 JM isoforms were overexpressed in MDA-MB-468, HC11, MCF-7 or COS-7 cells. To minimize the background effect from endogenous ErbB receptors other than ErbB4, ligand stimulation was not used in experiments with MDA-MB-

468 or HC11 cells that demonstrated basal ligand-independent ErbB4 activation.“ What the authors intended to communicate with this was that in the MDA-MB-468 and HC11 cells, NRG stimulation promoted background phosphorylation of STAT5 that was independent of the expression of ErbB4 in these cells, and thus probably an outcome of the presence of other NRG receptors in the cells. When expressed in these cells, ErbB4 was constitutively active and the activity was sufficient to promote STAT5 phosphorylation. The sentence on p. 7 of the manuscript has now been reformulated to: *“To minimize the background effect from endogenous ErbB receptors, ligand stimulation with neuregulin-1 (NRG-1) was only used in experiments with MCF-7 or COS-7 cells that did not demonstrate ErbB4-mediated STAT5 phosphorylation without exogenous ligand stimulation.”*. However, in the case of STAT3, there was no background effect of NRG stimulation on STAT3 phosphorylation – possibly as the endogenous ErbB receptors failed to couple to STAT3 – and even in the case of overexpressed ErbB4, the basal ErbB4 activity (again clearly seen in new Suppl. Fig. 4C) was not sufficient to induce STAT3 phosphorylation. Thus NRG stimulation was used for the MDA-MB-468 cells when STAT3 (but not when STAT5) phosphorylation was studied (new Suppl. Fig. 4C-D). A sentence *“Interestingly, NRG-1 stimulation was required in MDA-MB-468 cells for the activation of STAT3.”* specifically indicating the need for NRG stimulation for the STAT3 experiment has now been added in the Results on p. 7. ---

Reviewer #2 (Remarks to the Author):

Vaparanta et al have investigated the underlying mechanism for selective signaling of two naturally occurring splice variants of ErbB4 RTK that differ in their membrane proximal extracellular juxtamembrane (eJM) domain. The JM-a isoform activated STAT5a and JM-b STAT5b and STAT3 and eJM cleavage was not determining selectivity of signaling. Surprisingly, expression of TYK2 but not other JAKs was involved in ErbB4 JM-b mediated STAT5b activation while the JM-a mediated STAT5a activation is not dependent on any JAK kinase. Kinase activity of TYK2 was not required for STAT5b activation suggesting a scaffolding role for the protein.

STAT5a activation by JM-a was found to rely on structural characteristics of eJM and particularly H631 and D632 residues were found to be critical possibly via hydrogen bond formation that would enable selective coupling with extracellular molecules. In JM-b the 631-632 residues did not affect signaling. Glycosylation of ErbB4 was not affecting the signalling characteristics and both variants bound equally to lipids but JM-a was found to bind selectively complex N-glycans. B1 integrin was identified as a selective ErbB4 eJM domain binder suggesting that N-glycans mediate interaction between ErbB4 eJM and B1 integrin that target ErbB4 to specific cell surface compartment for STAT5a selective activation.

This is a well-written profound characterization of two ErbB4 variants that identified novel mechanisms for selective signaling and STAT5 activation. Extensive methodology has been exploited and the results seem reliable. Some Westerns are a bit hazy but the quantifications seem right. Molecular dynamic simulations and structural analysis seem reliable.

--- The authors wish to thank the Reviewer for the very positive and encouraging comments.

We have now repeated several of the Western analyses both to be able to present blots with better quality (e.g. please see new Fig. 4D; new Suppl. Fig. 4E,G), as well as to improve the statistical power of quantification of the data. ---

Specific comments

Characterizing the eJM sequence motif as “consensus” in Fig.3A may not be exactly right term as quite many RTKs do not seem to follow that. More extensive analysis of glycan-lectin sensitivity of RTKs could strengthen the concept.

--- The authors agree that the word *consensus* might be misleading. We have now chosen to use "*RTK eJM sequence motif*" instead of "*consensus sequence motif*" throughout the manuscript. However, as mentioned above in our response to Reviewer #1, the motif region is indeed significantly enriched in the eJM region of all human RTKs with a $P = 0.017$.

As suggested by the Reviewer, we have now expanded our analysis of the lectin sensitivity of endogenously expressed RTKs by addressing the effects of lectins on RTK ligand-stimulated cell survival (please see Rebuttal Fig. 4 above). In addition, we have studied the effect of PNGaseF, an enzyme that specifically cleaves N-glycans, on the ErbB4 JM isoform-mediated STAT5 activation (please see Rebuttal Fig. 5 above). Findings of these new experiments support a role for the JM-a-like eJM domain in interacting with N-glycans, in accordance with the original observations. ---

Comparison of phosphoproteomes in endogenously ErbB4 JM-a and JM-b expressing cells after ligand stimulation would be interesting to obtain a more objective insight of signalling landscape, obviously the cell type differences will cause some variation.

--- The authors agree that the experiment would provide interesting additional results. ---

Reviewer #3 (Remarks to the Author):

ErbB4 is the ErbB4 family member whose signaling activities/functions are perhaps least well characterized. The authors here characterize the differential signaling by two ErbB4 splice variants designated JM-a and JM-b, whose sequences differ only in the extracellular juxtamembrane domain. The primary observation is that the JM-a and JM-b isoforms preferentially activate the downstream signal transducers Stat5a and Stat5b, respectively. The authors rule out various mechanisms that might explain the differential signaling of JM-a and JM-b (such as differing susceptibility to extracellular proteases), and ultimately make conclusions regarding their differential signaling, including that the isoforms might traffic differentially to membrane subdomains (caveolae), that only JM-b appears dependent upon the Tyk2 kinase, that the JM-a and JM-b sequences show differential interactions with glycan among a commercial array of glycan substrates, and that JM-b appears to interact with a specific N-glycan-linked molecule present in the cell lines examined (specifically integrin-beta-1). It is suggested that a JM-b/integrin-beta-1 interaction is somehow involved in JM-b-specific activation of Stat5b.

The authors present a large body of data, generated with a wide-ranging combination of methods, in their attempt to elucidate the mechanism(s) of ErbB4 signaling to Stat5a and Stat5b. However, while the essential conclusions of the manuscript might be valid, the manuscript stops short of elucidating the full mechanism by which Stat5b is activated by JM-b, and in many cases presents ancillary observations of unclear significance to the activation mechanism. It appears yet more uncertain how Stat5a is activated by JM-a, although the mechanism is apparently different in ways from that of JM-b/Stat5b. It seems then that the manuscript is somewhat unfocused and attempts to present too many observations, versus solidly nailing down a specific mechanism. Although the proposed mechanism is an interesting one and apparently might extend to a number of different cell surface receptors, it might be premature to publish it in this form.

--- We want to thank the reviewer for the insightful comments. As also suggested by the Reviewer #1 we have now fully reorganized and rewritten the manuscript to bring forth more focus on the mechanism of differential signaling of ErbB4 JM-a and JM-b. To better demonstrate the proposed mechanisms of JM-a-mediated STAT5a and JM-b-mediated STAT5b activation, a schematic illustration is now provided below (please see Rebuttal Fig. 9). In short, the JM-a motif in the eJM region of JM-a-like receptors associate with the complex N-glycans of cell surface proteins such as β 1 integrin. Due to this association, the JM-a receptors are differentially localized to different membrane subcompartments than the JM-b receptors where they activate STAT5a. The JM-b-like receptors in turn associate with TYK2 which is needed in a scaffolding role in the activation of STAT5b. Our data suggest the JM-b receptors do not interact with the complex N-glycans or β 1 integrin, because of a lack of the JM-a motif in the eJM region. To achieve a more comprehensive presentation of our observations, the different mechanism underlying the STAT signaling promoted by the two ErbB4 JM isoforms are now more thoroughly described in the Discussion.

Rebuttal Figure 9. Schematic illustration of the proposed mechanisms of STAT5a or STAT5b activation by receptor tyrosine kinases (RTKs) with JM-a- or JM-b-like extracellular juxtamembrane domains.

To further improve the mechanistic understanding, new experimental data have been included demonstrating i) the interaction and colocalization of $\beta 1$ integrin with ErbB4 JM-a and not with ErbB4 JM-b (new Suppl. Fig. 13), ii) the sensitivity of ErbB4 JM-a-mediated STAT5a activation, but not JM-b-mediated STAT5b activation to an N-glycan cleaving enzyme (Rebuttal Fig. 5 above), and iii) the sensitivity of JM-a-like receptor-mediated, but not JM-b-like receptor-mediated, cell survival on lectins (Rebuttal Fig. 4 above).

The authors agree that some of the data in the original manuscript may have seemed ancillary to the main focus. Thus, we have now omitted data describing the differential localization of STAT5b in caveolae in MDA-MB-468 cells overexpressing ErbB4 JM-isoforms (original Suppl. Fig. 1E-G) and the data on the effect of the Y984F/Y974F mutation on the ErbB4 mediated STAT5 activation (original Suppl. Fig. 1A-D). However, we have chosen to retain the negative data describing the roles of JAK kinases, proteolytic cleavage, glycosylation as well as lipid interactions in JM-a-mediated STAT5a activation, as we feel that the observations provide additional value to the manuscript. All these processes have previously been demonstrated to play significant roles in the signaling of ErbB4 JM-a or other JM-a-like RTKs in other cellular contexts. However, to improve the focus of the presentation, these negative data are now mostly shown in the Supplement (new Suppl. Fig. 3; new Suppl. Fig. 5; new Suppl. Fig. 6; new Suppl. Fig. 10). ---

There are also concerns with the quality of some of the experimental data. Various Western blots do not seem to show very definitive bands, although quantification of replicates blots was apparently done to demonstrate significance. For example, the blots of Sup. Fig. 1A-D, which aims to demonstrate a differential

sensitivity to ErbB4 phosphorylation site mutation, do not seem to show a convincing effect.

--- The authors agree that the Western blots in the original Suppl. Fig. 1A-D were not technically the most successful. These have now been replaced by new representative blots, and the number of quantified repetitions increased by omitting the data on the Y984F/Y974F variant (new Suppl. Fig. 4E-H). A statistical significance was reached for a role of JM-b Y1012F in STAT5b activation ($P = 0.005$).

Other Western blots have also been replaced by more representative ones, a number of biological repetitions have been carried out, and quantification of independent Western analyses is now extensively provided throughout the revised manuscript. More specifically, we have now improved the quality of the immunoblots by:

- i) repeating experiments
 - new Fig. 3C-D; new Fig. 4D-E; new Fig. 5B,D; new Suppl. Fig. 4A-B,D,F,H; new Suppl. Fig. 7D; new Suppl. Fig. 12C
- ii) analyzing the statistical significance of the immunoblot quantifications
 - new Fig. 2E,F-H; new Fig. 3C-D; new Fig. 4E; new Fig. 5B,D-F,I-J; new Fig. 6B; new Fig. 7C-D; new Suppl. Fig. 4A-B,D,F,H; new Suppl. Fig. 5C,E,G,I; new Suppl. Fig. 6B,D; new Suppl. Fig. 7D; new Suppl. Fig. 12A-C
- iii) replacing representative immunoblots
 - new Fig. 4D; new Suppl. Fig. 4E,G
- iv) clarifying the immunoblot quantification process at points where accuracy of the quantification values were questioned
 - Rebuttal Fig. 7; Rebuttal Fig. 12; Rebuttal Fig. 13; please find rebuttal figures below in "Response to Reviewers".
- v) by performing totally new experiments
 - new Fig. 6A-B; Rebuttal Fig. 5 ---

Also, the conclusions about Stat5 translocation to caveolin-rich domains (Sup. Figure 1E-G) seem questionable in that "translocation" (movement) is not really analyzed and the entire gradient was not analyzed to see exactly how Stat5b and ErbB4 were distributed among the various fractions.

--- The authors agree that the term "*translocation*" was not used accurately in this context, when rather "*presence*" could have been a more accurate expression. The STAT5b and ErbB4 location was indeed analyzed in the whole gradient, but since a difference was observed only in the caveolin-1-rich fraction, we chose to only visualize it in the manuscript. We have now opted to remove this ancillary data (original Suppl. Fig. 1E-G) from the manuscript to improve the focus. ---

Another example is the reciprocal immunoprecipitation used to show ErbB4 integrin-beta-1 association (Fig. 7I), in which at least one of the two IPs does not show a conclusive interaction.

--- The authors agree that the technical quality of the blots showing HA immunoprecipitation (original Fig. 7I) was moderate and have chosen to omit it from the new Fig. 7H. To compensate for this, we now provide evidence that $\beta 1$ integrin is precipitated by anti-ErbB4 using targeted mass spectrometry. These new data confirming that $\beta 1$ integrin associates

preferably with ErbB4 JM-a as opposed to JM-b are now shown as new Suppl. Fig. 13A-C. In addition, new data confirming the colocalization of β 1 integrin with ErbB4 JM-a is shown based on SIM super-resolution microscopy (new Suppl. Fig. 13D-H). ---

A minor point is the ErbB4 variants CYT-1 and CYT-2 are presented, but the reader will likely not know the meaning of these terms and the purpose for which they are used is unclear.

--- The authors wish to thank the Reviewer for bringing the matter to our attention. The sentence "*While experimentation was routinely carried out in the context of the CYT-2 cytoplasmic domain of ErbB4 that has previously been shown to be more stable as a result of impaired interaction with ubiquitin ligases³⁸, the effect of the alternative cytoplasmic domain CYT-1 on STAT5 activation was also controlled. However, the difference in the potential of the ErbB4 JM isoforms to activate different STAT5 isoforms was independent of their coupling to different CYT variants (Supplementary Fig. 4A-B)*" has now been added on p. 7 of the Results. These data have also been moved to the Supplement (new Suppl. Fig. 4A-B). ---

Reviewer #4 (Remarks to the Author):

The manuscript by Varapanta et al explores the significance of differences in the extracellular juxtamembrane (eJM) domain of the natural juxtamembrane isoforms of ErbB4. This is an interesting and worthwhile subject of research, not just in the ErbB4 field, but also as part of the community endeavours to understand structure-function relationships in receptor tyrosine kinases (RTKs).

ErbB4 is an ideal target for these studies because it is the only member of the EGFR family that has naturally evolved into two eJM isoforms. It is also known that there is differential activation of STAT proteins not just by the two ErbB4 isoforms, but also in several other RTKs.

Compelling results are included in this manuscript. For example, the link between the JM-a isoform and the recruitment and activation of STATa and JM-b and STATb and STAT3, the importance of residue H631 for selective STATa activation by JM-a, and the consensus motif in the eJM.

--- The authors wish to thank the Reviewer for the insightful and positive comments! ---

The molecular dynamics simulations don't appear to show a big difference between the two isoforms. It would have been reasonable to expect (and one imagine the authors may have thought the same) that a change of JM structure might explain at least some of the differences in isoform function. Indeed, that change would have been revealing in explaining how extracellular interactions could initiate and propagate diverging signalling events across the membrane. However, as recorded in the literature, and also in my own experience, the absence of differences in MD simulation results in solvated protein fragments often don't correlate with the findings if the membrane bilayer is present (this would be potentially more acute for the JM segment). Restraints from other protein domains may also matter. Without both of this, it is hard, in my opinion, to draw many conclusions from the MD results.

--- We have now performed the MD simulation by including the transmembrane domain in a lipid bilayer (as well as without) and the results are very similar (new Fig. 4B-C; new Suppl. Fig. 7E). Both analyses show a similar conformational landscape for the two isoforms, and we have now examined the differential interactions possible for the key residues H631/E631. ---

My main concern is with the STED experiments and the conclusions drawn regarding the localisation of RTKs into different plasma membrane microdomains as the means to controls a subtype specific STAT activation mechanism. The STED experiments are not at all well described. I could not find a description of the type of STED deployed here: Is it 2D or 3D? What is the resolution and/or the NA of the objective? There is no scale bar in the figures.

--- We thank the reviewer for the well-justified comments. We have now amended the description of the STED experiments in the Materials and Methods section. The STED microscopy deployed for the images was in 2D. The resolution achieved in the X-Y axis, in which orientation the images were acquired, was around 50 nm. The numerical aperture of the 100x Olympus UPLSAPO objective was 1.4. Scale bars have been added to the figures showing microscopic images. ---

The STED data show what it seems to be to be zoom in images from what it appears to be membrane cross-sections. To have a zoom out image from where these regions have been selected would be quite helpful.

If the orientation of the plasma membrane in the images is as it seems, I am unsure of how these experiments can be robustly interpret.

If one is looking for domains on the plasma membrane, I would assume one needs to look at the plasma membrane head on, i.e. to image the x,y in-plane of the membrane. If you look at the plasma membrane sideways (x,z or y,z direction), i.e. like in an equatorial section of a confocal microscope, which is what it looks like to me in the images provided, then if STED is performed in 2D one would see a >500 nm projection (i.e. the membrane in cross-section). In 3D one would still be projecting across >80 nm depth. I fail to see how one would draw robust conclusions about protein colocalisation on the x,y plasma membrane plane if one is looking at x,z or y,z projections.

If, as it looks to me from the images, this is the case, why has such an orientation been chosen? The deleterious risks of doing so would need to be explained or rectified by recording x,y areas. It is difficult to ascertain what has been done. All this needs to be corrected, explained, or both.

--- The original STED images were acquired from the peripheral membrane in the X-Y axis, as the ErbB4 signal predominantly concentrates in this projection. To better illustrate the different projections and different cell surface compartments, we have now carried out SIM super-resolution imaging on the ErbB4 JM-a and ErbB4 JM-b colocalization separately in the bottom, peripheral and top cell surfaces in all X-Y, X-Z, and Y-Z projections with zoom-ins and zoom-outs (please see new Suppl. Fig. 1). The figure is accompanied with a schematic describing the orientation of the image acquirement (new Suppl. Fig. 1A). The results from the SIM super-resolution imaging indicate that the ErbB4 JM-a colocalizes less with ErbB4 JM-b than with a differently tagged ErbB4 JM-a in both the peripheral and bottom cell surfaces in all X-Y, X-Z and Y-Z projections.

New SIM analyses were further used to demonstrate co-localization of ErbB4 JM-a and β 1 integrin in all projections (new Suppl. Fig. 13D-H). -

--

There also appears to be chromatic aberration in some STED images. If so has this been considered? This will change the results.

--- The chromatic aberration of the channels had been checked and corrected before each imaging session by the staff on the Turku Bioscience Cell Imaging and Cytometry Core. Although certainty on absent misalignment of the channels can never be claimed, the authors are confident that the observed differences in the location are not due to misaligned lasers. The new consistent findings obtained with another super-resolution imaging technique (SIM; new Suppl. Fig. 1) warrant further confidence to the observations. ---

Other points.

The means of analysis could also be considered, for example, whether the presence of clusters may affect the means to analyse colocalisation using Pearson?.

--- The authors agree with the Reviewer that relying on only one measure of

colocalization may bias the results. The colocalization in the manuscript was indeed assessed with 4 generally accepted measures of colocalization (Pearson correlation, Manders overlap, M1/M2, and k1/k2) resulting in identical results, while only Pearson correlation was visualized in the manuscript. This was mentioned in the original Materials and Methods on p. 32. These analyses of STED microscopy data shown in the original Fig. 5B-C (new Fig. 1B-C) are now shown below for reviewing purposes (Rebuttal Fig. 10).

Rebuttal Figure 10. Colocalization analyses of STED microscopy data shown in new Fig. 1B. Four different colocalization coefficients were calculated to estimate the level of colocalization in the same data. Statistical significance was assessed with Kruskal-Wallis non-parametric ANOVA and Mann-Whitney U test was used for post-hoc analyses. The resulting P-values were corrected with the method of Benjamini, Krieger and Yekutieli.

The rationale of the choice of the four cell lines could be better explained; part of it becomes clear throughout the manuscript but it is on occasions confusing.

--- We have now modified the sentence "As the natural *ErbB4* JM isoforms are not endogenously expressed in the same cell types ", the observation was validated by overexpressing *ErbB4* JM-a or JM-b in four different cell backgrounds – MDA-MB-468, HC11, MCF-7, or COS-7 cells – to ensure reproducibility of the findings." on p. 6-7 of the Results to better explain the choice for the different cell lines. Please see also new Fig. 3C-D and Rebuttal Fig. 1 provided in our response for Reviewer #1 above for the reproducibility of the key finding of STAT5 subtype activation in the different cell backgrounds. ---

In Fig. 1B, the two spliced isoforms of *ErbB4* are tested. This could be mentioned in the introduction to help readers outside the EGFR field.

--- We thank the Reviewer for pointing this out. As explained above for Reviewer #3 as well, we have now added a sentence "While experimentation was routinely carried out in the context of the CYT-2 cytoplasmic domain of *ErbB4* that has previously been shown to be more stable as a result of impaired interaction with ubiquitin ligases³⁸, the effect of the alternative cytoplasmic domain CYT-1 on STAT5 activation was also controlled. However, the

difference in the potential of the ErbB4 JM isoforms to activate different STAT5 isoforms was independent of their coupling to different CYT variants (Supplementary Fig. 4A-B)" on p. 7 of the Results. The data involving the spliced isoforms have now been moved to the Supplement (new Suppl. Fig. 4A-B). ---

End of second paragraph, page 5, I would replace 'structure' with 'primary structure'.
 --- The text has been fully reorganized and edited. ---

When overexpression is mentioned, it would be helpful to get an idea of what this is meant to be and whether it is reproducible.

--- When *overexpression* is mentioned, the receptor is expressed from an exogenous plasmid construct that has been either transfected or virally transduced into the cells. The expression level is reproducible as long as the same plasmids and same transfection or transduction protocol is used. How the expression levels refer to the endogenous levels is dependent on the cell line and the used plasmids. To further contextualize the term *overexpression* and how it relates to the endogenous levels of ErbB4 in the cell lines, we quantified the expression of overexpressed ErbB4 and endogenous ErbB4 in different cell lines from Western analyses. COS-7 cells do not express any endogenous ErbB4 and were thus omitted from the analysis. The overexpression led to a median 5-8-fold increase in ErbB4 expression in MCF-7, HC11, and MDA-MB-468 cell lines (please see Rebuttal Fig. 11).

Exogenous ErbB4 / Endogenous ErbB4

Rebuttal Figure 11. Expression of exogenous versus endogenous ErbB4. Expression was evaluated by densitometric analyses of Western data from the three indicated cell lines.

Typos: In Methods substitute Abbrerrior for Abberrior (twice).

--- The authors wish to thank the Reviewer for locating the mistake. This has been corrected. ---

Reviewer #5 (Remarks to the Author):

The manuscript by Vaparanta et al. describes the discovery of a sequence motif in the extracellular juxtamembrane regions of ErbB4 receptor isoforms JM-a and JM-b, which is responsible for selective activation of STAT5a or STAT5b and subsequent differential cellular effects. The authors show that this feature is not restricted to ErbB4 isoforms but its present in many RTKs. The study suggests that the JM-a and JM-b receptors associate with different signaling complexes at distinct compartments at the cellular surface. JM-a preferentially interacts with beta1-integrin via N-glycans, while Tyk2 association with JM-b is necessary for the activation of STAT5b. It is potentially very interesting and relevant discovery. The authors have done massive amount of work involving numerous experiments and advanced technologies. However, the technical quality of many of the experiments is unfortunately quite poor, questioning the validity of findings and claims.

--- We thank the reviewer for carefully considered and encouraging comments. The quality of many of the experiments has now been improved by repeating the assays and by adding more power to the statistical analyses, as explained below in more detail. ---

Specific points:

1 > Virtually all claims and conclusions in the manuscript are based, and rely entirely, on accurate quantitation of small differences among various experimental conditions. And in many cases, the differences do not appear dramatic. Statistical significance tests should be performed and shown for all quantitative results (all boxplot panels in all figures).

--- We have now added statistical analysis for all Western quantifications into the manuscript and indicated statistically significant changes in the boxplots (new Fig. 2E,F-H; new Fig. 3C-D; new Fig. 4E; new Fig. 5B,D-F,I-J; new Fig. 6B; new Fig. 7C-D; new Suppl. Fig. 4A-B,D,F,H; new Suppl. Fig. 5C,E,G,I; new Suppl. Fig. 6B,D; new Suppl. Fig. 7D; new Suppl. Fig. 12A-C), as recommended by the Reviewer. ---

2 > The quality of some Western blots (WBs) used for quantitation is very poor (e.g. Fig. 2B, Fig. 3L, Supplementary Fig. 1A and 1C, Supplementary Fig. 4A).

--- As explained above in our comments to Reviewer #1 and #3, key Western analyses with moderate quality have been replaced and several biological repeats carried out to increase the statistical power.

The original Fig. 2B has been moved into the Supplement (new Suppl. Fig. 7C) and is now accompanied by a quantitation of more repeats (new Suppl. Fig. 7D). Please see our discussion about the STAT5b blot above in response to Reviewer #1, and the short exposure shown in Rebuttal Fig. 7.

The ErbB4 panel in original Fig. 3L (new Fig. 6A) has been replaced. In addition the analysis of the effect of endogenously expressed RTKs on STAT5 activation has been expanded by data on more RTKs analyzed by Western and statistics of densitometric quantification (new Fig. 6A-B).

As discussed above in response to a comment by Reviewer #3, the blots in original Suppl. Fig. 1A and C (new Suppl. Fig. 4E and G) have been

replaced by new representative blots, and the number of quantified repetitions increased by omitting the data on the Y984F/Y974F variant (new Suppl. Fig. 4E-H).

Data shown in original Suppl. Fig. 4A (new Suppl. Fig. 9A) has been improved by statistical analysis of the densitometric data shown in the new Fig. 5E. Please also see our response to the comment on the quantification of original Suppl. Fig. 4A below. ---

And in some cases, it seems to be discrepancies between the WBs and the quantitation derived from it. For example, on the WB in Fig. 3L phosphoSTAT5b band seems ~ 2x stronger in the NRG-1 treatment compared to control, while the quantitation of it on Fig. 3M shows reduction.

--- The method to densitometrically quantify the Western analyses undertaken in the manuscript systematically took into account the background signal in the same sample and Western lane. The acquired pSTAT5 signal value was additionally systematically normalized by the amount of total STAT5 in the same analysis. To visualize this in the context of the example brought up by the Reviewer, images of the original quantifications of the phospho-STAT5b and total STAT5b signals from the original Fig. 3L are shown below (Rebuttal Fig. 12). The densitometric analyses demonstrate no increase in the relative phospho-STAT5b signal when signals derived from total STAT5 are taken into account. Please also note the amount of signal in the positive control EGF lane as a positive reference. The densitometric analysis is consistent with the data shown in original Fig. 3M, as well as new Fig. 6B for which all the densitometric analyses were carefully repeated to control the data produced for the first submission of the manuscript.

The process of densitometric quantification of the Western data with ImageStudio Lite is now better described in the Materials and Methods on p. 26.

Rebuttal Figure 12. Densitometric quantification of western analysis shown in Fig. 6A Image Studio Lite was used to quantify the western signal from Odyssey CLx Imager (LI-COR).

Similar in Suppl. Fig. 1C and 1D, phosphoSTAT5b band on the WB is the weakest in lane 1, while the quantitation is telling us that it's the strongest among all.

--- The appearance of discrepancy was probably an outcome of high background around the lane 3 of the phospho-STAT5b blot shown in the original Suppl. Fig. 1C. This was evident also from the original densitometric analysis with the Image Studio Lite software that indicated a value of 136 for the background on lane 3, 88 for lane 2, and 37 for lane 1. After background correction, which is automatically performed in the Image Studio Lite software, the lane 1 had the most signal left, while the lane 3 the least. This background correction is now also explained in the Materials and methods on p. 26.

However, as the authors agree that the blot in the original Suppl. Fig. 1 was not optimal, we have repeated the experiment and now provide new blots as well as quantification of more repetitions in new Suppl. Fig. 4G-H. ---

Similarly, it is very difficult to imagine how the WBs on Suppl. Fig. 4A have yielded the quantitative data shown in main Fig. 3G. Please explain?.

--- Please find the original quantification of data shown in original Suppl. Fig. 4A (new Suppl. Fig. 9A) with Image Studio Lite attached (Rebuttal Fig. 13). The quantitation in the manuscript is now shown as new Fig. 5E.

Rebuttal Figure 13. Densitometric quantification of western analysis shown in Suppl. Fig. 9A. Image Studio Lite was used to quantify the western signal from a developed X-ray film.

3 > Authors state that ErbB4 JM-b co-immunoprecipitates more STAT5b based on the experiment shown on Fig 1F. However, the figure shows IP of ErbB4 followed by WB with phospho STAT5b antibody. The phosphoSTAT5 antibody recognizes equally well both STAT5a and STAT5b. The stronger band in the JM-b IP could as well be STAT5A. The experiment should be done correctly, similar to the STAT5A experiment (Fig 1E).

--- The authors thank the Reviewer for locating this typing mistake (*pSTAT5b* instead of *STAT5b*) in the original Fig. 1F. Indeed the blot was probed with an antibody recognizing total STAT5b, similar to STAT5a experiment in the original Fig. 1E. The blot in the new Fig. 3F is now labeled correctly. ---

4 > Page 5: "...overexpression of ErbB4 JM-a promoted the translocation of STAT5b to caveolin-1-rich membrane microdomains in MDA-MB-468 cells (Supplementary

Fig. 1E-G), in which STAT5 is dephosphorylated”. Equivalent experiment for STAT5a should be presented for comparison - is overexpression of ErbB4-JMa driving STAT5a to a different location? Or STAT5a is also translocated to the caveolin-1-rich microdomains – but then it should get dephosphorylated like STAT5b?

--- We have chosen to omit this ancillary data to bring more focus to the manuscript as suggested by the Reviewers #1 and #3. Indeed the original Suppl. Fig. 1E-G served only as an additional indicator of differential regulation of the STAT5s by the JM-a and JM-b isoforms and we feel that the evidence to support the hypothesis is strong even in the absence of this additional data. ---

5 > The mass spectrometric data for the JM-a/b protein complexes is unacceptable in its current form. The variability in label-free quantitation is bigger compared to other mass spectrometric approaches. Therefore, it strictly requires minimum of 3-5 replicates and proper statistical analyses of the data in order to define with higher probability the differences between the samples. Most of the proteins presented in the manuscript as ErbB4 JM-a/JM-b interactors are based on a single experiment. Label-free quantitation based on 1-2 replicates is very dangerous and much of the observed differences can be false. And thereby many of the assumed interactors. Especially when combined with puny criteria for accepted changes. To convince the reader that this is not the case, the authors should present the overlap of the two experiments as well as correlation analysis, e.g. Pearson.

In this respect, 7 out of the 9 complexes shown as unique to JM-a are based on Grb2 and Src, and both proteins are described as exclusively interacting with the JM-a isoform (Fig 5A). Consulting with Suppl. Table, Src is found only in one of the two experiments and identified by merely 2 peptides (and we have no information what criteria were used for accepting peptide identifications, comment below!). The values given for Grb2 in the same experiment for Control:JM-a:JM-b are 15:14:13. How could these proteins be counted as exclusively interacting with the JM-a and dictating most of the conclusions from these experiments?

--- The authors agree about the limitations of the mass spectrometry data included in the previous version of the manuscript. To improve this, the interactome mass spectrometry experiments have now been repeated for three additional rounds, bringing the amount of biological replicates to four. As a consequence, the new Fig. 1A and new Suppl. Files 1-4 include now totally new data. The new Fig. 1A now depicts only interactions that were identified in each of the 4 experiments with an adjusted $P \leq 0.05$. New Suppl. Files 1-4 include all the protein quantification data about all the four experiments. Raw proteomics data has been uploaded to the PRIDE repository (<https://www.ebi.ac.uk/pride/> ; accession: PXD017783, PXD026546, PXD026617). ---

6 > What is currently done, in terms of mass spectrometric analyses, may be ok for cherry-picking individual candidates for follow-up experiments amongst the proteins with clear-cut differences between conditions. But accepting the current mass spectrometric analyses as bonafide evidence that all these proteins are in complex with ErbB4 will be naïve and wrong. Also, different criteria for “interactors” were used for the 2 datasets? The criteria for processing the data should be identical to avoid biases. In addition, no information is provided regarding peptide identifications: what search parameters were used, the criteria used for accepting peptide identification – FDR,

minimum peptide score, etc.? The reader is left with no other choice, but to trust that all is done correct! The analyses need to be performed in a minimum 3 replicates, data should be properly analyzed, and only reproducible & significant changes must be considered as ErbB4 JM-a/JM-b interactors. Otherwise, these sections should be removed from the manuscript.

--- The authors agree that the original ErbB4 JM-a and JM-b interactome mass spectrometry experiments were exploratory and as lone evidence not sufficient to claim differential coupling. No statistical significance was thus indicated in the original manuscript. The complex modelling was performed for visualization purposes and not intended to mislead the reader into considering the complexes as bonafide ErbB4 complexes.

The interactome mass spectrometry experiments have now been repeated for three additional rounds and conducted in the same facility as the first original experiment. The data are now also processed using the same criteria in all experiments.

We have now also amended the Materials and methods to include the requested technical details for the mass spectrometry data. ---

7 > Tables with identifications from the PHA-L pull-downs should be included in the supplementary as well.

--- We have now included the PHA-L pull down identifications as a new Suppl. File 7. ---

Reviewers' comments:

Reviewer #1 (Remarks to the Author):

This resubmission has been significantly re-organized, which has improved the focus to some degree. Substantial additional data have been added, including appropriate repeats and analysis of statistical significance. There remains, however, an over emphasis on defining a molecular mechanism to explain the findings, which dilutes the impact of the observations themselves. Overall, I still have concerns, as detailed below.

The mass spec data summarized in Figure 1A have not been well integrated with the rest of the manuscript. As such this figure is a distraction to the story rather than enhancing it.

The data in Figure 1B/C in MCF7 cells are convincing and the correlations of good statistical significance. The additional COS7 data in the Supplementary Figure 1 is less convincing and it is unclear what it adds.

The huge variation in the efficiency of pull down of TYK2 by ErbB4 JM-b casts doubt on the data in Figure 2A,E. In the representative blot show in Figure 2A, the lower expression level of ErbB4 JM-a relative to ErbB4 JM-b in lanes 2 and 3 reduce the impact of the more intense TYK2 pulldown. In general difference in expression levels of the transiently expressed proteins, in particular in the ErbB4, reduce impact of the blots. The inefficient pulldown of TYK2 by ErbB4 JM-b in MCF7 cells compared to MDA-MB-468s is also concerning. In this and other figures, some panels are extremely low signal with bands barely visible (Figure 7H for example). There is also a problem with alignment of the JAK2 panel in Figure 2G.

Figure 3 is improved by addition of statistical analysis, although the blots remain of poor quality, and equivalent pairs (Figure 3A,B and Figure 3 E,F) do not show the same panels.

I remain of the opinion that the MD simulations add nothing but distraction to this manuscript as does the long discussion of the role of a putative hydrogen bond between H631 and D632 (which is seen in only 3 of 6 ErbB4 chains in 3U7U). The analysis of the JM-a sequence motif is, however, substantially improved. There is suggestion of a correlation of presence of this amino acid signature in the eJM region and STAT5a activation. There is value to the mutational analysis in this region, but an over emphasis on the side chain interactions. If the loss of function observed with the H631E

mutation could be restored by a compensating alteration at D632 to restore an interaction then the detailed discussion of this interaction might be justified.

It is highly inappropriate to subtitle a section discussion the structures as “Differences in the crystal structures of ErbB4 JM isoforms”. There is no crystal structure of the JM-b isoform.

Figure 4E lacks adequate repeats, as do several other panels.

As it stands the manuscript is very long. The middle section of the manuscript could be largely removed (all the MD simulation and most of the discussion of structure – from line 238 -321). This would then afford opportunity to link the findings in a more direct manner. It is disappointing that there is no link back to the mass spectrometry-derived interactomes based on glycan interaction data.

Reviewer #3 (Remarks to the Author):

It appears the manuscript is much improved in quality by the incorporation of additional data (particularly the immunoblotting results) and by its reorganization. I now find it suitable for publication.

Reviewer #4 (Remarks to the Author):

In the original version I was asked to concentrate on reviewing the imaging parts of the work. Regarding the latter, I am afraid I remain unimpressed by what the authors have done (or rather not done). The original STED figures, which did not make sense, are still there. One cannot measure crowding in x,y looking in projection at x,z. When one does this, everything would seem to colocalise with everything. The STED images remain therefore not useful for the purpose of this manuscript, simply because the orientation is incorrect.

STED could be the right technique, but, to be useful, it needs to be at the right orientation.

The new SIM data has been collected at the right orientation, but to me the results seem inconclusive. I imagine that is the reason the authors did not remove the STED images in the wrong orientation. But this does not salvage the results in my view.

Regarding chromatic aberration, this is not the product of laser misalignment.

Unfortunately, I remain of the opinion that this paper is still not ready for publication.

Reviewer #5 (Remarks to the Author):

This reviewer is satisfied with the revision.

Response to Reviewers

Comments received from the Reviewers on 8th of September 2021 are included in plain font, followed point-by-point by the authors' comments in bold.

Reviewer #1 (Remarks to the Author):

This resubmission has been significantly re-organized, which has improved the focus to some degree. Substantial additional data have been added, including appropriate repeats and analysis of statistical significance. There remains, however, an over emphasis on defining a molecular mechanism to explain the findings, which dilutes the impact of the observations themselves. Overall, I still have concerns, as detailed below.

We thank the reviewer for the encouraging words. As explained below, the manuscript has been further restructured to reach a better level of integration and to highlight the key findings.

The mass spec data summarized in Figure 1A have not been well integrated with the rest of the manuscript. As such this figure is a distraction to the story rather than enhancing it.

We appreciate this suggestion and agree with the Reviewer's comment. Prompted by this insightful comment, the manuscript has been reorganized to better integrate the mass spectrometry data. The original Figure 1A is now presented as new Suppl. Fig. 2. Text has also been rewritten to reduce the focus on these data. The mass spectrometry data now serves as a basis that leads the paper to the JAK/STAT pathway and, subsequently, to the mechanisms by which different RTKs activate this pathway.

To further integrate the mass spectrometry data, we now also provide new mass spectrometry data on a receptor kinome level. These new data indicate that, indeed, the JM-a-like RTKs versus the JM-b-like RTKs can be clustered into two distinct groups with unique molecular interactions and locations (new Fig. 5D, new Suppl. File 6-7, new Suppl. Fig. 11). The observation of a significant overlap in the location enrichment analysis of the interactions derived from the original mass-spectrometry data, restricted to ErbB4 isoforms, with the new data covering a large number of human RTKs (new Suppl. Fig. 11) provides better integration of the original mass spectrometry analyses with the current presentation.

The data in Figure 1B/C in MCF7 cells are convincing and the correlations of good statistical significance. The additional COS7 data in the Supplementary Figure 1 is less convincing and it is unclear what it adds.

The additional data shown in Suppl. Fig. 1A was added to the manuscript to address previous concerns of Reviewer #4 regarding the orientation of the super-resolution confocal images.

These data in the modified Suppl. Fig. 1 adds the following:

- **The difference in the plasma membrane localization between ErbB4 JM-a and JM-b is evident on cell surface compartments representing both cell periphery as well as the bottom of the cells.**
- **Since structured illumination microscopy (SIM) allows for 3D imaging, which is not feasible with STED due to signal bleaching, the signals could be analyzed in all X-Y, X-Z and Y-Z orientations. This provides additional documentation that the observations are valid in 3D space and do not arise from incorrect 2D projections of the 3D signals (a concern raised earlier by the Reviewer #4).**
- **The initial observation obtained using the MCF-7 cells are reproducible in another cell line, i.e. COS-7 cells, and with differently tagged ErbB4 constructs (eGFP instead of MYC-tag).**

The huge variation in the efficiency of pull down of TYK2 by ErbB4 JM-b casts doubt on the data in Figure 2A,E. In the representative blot shown in Figure 2A, the lower expression level of ErbB4 JM-a relative to ErbB4 JM-b in lanes 2 and 3 reduce the impact of the more intense TYK2 pulldown.

The variation in the original Fig. 2E was a result of quantification and normalizing the signals across independently performed Western analyses that are semiquantitative by nature. The authors would like to emphasize that the results of all the independent experiments were consistent with our conclusion and regardless of the variation, statistical significance at the level of $P \leq 0.05$ was reached.

A particular phenomenon related to the experimentation with ErbB4 JM-a isoforms is that the receptor is cleaved producing the 80 kD intracellular domain (ICD) fragment, and the extent to which this 80 kD ICD is cleaved and precipitates with TYK2 is sensitive to cell culture conditions and thus highly variable. In the quantifications shown in the original Fig. 2E, data were shown separately for the full-length 180 kD ErbB4 and the 80 kD ICD fragment, producing variation particularly for the 80 kD ErbB4 JM-a fragment.

We have now carried out additional co-precipitation analyses, and to avoid the additional level of complexity and variation resulting from the separate analysis of the 80 kD fragment, focused only on comparing the full-length JM-a versus JM-b. As indicated by the blots shown below (Rebuttal Fig. 1; quantification of the same data is shown in the

new Fig. 1D), the formation and co-precipitation of the 80 kD JM-a fragment with TYK2 is variable but, importantly, this variation does not influence the conclusion that the 180 kD full-length JM-b is reproducibly in each experiment more efficiently pulled down with TYK2 as compared to full-length 180 kD JM-a.

Rebuttal Figure 1. Co-immunoprecipitation analyses of TYK2 and ErbB4 JM isoforms in MCF-7 cells overexpressing GFP-tagged TYK2 and HA-tagged ErbB4 JM-a or JM-b after NRG-1 stimulation. Blots presented in the original Fig. 2A are shown in panel A. Blots presented in the new Fig. 1C are shown in panel B. Densitometric quantitation of all the four analyses is visualized in the new Fig. 1D.

The variation of the relative intensity values shown in the new Fig. 1D after median normalization without the ErbB4 JM-a ICD fragment quantification is now significantly reduced as compared to the original Fig. 2E. We have additionally changed the representative blot shown in original Fig. 2A with the new Fig. 1C to present an example of experimental settings in which the 80 kD JM-a fragment was not present in significant quantities.

The authors are unsure what the Reviewer is referring to with the “lower expression level of ErbB4 JM-a relative to ErbB4 JM-b in lanes 2 and 3” in original Fig. 2A. The amount of full-length ErbB4 JM-a and JM-b in the experiment was very similar (103,000 densitometric units for JM-a versus 104,000 for JM-b; Rebuttal Fig. 2). If one takes into account the amount of cleaved ErbB4 ICD fragment there is even more total ErbB4 JM-a in the cells than ErbB4 JM-b ($103,000 + 57,400 = 160,400$ for JM-a versus 104,000 for JM-b; Rebuttal Fig. 2; panel A representing original Fig. 2A). The quantification of all the four experiments used for the new Fig. 1D are now also shown in the Rebuttal Fig. 2. The data do not indicate any significant differences in the expression of ErbB4 JM-a relative to ErbB4 JM-b.

Rebuttal Figure 2. Densitometric quantification of the expression levels of ErbB4 JM-a and ErbB4 JM-b from co-immunoprecipitation analyses of TYK2 and ErbB4 JM isoforms in MCF-7 cells. The quantification was performed with Image Studio Lite. Panel A represents the anti-HA (ErbB4) Western shown in the original Fig. 2A. Panel B represents the anti-HA (ErbB4) Western shown in the new Fig. 1C.

In general difference in expression levels of the transiently expressed proteins, in particular in the ErbB4, reduce impact of the blots.

We appreciate this concern. However, as shown in the quantifications provided below in Rebuttal Fig. 3, there is no significant difference of the expression of ErbB4 JM-a and JM-b in these cell lines. Furthermore, we see similar fluctuations on the ErbB4 expression levels in the transiently transfected (MCF-7, COS-7) as well as in the stably expressing virally transduced cell lines (MDA-MB-468, HC11) so based on our experience the variation is not necessarily linked to transient vs. stable expression. We opted mainly for the transient transfections because the stable exogenous expression tends to change the cell lines over time making them less comparable after a few passages. By using transient transfection we were able to ensure that the cell background is equal in the experiments.

Rebuttal Figure 3. The expression level of ErbB4 JM-a and JM-b in the Western analyses in transiently transfected (MCF-7, COS-7) and virally transduced (HC11, MDA-MB-468) cell lines. The quantification was performed with Image Studio Lite and the data normalized with signals derived from loading control genes. The values were median normalized.

The inefficient pulldown of TYK2 by ErbB4 JM-b in MCF7 cells compared to MDA-MB-468s is also concerning.

The authors are a little unsure what this comment is referring to. The only experiments presented in the manuscript in which TYK2 was used as a bait to pull down ErbB4 isoforms were carried out with MCF-7 cells and shown in the original Fig. 2A and E (new Fig. 1C and D). No similar experimentation was conducted using MDA-MB-468 cells. On the other hand, TYK2 was identified to specifically associate with ErbB4 JM-b in the anti-ErbB4 immunoprecipitates from MDA-MB-468 cells in mass spectrometry analysis shown in the original Fig. 1A (new Suppl. Fig. 2). No similar experimentation was conducted using MCF-7 cells. Given the significant differences between the experimental set-ups, and the lack of direct side-by-side comparison, it is very difficult to draw conclusions about differences in the efficiency of the TYK2-ErbB4 association between the two cell types.

In this and other figures, some panels are extremely low signal with bands barely visible (Figure 7H for example).

Our original aim was to present blots with signals at a quantitative range, especially when acquired using exposure on X-ray films. We have now paid more attention to the visualization of the panels and replaced the co-immunoprecipitation experiment in the original Fig. 7H with a proximity ligation (PLA) experiment (new Fig. 6H and I). These new PLA data confirm the original finding that the association of ErbB4 JM-a with β 1-integrin is sensitive to lectins and a mutation at the ErbB4 JM-a residue H631.

There is also a problem with alignment of the JAK2 panel in Figure 2G.

We thank the reviewer for bringing this to our attention and apologize for the mistake. The blot was accidentally horizontally flipped. The alignment has been corrected into the new Fig. 1E.

Figure 3 is improved by addition of statistical analysis, although the blots remain of poor quality, and equivalent pairs (Figure 3A,B and Figure 3 E,F) do not show the same panels.

To improve the presentation, and to obtain additional independent data supporting the hypothesis we have:

- **Unified the loading control blots of the original Fig. 3A and B to both show actin expression. These are now shown as new Fig. 2A and B.**
- **Replaced the co-immunoprecipitation experiments shown in the original Fig. 3E and F with proximity ligation assays (PLA). The PLA experiments are now shown**

as new Fig. 2E-H. These new data again provide an independent validation of the conclusion that ErbB4 JM-a preferably associates with STAT5a and ErbB4 JM-b with STAT5b.

I remain of the opinion that the MD simulations add nothing but distraction to this manuscript as does the long discussion of the role of a putative hydrogen bond between H631 and D632 (which is seen in only 3 of 6 ErbB4 chains in 3U7U).

We are grateful for this suggestion. The section of the manuscript describing the structural data and MD simulations has now been edited and shortened. Focus is given to describing the structural differences between the two isoforms, observation from the MD simulation that suggests minimal local conformational difference at the region of sequence variation between the isoforms, and the likely role differences in amino acid interactions might play in STAT5 activation.

The analysis of the JM-a sequence motif is, however, substantially improved. There is suggestion of a correlation of presence of this amino acid signature in the eJM region and STAT5a activation. There is value to the mutational analysis in this region, but an over emphasis on the side chain interactions. If the loss of function observed with the H631E mutation could be restored by a compensating alteration at D632 to restore an interaction then the detailed discussion of this interaction might be justified.

We appreciate this criticism and agree. We have now amended the manuscript text to reduce the emphasis on the side chain interactions. Please see the modified text on p. 8 and 12 in the *Differences in the structural models of ErbB4 JM isoforms and Mutational analysis of the RTK eJM motif sections*.

It is highly inappropriate to subtitle a section discussion the structures as “Differences in the crystal structures of ErbB4 JM isoforms”. There is no crystal structure of the JM-b isoform.

Thank you for pointing this out. This was unintentional and we apologize for the use of an inaccurate expression. The subtitle has now been changed to “Differences in the structural models of ErbB4 JM isoforms”.

Figure 4E lacks adequate repeats, as do several other panels.

We have now produced additional repetitions for the experiments shown in the original Fig. 4E (now new Fig. 3E), original Fig. 5I,J (new Fig. 4 I,J), original Fig. 6B (new Fig. 5B), original Fig. 7C,D (new Fig. 6C,D), and original Suppl. Fig. 6B (new Suppl. Fig. 7B). All the quantification panels now have at least three independent repeats of all analyses.

As it stands the manuscript is very long. The middle section of the manuscript could be largely removed (all the MD simulation and most of the discussion of structure – from line 238 -321). This would then afford opportunity to link the findings in a more direct manner.

The structural and MD parts of the manuscript have been reorganized and trimmed to deliver a more focused presentation.

It is disappointing that there is no link back to the mass spectrometry-derived interactomes based on glycan interaction data.

We have now conducted a pathway analysis on the mass spectrometry-derived interactomes of ErbB4 JM-a and JM-b (new Suppl. Fig. 3) and discovered that two pathways related to $\beta 1$ integrin (PID_INTEGRIN_A9B1_PATHWAY and PID_INTEGRIN_A4B1_PATHWAY) are significantly differentially regulated by ErbB4 JM-a and ErbB4 JM-b. We have now also used this finding to link the mass spectrometry-derived interactomes to the glycan interaction data in the manuscript text.

$\beta 1$ integrin was also detected in one mass spectrometry analysis of the interactome of ErbB4 JM-a which is why we utilized targeted mass spectrometry to further address the preferred interaction between $\beta 1$ integrin and ErbB4 JM-a isoform. The results of this targeted mass spectrometry analysis were presented in the original Suppl. Fig. 13A-C and are now shown in new Suppl. Fig. 15A-C. The fact that we were able to find consistently higher levels of $\beta 1$ integrin in ErbB4 JM-a immunoprecipitates than in the ErbB4 JM-b immunoprecipitates with targeted mass spectrometry indicates that $\beta 1$ integrin was also present in the non-targeted interactome samples but was masked due to the complex mixture of peptides. It could be speculated that the glycan-protein interaction between $\beta 1$ integrin and ErbB4 JM-a is not as strong as a protein-protein interaction between e.g. ErbB4 JM-b and TYK2, which might be why it was not readily detected by the non-targeted MS analysis.

To further control the specific association of ErbB4 JM-a and $\beta 1$ integrin we now provide new PLA data validating once again that the two proteins associate with each other and that the association is sensitive to the presence of lectins and significantly reduced when the H631E mutant ErbB4 JM-a is analyzed (new Fig. 6H and I).

Reviewer #3 (Remarks to the Author):

It appears the manuscript is much improved in quality by the incorporation of additional data (particularly the immunoblotting results) and by its reorganization. I now find it suitable for publication.

We thank the reviewer for the encouraging words and helping us improve the manuscript.

Reviewer #4 (Remarks to the Author):

In the original version I was asked to concentrate on reviewing the imaging parts of the work. Regarding the latter, I am afraid I remain unimpressed by what the authors have done (or rather not done). The original STED figures, which did not make sense, are still there. One cannot measure crowding in x,y looking in projection at x,z. When one does this, everything would seem to colocalise with everything. The STED images remain therefore not useful for the purpose of this manuscript, simply because the orientation is incorrect. STED could be the right technique, but, to be useful, it needs to be at the right orientation. The new SIM data has been collected at the right orientation, but to me the results seem inconclusive. I imagine that is the reason the authors did not remove the STED images in the wrong orientation. But this does not salvage the results in my view.

The authors apologize for failing to comprehensively communicate the visualization of the original STED figures. As we attempted to indicate in our previous Response to Reviewers letter, all the STED images were 2D STED images indeed obtained in the X-Y orientation (and not X-Z orientation). We have now paid particular attention to the presentation and replaced the STED images with new ones indicating clearly the x-y axis:

- **Original Fig. 1B and C are now replaced by new Fig. 1A and B.**
- **Original Fig. 7I and J are now replaced by new Fig. 6J and K.**
- **Original Suppl. Fig. 8 is now replaced by new Fig. 3F and G.**
- **All new panels with STED images include zoom-ins with higher magnification of regions of interest, as well as lower magnification images showing the cellular areas representing the higher magnified areas.**
- **In all STED images, the X-Y orientation is now clearly indicated by arrows.**
- **New data has been quantified and statistically analyzed producing results supporting the initial conclusions.**

The authors are a little uncertain about the specific reasoning behind the Reviewer's comment "the SIM results seem inconclusive". We assume this refers to the fact that there was no significant difference in the colocalization between ErbB4 JM-a and JM-b specifically at the top of the cells (while there was a significant difference in both the peripheral and bottom cell surfaces). This is indeed, a valid concern. We consulted an expert on SIM microscopy and discovered that the resolution of the SIM greatly diminishes at distances further away from the coverslip. Our top cell surface images were acquired from a distance where this is already a major limitation. Therefore, we have omitted the imaging and quantification of the top of the cell and only focused images obtained from the peripheral and bottom cell surfaces (new Suppl. Fig. 1; new Suppl. Fig. 15D-G) and are confident the SIM data presented in the manuscript is conclusive, reproducible and consistent with the STED imaging.

Regarding chromatic aberration, this is not the product of laser misalignment.

We fully agree. Sorry for the confusion. What we had wanted to communicate was that chromatic aberration can be corrected by laser alignment or by computational signal alignment.

Unfortunately, I remain of the opinion that this paper is still not ready for publication.

We hope the newly revised manuscript is now suitable for publication.

Reviewer #5 (Remarks to the Author):

This reviewer is satisfied with the revision.

We thank the reviewer for the valuable contribution in helping us improve the manuscript.

Reviewers' Comments:

Reviewer #1:

Remarks to the Author:

In this revised version of their manuscript, Vaparanta et al have further re-organized the work and reinforced interpretations with additional repeats and quantitation. The leading results on differences in signaling from the JM-a and JM-b isoforms are substantially more compelling.

The structural questions and MD simulations are now appropriately discussed and used to guide the next experiments.

The extension of the interaction analysis to consider a range of RTKs with JM-a- or JM-b-like eJMs (Fig. 5D) is a useful addition.

The presentation and interpretation of the data supporting a role for glycan interactions is enhanced significantly in this version of the manuscript.

The discussion is long and is largely a re-statement of the results adding very little new information.

Reference to panels in Figure 1 is incorrect (1B when it should be 1C etc).

Reviewer #4:

Remarks to the Author:

Questions to authors:

1) Though the authors specify the secondary antibodies used for STED (all suitable), they don't provide information about which depletion lasers they've used, or whether they've used more than one. This could affect co-localisation data, although in every co-localisation analysis they compare ErbB4 co-localisation with the same isoform (a with a or b with b) as a control, so potential

chromatic aberrations due to multiple depletion lasers would be corrected if the fluorophore combination is always the same. A comment on this in the manuscript would be beneficial.

2) The resolution achieved doesn't seem drastically better than that of a confocal, probably the best justification for using STED would be in Fig6. The authors don't provide much information regarding size of clusters or domains at the plasma membrane or other compartments so it's hard to get an idea of how much these samples have benefited from STED imaging. This is not a problem in itself, but I wonder if a different labelling approach would improve the spatial discrimination of the isoforms. They expressed tagged proteins (Myc or HA), and fixed and labelled those with anti-Myc or anti-HA, and then used a secondary antibody on top of that. I wonder if using nanobodies or other tags (such as Halo) or other combinations, would have reduced the size of the imaging target and would have better resolved the structures in the images.

3) Methanol is very harsh on the membrane. Why is this preferable to using PFA or PFA/glutaraldehyde (the antibodies they use all work with PFA fixation) and either low concentration of Triton X-100 or saponins, in order to better maintain the plasma membrane structure and other membrane compartments. If it's a matter of epitope accessibility for antibodies all the more reason to use fluorescent proteins or Halo/snap tags for the different isoforms.

Response to Reviewers

Comments received from the Reviewers on 12th of October 2022 are included in plain font, followed point-by-point by the authors' comments in bold.

Reviewer #1 (Remarks to the Author):

In this revised version of their manuscript, Vaparanta et al have further re-organized the work and reinforced interpretations with additional repeats and quantitation. The leading results on differences in signaling from the JM-a and JM-b isoforms are substantially more compelling.

The authors wish to thank the Reviewer for the comment.

The structural questions and MD simulations are now appropriately discussed and used to guide the next experiments.

The authors wish to thank the Reviewer for the comment.

The extension of the interaction analysis to consider a range of RTKs with JM-a- or JM-b-like eJMs (Fig. 5D) is a useful addition.

The authors wish to thank the Reviewer for the comment.

The presentation and interpretation of the data supporting a role for glycan interactions is enhanced significantly in this version of the manuscript.

The authors wish to thank the Reviewer for the comment.

The discussion is long and is largely a re-statement of the results adding very little new information.

The authors wish to thank the Reviewer for pointing this out. The Discussion has now been shortened to remove redundancy with results.

Reference to panels in Figure 1 is incorrect (1B when it should be 1C etc).

The authors apologize for the mistake. This has been corrected.

Reviewer #4 (Remarks to the Author):

Questions to authors:

1) Though the authors specify the secondary antibodies used for STED (all suitable), they don't provide information about which depletion lasers they've used, or whether they've used more than one. This could affect co-localisation data, although in every co-localisation analysis they compare ErbB4 co-localisation with the same isoform (a with a or b with b) as a control, so potential chromatic aberrations due to multiple depletion lasers would be corrected if the fluorophore combination is always the same. A comment on this in the manuscript would be beneficial.

The authors wish to thank the Reviewer for the comment. For the initial STED experiments, the 580 nm pulsed and 775 nm continuous depletion lasers were used with the Alexa-488 and Abberior-STAR-635 probes, respectively. In the later experiments, only the 775 nm continuous depletion laser was used with Abberior-STAR-595/Alexa-555 and Abberior-STAR-635 probes as per suggestion by the Imaging Core of Turku Bioscience. This is now specified in the Materials and Methods section. Please note that we acquired similar results with both depletion laser and probe combinations and as the Reviewer insightfully pointed out, all experimentation included the control of the same ErbB4 isoform coupled to different epitope tags. The samples were always compared within experiment and not across experiments and the images of one experiment were acquired within the same imaging session with the same settings.

2) The resolution achieved doesn't seem drastically better than that of a confocal, probably the best justification for using STED would be in Fig6. The authors don't provide much information regarding size of clusters or domains at the plasma membrane or other compartments so it's hard to get an idea of how much these samples have benefited from STED imaging. This is not a problem in itself, but I wonder if a different labelling approach would improve the spatial discrimination of the isoforms. They expressed tagged proteins (Myc or HA), and fixed and labelled those with anti-Myc or anti-HA, and then used a secondary antibody on top of that. I wonder if using nanobodies or other tags (such as Halo) or other combinations, would have reduced the size of the imaging target and would have better resolved the structures in the images.

The Reviewer correctly points out that the differences in the subcellular localization of ErbB4 JM-a and ErbB4 JM-b could also be seen with confocal imaging. The improvement in the resolution at the X-Y axis we were able to achieve with 2D STED compared to confocal imaging with the used probe and depletion laser combinations is now visualized in Rebuttal Figure 1. The authors also agree that the utilization of some

new technologies like nanobodies or other tags could have provided an improvement in the resolution of the clusters. We used the MYC-tagged and HA-tagged ErbB4 JM-isoforms as we were confident from our previous work that comparable levels of ErbB4 expression and immunofluorescence signal could be achieved with these constructs.

Rebuttal Figure 1. Resolution improvement achieved with STED.

A: MCF-7 cells expressing ErbB4 JM-a HA and ErbB4 JM-b Myc were probed with HA- and Myc antibodies and Alexa-488 and Abberior STAR 635 secondary antibodies. The stained cells were imaged both with confocal and STED microscopy simultaneously. The 580 nm pulsed and 775 nm continuous depletion lasers were used.

B: MCF-7 cells expressing ErbB4 JM-a HA and ErbB4 JM-b Myc were probed with HA- and Myc antibodies and Alexa-555 and Abberior STAR 635 secondary antibodies. The stained cells were imaged both with confocal and STED microscopy simultaneously. The 775 nm continuous depletion laser was used.

3) Methanol is very harsh on the membrane. Why is this preferable to using PFA or PFA/glutaraldehyde (the antibodies they use all work with PFA fixation) and either low concentration of Triton X-100 or saponins, in other to better maintain the plasma membrane structure and other membrane compartments. If it's a matter of epitope accessibility for antibodies all the more reason to use fluorescent proteins or Halo/snap tags for the different isoforms.

The authors wish to thank the Reviewer for the insightful comment. Our aim was to use the same conditions to fix the STED samples as what were used for the other immunofluorescence analyses for comparability of results. In particular, some of the STAT5 IF stains did not work properly with 4% PFA fixation and Triton-X 100

permeabilization. Less background was also observed in the STED imaging with methanol as compared to PFA fixation. Moreover, we had already prior to conducting the STED analyses noticed that methanol fixation did not seem to adversely affect the localization of ErbB4 staining.